# Performance Prediction In Reinforcement Learning: The Good, The Bad And The Ugly

## Abstract

Reinforcement learning (RL) methods are known to be highly sensitive to their hyperparameter settings and costly to evaluate. In light of this, surrogate models that predict the performance of a given algorithm given a hyperparameter configuration seem an attractive solution for understanding and optimising these computationally expensive tasks. In this work, we are studying such surrogates for RL and find that RL methods present a significant challenge to current performance prediction approaches. Specifically, RL hyperparameter landscapes appear to be rugged and noisy, which leads to the poor quality of surrogate models. Even if surrogate models are only used for gaining insights into the hyperparameter landscapes and not as replacements for algorithm evaluations in benchmarking, we find that they deviate substantially from the ground truth. While our evaluation highlights the limits of surrogate modelling for RL, we propose a method for automatically reducing configuration spaces for improved surrogate performance. We also derive recommendations for RL practitioners that caution against blindly trusting surrogate-based methods for this domain and highlight where and how they can be used.

## 1 Introduction

Surrogate models are used in different ways in automated machine learning (AutoML) research and practice: they can significantly reduce the cost of benchmarking, especially in domains with high computational cost (Eggensperger et al., 2015; 2018a; Klein et al., 2019; Eggensperger et al., 2021; Zela et al., 2022), and help with understanding the optimisation landscape of a given target algorithm via analysis methods such as fANOVA (Hutter et al., 2014a) or HyperSHAP Wever et al. (2026). A domain that could significantly benefit from the cost reduction as well as the insights gained from using surrogates is reinforcement learning (RL), where hyperparameter optimisation is not yet well studied, partly due to the high cost of evaluation.

Recent work in RL re-emphasised how irregular the behaviour of RL algorithms can appear. Even on a well-studied benchmark such as the arcade learning environment (ALE; Bellemare et al. (2013); Machado et al. (2018)), it is unclear which algorithm is best for a given environment in the suite (Farebrother & Castro, 2024). Furthermore, hyperparameter configurations behave inconsistently across environments, with no clear pattern emerging as of yet (Ceron et al., 2024; Patterson et al., 2025). Given the immense computational cost of executing repeated runs of a benchmark such as ALE, the use of surrogates could significantly accelerate the development of algorithms as well as model selection and hyperparameter optimisation (HPO) methods. They could also facilitate deeper analysis of why we see such irregular behaviour in RL algorithms.

The computational costs of RL, coupled with its notorious difficulty of tuning hyperparameters, does not suggest an easy setting for performance prediction. Previous work has underscored this by showing that hyperparameter optima can move significantly during RL training (Mohan et al., 2023). Indeed, we find that fitting surrogate models on the comparatively large ARLBench dataset (Becktepe et al., 2024) containing numerous commonly used RL tasks results in rather poor predictions. In this work, we investigate possible reasons for this phenomenon, attempt to boost surrogate quality and assess how much surrogate models can contribute to hyperparameter interpretability. Contextualising our results with data from supervised learning tasks from the LCBench (Zimmer et al., 2021) and PD1 (Wang et al., 2024) benchmarks paints the picture of

a comparatively complex and noisy hyperparameter landscape far from what Pushak & Hoos (2018) found to be benign landscapes for supervised machine learning methods. Our evaluations suggest that vast amounts of data are needed to fit reliable RL surrogates; even if we are merely interested in local hyperparameter importance (LPI; Biedenkapp et al. (2018)) or ablation paths (Fawcett & Hoos, 2016; Biedenkapp et al., 2017), surrogates are not consistently showing the same qualitative behaviour as the algorithms whose performance they model. As a remedy, we introduce a practical methodology based on HyperShap (Wever et al., 2026) that allows a systematic reduction to a subset of RL hyperparameters that can be reliably modelled by surrogates. For insights into the full RL hyperparameter landscape, we recommend not relying on surrogates for now, but rather using methods that do not involve performance prediction. At the same time, we believe RL to be an interesting yet challenging domain in the context of improving surrogate modelling methods.

In summary, our contributions are: (I) an investigation of the difficulty of the hyperparameter landscapes of RL, particularly in terms of unimodality and the influence of randomness; (II) a thorough evaluation of surrogate model quality on RL tasks, both in terms of benchmarking performance and their use in surrogate-based hyperparameter interpretability methods; (III) an automatic configuration space reduction method to boost surrogate quality and scalability; and (IV) guidelines for the use of surrogate models in RL.

To foster reproducibility, we will open-source all our code and data in a public repository after publication.

## 2 Background

To provide a foundation for our empirical study, this section introduces the RL framework and formalises the central concepts of performance prediction, surrogate modelling and hyperparameter landscapes.

**Reinforcement Learning** We consider the standard RL formulation, wherein an agent interacts with an environment modelled as a Markov Decision Process (MDP) defined by $\mathcal{M} := (\mathcal{S}, \mathcal{A}, p, r, \rho_0)$ (Sutton & Barto, 2018). At each time step $t$, the agent observes a state $s_t \in \mathcal{S}$, executes an action $a_t \in \mathcal{A}$ sampled from a policy $\pi(a \mid s)$, and transitions to a state $s_{t+1} \sim p(\cdot \mid s_t, a_t)$, receiving a scalar reward $r_t = r(s_t, a_t)$. The objective is to find a policy maximising the expected cumulative return $\mathbb{E}_{\tau \sim \pi} \sum_{t=0}^{\infty} \gamma^t \cdot r(a_t, s_t)$, where $\tau = (s_0, a_0, s_1, a_1, \dots)$ denotes a trajectory generated by the policy $\pi$ starting from the initial state distribution $s_0 \sim \rho_0$ and $\gamma \in [0, 1]$ is the discount factor. Common RL algorithms are configured by various hyperparameters that determine the training behaviour and are often crucial for training success.

**Performance Prediction and Surrogate Models** We formulate the task of *RL performance prediction* as a supervised regression problem. Let $\mathcal{A}_\lambda$ denote an RL algorithm parametrised by a hyperparameter configuration $\lambda \in \Lambda$, where $\Lambda$ is the configuration space. The true performance of the algorithm $\mathcal{A}$ for and environment $\mathcal{M}$ (formalised as an MDP) is a function $f_{\mathcal{A}, \mathcal{M}} : \Lambda \to \mathbb{R}$, which yields a scalar performance metric $p$, typically the expected undiscounted return. We refer to the mapping $f_{\mathcal{M}, \mathcal{A}} : \Lambda \to \mathbb{R}$ for a fixed environment and algorithm as the *hyperparameter performance landscape* or short *hyperparameter landscape*. Because evaluating $f_{\mathcal{A}, \mathcal{M}}(\lambda)$ requires running the complete training process for multiple seeds, it is computationally expensive. A *surrogate model* $\hat{f}_{\mathcal{A}, \mathcal{M}} : \Lambda \to \mathbb{R}$ is a computationally cheap approximation of the true performance function $f$ for an algorithm and environment combination. In practice, surrogate models are fitted on a dataset of previously observed hyperparameter configurations with their respective training performance. Subsequently, such a surrogate model allows for cheaply evaluating the performance of unseen hyperparameter configurations, assuming good generalisation performance. Further, surrogate models can be used to obtain more general insights into hyperparameter landscapes by using hyperparameter importance metrics or studying the ruggedness of the landscape.

## 3 Related Work

Before discussing our investigation, we will give an overview of the context of our work within AutoRL, the construction of surrogate benchmarks, and surrogates for interpreting hyperparameter landscapes.

**AutoRL:** Automated reinforcement learning (AutoRL; Parker-Holder et al., 2022) is concerned with automatically adapting RL algorithms to any given setting. This includes HPO (Parker-Holder et al., 2020) just

as much as neural architecture search (Miao et al., 2022), task scheduling (Portelas et al., 2020; Jiang et al., 2021) and more. Notably, several previous studies have found unexpected RL algorithm behaviour concerning hyperparameters, e.g. related to noise between different runs (Henderson et al., 2018; Eimer et al., 2023) or significant differences between algorithms (Farebrother & Castro, 2024). To further explore this and to enable principled comparisons of HPO approaches for RL, the benchmarks HPO-RL-Bench (Shala et al., 2024) and ARLBench (Becktepe et al., 2024) have been recently proposed. We use both as data sources for our surrogates.

**Surrogate-Based Benchmarking:** Since the study of HPO is inherently costly, due to the need for many repeated function evaluations, the machine learning community has started publishing datasets containing evaluation results as a fast and efficient way for benchmarking: so-called tabular benchmarks. Notable examples for machine learning tasks include NASbench 101 (Ying et al., 2019) and 201 (Dong & Yang, 2020), HPO-B (Pineda et al., 2021), LCBench (Zimmer et al., 2021) and the recent PD1 (Wang et al., 2024). However, since these datasets are restricted in the number of configurations they cover, their utility remains limited. Thus, surrogate models capable of predicting the performance of configurations not in a given dataset have been proposed (Hutter et al., 2014b; Eggensperger et al., 2015; 2018a). Surrogate-based benchmarks are now important measures of HPO performance, e.g. as part of HPOBench (Eggensperger et al., 2021), in NASBench 301 (Zela et al., 2022) or YAHPOGym (Pfisterer et al., 2022). Surrogate modelling techniques are also continuously improved upon through ideas such as the direct modelling of the distribution of performance over multiple independent runs (Eggensperger et al., 2018b; Li et al., 2024). While in benchmarking, surrogate models are predominantly used to obtain individual performance values, there are also approaches for predicting full learning curves from either a configuration or incomplete runs (Klein et al., 2017; Adriaensen et al., 2024). While not yet applied in benchmarking, similar techniques have been suggested for HPO in RL (Nguyen et al., 2020; Shala et al., 2023).

**Surrogates for HPO Interpretability:** Surrogate models are also used for gaining insights into the behaviour of target algorithms; they can assist by simulating the evaluation of additional configurations for approaches such as ablation paths (Biedenkapp et al., 2017), local hyperparameter importance (Biedenkapp et al., 2018), HyperShap (Wever et al., 2026) or partial dependency plots (Moosbauer et al., 2021). Some specific surrogate model classes, such as random forests, can even be directly used to obtain insights into hyperparameter importance (Hutter et al., 2014a). Systems such as DeepCave (Sass et al., 2022) provide several such tools packaged together, such that even HPO runs with a constrained budget can easily provide meaningful insights into the response of a given target algorithm to hyperparameter settings.

## 4 Setup of Experiments

Before discussing the quality of surrogate models in RL, we describe the setup we used for training surrogate models, including datasets and training procedure.

### 4.1 Datasets

To analyse and compare the quality of surrogate models, we used two AutoRL benchmarks with public datasets mapping hyperparameter configurations to training performance on several RL environments. In both cases, the performance metric is the undiscounted return of the resulting policy.

1. **ARLBench** (Becktepe et al., 2024) is a hyperparameter optimisation benchmark that includes performance landscapes for three RL algorithms: PPO (Schulman et al., 2017), DQN (Mnih et al., 2013) and SAC (Haarnoja et al., 2018). These landscapes cover 21, 13 and 8 different environments and contain 9, 12 and 11 hyperparameters, respectively. Each landscape is based on 256 configurations, with 10 independent runs per configuration to account for performance variability.

2. **HPO-RL-Bench** (Shala et al., 2024) is a tabular benchmark for HPO in RL that is based on grids of precomputed hyperparameter configurations. The benchmark includes the algorithms PPO, DQN and SAC, among others, on 22 different environments. Due to the curse of dimensionality, it is limited

to three hyperparameters per algorithm on a $6 \times 6 \times 3$ grid, amounting to 108 unique configurations. Each configuration was evaluated on 10 seeds.

In addition to these RL datasets, we also include two datasets from supervised learning, to contextualise our RL results using a domain where surrogate use is already established.

1. **LCBench** (Zimmer et al., 2021) provides performance landscapes for hyperparameter tuning of simple multi-layer perceptrons on tabular datasets. It comprises 35 OpenML tasks (Vanschoren et al., 2014), each with 2000 configurations evaluated using 3 independent runs per configuration. Each configuration corresponds to a specific setting of seven hyperparameters. Notably, LCBench is included in the popular surrogate benchmark YAHPOGym (Pfisterer et al., 2022), and thus its inclusion into our study allows us to compare both our surrogate training process and the resulting scores on an established benchmark. For comparability with ARLBench, we randomly subsampled 256 configurations per task, calling this *LCBench 256*, while keeping the full dataset as *LCBench*.

2. **PD1** (Wang et al., 2024) is a dataset capturing the performance of CIFAR-100 (Krizhevsky, 2009) training runs. It includes four tasks, each with 500 configurations of 5 different hyperparameters evaluated using a single run. Overall, these can be considered harder ML tasks, allowing us to compare how RL surrogates perform compared to both simpler and harder ML tasks. We randomly subsampled 256 configurations for comparability to ARLBench.

For all datasets, performance values were normalised between 0 and 1 using appropriately selected bounds. For the supervised learning benchmarks, performance was measured in terms of accuracy, which is already normalised. For RL, wherever available and appropriate, our bounds corresponded to the theoretical maximum and minimum returns. Otherwise, competitive baseline values from the literature were utilised as reliable performance estimates for the given environment. In instances where the landscape data contained configurations that exceeded or fell below these established thresholds, the respective empirical landscape values were used for normalisation instead. The details for each environment can be found in Appendix A.

**Extended Evaluation Datasets**  To evaluate data scaling and establish high-fidelity ground truth for our interpretability analysis, we extended the ARLBench datasets for the Lunar Lander and Acrobot environments. We expanded both performance landscapes from 256 to 2048 configurations, evaluating each across 30 independent runs. For Lunar Lander, we utilised PPO, DQN, and SAC and for Acrobot we used PPO. These datasets, are referred to as *Lunar Lander 2048* (*LL 2048*) and *Acrobot 2048* (*AC 2048*).

## 4.2   Training Surrogate Models

To train surrogates for our datasets, we considered four model classes commonly used in this context: random forests (RFs, Breiman (2001), applied in (Segel et al., 2025; Hutter et al., 2014a)), Gaussian processes (GPs, Rasmussen & Williams (2006), employed in Lindauer et al. (2022); Falkner et al. (2018)), support vector machines (SVMs, Cortes & Vapnik (1995), utilised in (Eggensperger et al., 2015)) as implemented in Scikit-learn (Pedregosa et al., 2011), and XGBoost (XGB, Chen & Guestrin (2016), adopted in Zela et al. (2022)). These surrogates were trained to predict the average final performance, maximum performance and area under the (anytime) performance curve (AUC) of a given hyperparameter configuration across the independent runs in each dataset. We focus on final performance, given its prominence in the literature and because we observed qualitatively similar results under different performance metrics. Corresponding results for maximum performance and AUC are provided together with the extended results for each section in Appendix B.3, E, F, and we mention in the following where there are substantial qualitative differences.

All surrogate models were optimised via random search comprising 200 trials, with the exception of GPs, which were allocated 50 trials owing to their substantially smaller hyperparameter space. The surrogate model configuration spaces are analogous to those utilised in NAS-Bench-301 (Zela et al., 2022) and are detailed in Appendix B. Each surrogate model configuration was assessed using 10-fold cross-validation. For each fold, the data was partitioned into an 80% training set, a 10% validation set (utilised to identify the optimal

configuration from the random search), and a 10% test set. The optimal configuration was determined by the maximum mean performance across all 10 validation folds. Subsequently, the generalisation capability of this selected model was computed as its mean performance across the 10 test folds. All surrogate performance metrics reported in this paper reflect these test set evaluations. To obtain an uncertainty value over the optimisation of the surrogate models, the random search was repeated with 5 different pseudo-random seeds, each yielding one surrogate model.

To validate our implementation and training pipeline, we compared the performance of our surrogate models on LCBench to the surrogate models provided by YAHPOGym (see Appendix B.2). These initial experiments showed RFs to achieve the best predictive performance, and therefore, in the following, we focus on the RF results for the sake of brevity; however, a more detailed comparison of different performance metrics and surrogate models is provided in Appendix B.

### 4.3 Methods Surrogate-Based Insights

We now briefly outline the three surrogate-based analysis methods we use in our experiments.

**Local Parameter Importance** (LPI; Biedenkapp et al., 2017) quantifies hyperparameter importance within a configuration by the proportion of performance variance it explains. Unlike methods such as fANOVA Hutter et al. (2014a), this provides a *configuration-specific* assessment of which hyperparameters are most influential for a particular setup. More details on LPI can be found in Appendix E.

**Ablation Paths** (Fawcett & Hoos, 2016; Biedenkapp et al., 2017) quantify both the importance and interdependencies of hyperparameters by ordering changes from a source configuration to a target configuration based on the performance improvement induced by each change. This is done step-wise by identifying what hyperparameter results in the best performance improvement when changed from its source value to the target value and repeating the ablation with the hyperparameters that have not yet been considered. When repeated until all hyperparameters have been set to their target values, this results in a ranking of hyperparameter importance and their respective performance improvements.

**HyperShap** (Wever et al., 2026) formulates the analysis of hyperparameter landscapes as distinct cooperative games, for which Shapley values (Shapley, 1953) can quantify the additive contribution of individual hyperparameter settings to performance outcomes. In this work, we focus on the HyperSHAP ablation game to measure hyperparameter importance. Here, subsets of hyperparameters, or coalitions, are changed from a source configuration $\lambda^0$ to a target configuration $\lambda^*$. The performance difference between $\lambda^0$ and $\lambda^*$ for different coalitions lets us compute Shapley values, i.e. the marginal contribution of each hyperparameter and coalition to the performance of $\lambda^*$. To approximate global importance, we set $\lambda^*$ as our overall best configuration and averaged the results across 100 randomly sampled source configurations $\lambda^0$.

## 5 Performance Surrogates in Reinforcement Learning

The two most common uses for surrogate models in AutoML are performance prediction for benchmarking purposes and performance surrogates for analysis of target algorithm behaviour. Therefore, we compare how well surrogates for RL algorithm performance perform for these use cases. Our goal is to judge whether current surrogate training methods are suitable for RL and to what extent there is a need for further improvements.

### 5.1 Surrogates for Performance Prediction

Surrogate models with high accuracy in predicting target algorithm performance can be used as surrogate benchmarks instead of running the target algorithm, making evaluations very efficient (Eggensperger et al., 2015). Most important when using a surrogate benchmark is obviously a high degree of faithfulness to the ground truth when ranking configurations or predicting the performance of the best configurations. Therefore, following Bansal et al. (2022), we evaluated surrogate quality using Kendall's $\tau$ to assess the consistency of the surrogate prediction rankings. Kendall's $\tau$ admits a simple probabilistic interpretation with $(1 - \tau)/2$

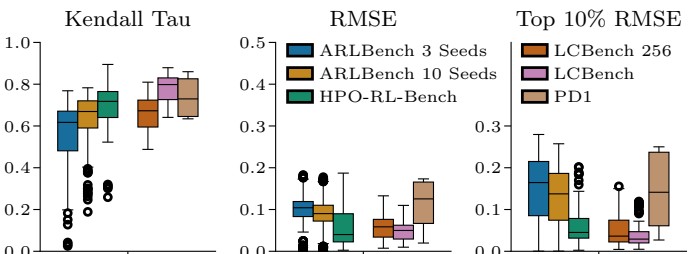

| Benchmark | Kendall Tau | RMSE | Top-10% |
|---|---|---|---|
| ARLBench 3 Seeds | 0.62 | 0.11 | 0.16 |
| ARLBench 10 Seeds | 0.67 | 0.09 | 0.14 |
| HPO-RL-Bench | 0.71 | 0.04 | 0.05 |
| LCBench 256 | 0.67 | 0.06 | 0.04 |
| LCBench 2000 | 0.80 | 0.05 | 0.03 |
| PD1 | 0.73 | 0.13 | 0.14 |

Figure 1: **Left:** Boxplots aggregating the test-set prediction performance of the RF surrogate models across all tasks, algorithms and surrogate model fits in each benchmark suite. The top-10% RMSE is the RMSE computed on only the 10% best-performing configurations in the landscape. The RL hyperparameter landscapes of ARLBench are subsampled to 3 training seeds in addition to the default 10 training seeds per configuration to be comparable to LCBench. **Right** For better visibility, the median values of the boxplots are depicted in a table.

being the probability that a randomly chosen pair of configurations is ranked incorrectly by the surrogate. Additionally, we evaluate RMSE across all configurations and the top 10%, to measure absolute accuracy of performance predictions (see Figure 1).

In Figure 1, we compare the RL data from ARLBench and HPO-RL-Bench to LCBench and PD1 to judge the raw results in context. Since LCBench uses only 3 training seeds, we give results for ARLBench subsampled to 3 seeds for a fair comparison in addition to the ARLBench results with all 10 training seeds. We observe that LCBench – which we use as our baseline, because it is well established and provides standardised preprocessing and evaluation protocols via YAHPOGym (Pfisterer et al., 2022) – achieves the highest overall Kendall's $\tau$ and a low RMSE that further improves when restricting the data to the best 10% of configurations in the dataset. Its smaller LCBench 256 variant keeps RMSE improvement, but its Kendall's $\tau$ worsens from about 0.8 to roughly 0.67. In terms of Kendall's $\tau$, the mean of our other datasets is similarly between 0.6 to 0.75. It is noticeable, however, that our RL datasets feature outliers with very low Kendall's $\tau$ (down to 0.3 for HPO-RL-Bench and even to below 0.1 for ARLBench). This is also true for the RMSE, where the mean for PD1 is worse than for the RL data, but both datasets give rise to outliers up to 0.2 for the full data and up to 0.3 in the case of ARLBench for the top-10%. We can clearly see that, especially PD1 and ARLBench, are very challenging in terms of RMSE when compared to LCBench, to a degree where the high top-10% RMSE renders surrogate benchmarks unreliable. ARLBench has at least 3× larger configuration spaces compared to HPO-RL-Bench across its algorithms, making HPO-RL-Bench an easier task for surrogate modelling, as can be seen by its substantially lower RMSE and higher Kendall's $\tau$. However, even for HPO-RL-Bench, we still see outliers for all metrics in low-performance regions, meaning that here as well, the performance may not be reliable enough to benchmark many algorithm-environment combinations.

To quantify the differences in Figure 1 further, we applied a Mann-Whitney U test comparing the combined RL datasets (ARLBench with 3 training seeds, HPO-RL-Bench) against the supervised learning datasets (LCBench 256, PD1). The test confirms a statistically significant degradation in surrogate performance for RL tasks across all metrics ($p < 0.001$). Computing the common language effect size (probability of superiority) from the U-statistic demonstrates the qualitative size of this gap: A supervised learning task yields a superior ranking correlation 60.7% of the time. Further, an RL task yields a higher overall RMSE 64.0% of the time and for top-10% RMSE 71.0% of the time.

> **Observation 1** Performance prediction for RL algorithms with full configuration spaces (i.e. ARLBench) is generally harder than for simple supervised learning tasks commonly found in surrogate benchmarks (e.g. LCBench). While on some tasks, similar performance can be achieved, RL shows more tasks with low prediction performance in both Kendall's $\tau$ and RMSE. Using substantially smaller configuration spaces (as in HPO-RL-Bench) or collecting more random seeds per configuration closes this gap.

What is potentially more concerning than the worse absolute scores of our RL surrogates compared to LCBench is the difference between RMSE and top-10% RMSE. In contrast to the situation for both LCBench and LCBench 256, the surrogate predictions do not improve when moving from the entire dataset to only the top 10% of configurations. HPO-RL-Bench shows similar RMSE on both sets, while the ARLBench surrogates look significantly worse, moving from just above 0.1 to about 0.15 with outliers up to 0.3 for top-10% RMSE. This is an especially important metric for benchmarking, as these are the interesting regions for HPO methods to explore.

> **Observation 2** Performance prediction quality for RL on the full algorithm configuration spaces is lower for better-performing configurations compared to simple supervised learning tasks (e.g. LCBench) and is instead comparable to more complex deep learning problems, e.g. PD1. Reduced configuration spaces as given in HPO-RL-Bench allow us to close this gap, whereas an increased number of seeds does not seem to narrow the gap.

## 5.2 Surrogates for Interpretability

Beyond surrogate benchmarking, an important use of performance prediction in AutoML is generating insights into the HPO landscape. Examples are surrogate-based hyperparameter importance (e.g. fANOVA; Hutter et al., 2014a) or the use of surrogates for the cheap evaluation of metrics that would require additional configuration evaluations (e.g. in the context of ablation analysis; Fawcett & Hoos, 2016). Even if accurate surrogate benchmarks for RL tasks appear to be out of reach for now, RL surrogates could still provide useful insights into HPO characteristics of these scenarios. To evaluate the quality of surrogate-based insights in RL, we chose methods that allow us to obtain insights using both surrogate models and ground-truth performance values: LPI (Biedenkapp et al., 2018) and ablation paths (Fawcett & Hoos, 2016).

To obtain the most accurate results, we retrained the five surrogate model configurations obtained in our random search on all the available configurations in our collected datasets for each landscape. Since both LPI and ablation paths subsequently utilise the resulting surrogate models on unseen configurations in their internal analysis, this does not compromise the integrity of our experiments.

**Local Hyperparameter Importance.** To evaluate the accuracy of surrogate models in estimating LPI, we computed ground-truth LPI values for three representative configurations of PPO and DQN on the extended LunarLander 2048 benchmark: the best-performing, 25th-percentile and median configuration. For computing the ground-truth LPIs, a maximum of 100 neighbour configurations were evaluated for the numerical hyperparameters and all possible values for categorical hyperparameters, similar to the DeepCave implementation (Sass et al., 2022). This amounts to 705 and 907 configurations per LPI analysis for PPO and DQN, respectively. Each configuration was trained with 30 seeds. We then compared these ground-truth LPIs to the corresponding estimates produced by surrogate models trained on subsets of the LunarLander 2048 dataset, aiming to assess the ability of the model to estimate configuration-specific hyperparameter importance. Further details can be found in Appendix E.

We measure the accuracy of LPI by its top-K consistency, which we define as the number of true top-K hyperparameters that are included in the predicted top-K set, regardless of their internal ranking. Figure 2 shows the consistency of surrogate-based LPI in predicting the most important hyperparameters. We observe that with a perfect top-1 consistency and 60% and 40% top-3 and top-5 consistencies, our surrogate models often correctly identify the most important hyperparameters. However, the absolute difference between ground truth and importance estimate can be quite large, even if the hyperparameter is correctly identified (e.g. up to a factor of 2 for each of the PPO configurations). Furthermore, on average, PPO and DQN have 4.3 and 8 hyperparameters with an importance above 5%, respectively, while the surrogate models estimate only around 3.3 and 2.6. Therefore, the surrogates cannot be relied upon for a full picture of hyperparameter importance, and their output should be seen as rough guidance rather than an accurate assessment. Similar to the results of Eimer et al. (2023), the specific hyperparameters that are most important vary substantially between algorithms and configurations. We present the most important hyperparameters for each algorithm and configuration in Appendix E.1.

**Ablation Paths.** Predicting ablation paths is a challenging task, due to the complexity of the ARLBench landscapes and the high errors observed in our surrogate models. To assess surrogate quality for ablation paths, we selected source configurations for LunarLander 2048 PPO and DQN across several performance percentiles (2nd, 13th, 25th, 38th, 50th, 63rd, 75th and 99th), using the overall best configuration as the target. We then computed the ground-truth ablation paths with 30 seeds per configuration and compared them to those predicted by our tuned surrogate models. We only include paths in our analysis where performance shifts by at least 10% in the ground truth. More details and raw results are provided in Appendix F.

We measure the accuracy of ablation paths by their top-K prediction accuracy, which we define as the accuracy of predicting a hyperparameter that is contained in the set of the ground truth top-K hyperparameters, causing the biggest performance shifts. Top-1 accuracy, where the surrogate identifies the exact hyperparameter responsible for the shift, is generally around 20% for 9 different hyperparameters of PPO, and between 30% and 40% for the 12 hyperparameters of DQN. Top-3 accuracy ranges from 60% for PPO and 80% for DQN, and top-5 accuracy is approximately 90% for both algorithms, supporting the finding in the context of LPI that strong trends can be gleaned from the surrogate, but details are not necessarily reflected accurately. Furthermore, the low top-1 accuracy score also means that the general shape of the path will be unreliable, since the order of hyperparameters will generally not be correct, making this analysis overall less faithful to the ground truth than LPI.

> **Observation 3** Surrogate-based insights about RL algorithms regarding hyperparameter importance and effects only match high-level trends in the ground truth data.

## 6 RL Hyperparameter Landscape Properties

Our results presented thus far show that the performance surrogates we trained based on the ARLBench and HPO-RL-Bench data are not meeting the level of quality required for reliable downstream analysis of the hyperparameter landscapes. Therefore, we explore what makes the performance landscape in our RL datasets hard to predict. It is known from work on supervised learning (Pushak & Hoos, 2022) that an easy-to-model performance landscape is relatively smooth, enabling accurate predictions on neighbouring configurations. The more we move away from these characteristics, the more effort is likely needed for data collection and model training. This is why we investigate the hyperparameter landscape smoothness and the influence of randomness within our datasets. Additionally, we investigate the performance distributions, that is the distributions of performance values across hyperparameter configurations, within each data set.

### 6.1 Performance Distributions

The performance distributions within each dataset show how much training data is available for different performance regions. If the distributions of performance values are skewed within a dataset and the data thus imbalanced, surrogate performance may be lower in low-data performance regions. Since each of our selected benchmarks consists of several different tasks of various difficulty levels, Figure 3 shows the performance distributions per task for each benchmark; analogous data for each individual task can be found in G. The normalisation of performance values is carried out as described in Section 4.1.

We observed that our RL datasets contain a higher performance spread per task than the supervised learning datasets. The average per-task spread in LCBench is around 50%, around 70% for PD1 and around 90% for ARLBench. HPO-RL-Bench shows an extremely high variability of performance spreads across tasks because many tasks gave rise to very few or no good configurations. This is an example of why such large spreads are not necessarily good: since our interest is in better performing regions, a lot of data in the bottom 50% of performance might not actually be useful for surrogate training.

This is clearly visible in the overall performance distributions, where LCBench shows a high density of values in the upper half of the performance spectrum while the other three datasets are predominantly characterised by low performance values. Most of the performances in PD1 are between 0.05 and 0.7, between 0.1 and 0.6 in HPO-RL-Bench and between 0.0 and 0.5 in ARLBench. The two RL datasets also contain less data in the

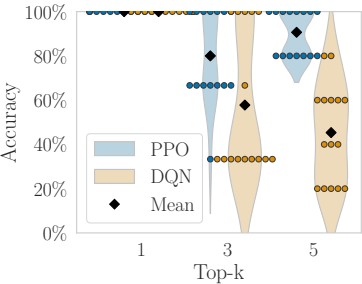

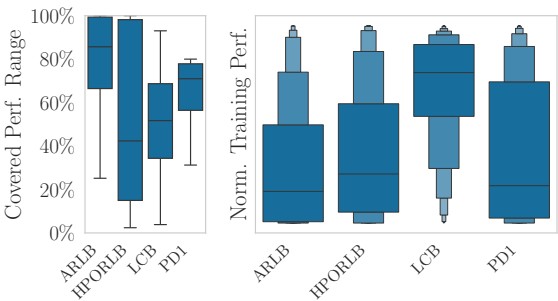

Figure 2: Accuracy of LPI predictions for the best-performing, 25th-percentile and median configuration for PPO and DQN obtained using the 5 RF surrogate fits. We report top-1, top-3 and top-5 consistency using the ground-truth values of the LPI.

Figure 3: **Left:** Relative coverage of the theoretical performance bounds per task in the empirical performance data. **Right:** Distributions of performance values in the datasets. ARLBench shows a higher concentration of low-performing configurations.

very well-performing regions above 0.8 than the supervised learning dataset LCBench. This is obviously an important factor for the top-10% RMSE metric, where we observed a different trend for LCBench than for the other datasets.

> **Challenge 1** The performance distributions of RL algorithms tend to be bottom-heavy, resulting in less data for the relevant high-performance regions. This is likely to cause challenges for training surrogates that are accurate where it matters most, e.g., in the context of hyperparameter optimisation.

## 6.2 Landscape Unimodality

The shape of RL hyperparameter landscapes could also be a factor in why they are hard to model via surrogates. Features like local optima or multiple high-performance regions can make learning the landscape shape more difficult. We follow the approach of Pushak & Hoos (2018) to investigate the unimodality of RL hyperparameter landscapes. While continuous optimisation often relies on the analysis of convexity, hyperparameter landscapes typically mix continuous, discrete and categorical variables. Therefore, we assess the smoothness of the hyperparameter landscape using the concept of unimodality, defined here by the presence of monotonically improving paths to the global optimum.

To this end, we count all configurations that can reach the global optimum via step-wise improving paths, defined as sequences of neighbouring configurations along which performance does not decrease, starting from a given configuration and ending at the global optimum. The fraction of configurations that can reach the global optimum relative to the total number of configurations is called the *reachability ratio*. This ratio estimates the proportion of trajectories in the performance landscape that can be followed without becoming trapped in local optima (see Figure 4). In most of our datasets, configurations are obtained via random search, so conventional neighbourhood relations are typically unavailable. Therefore, we define neighbourhood relations using a $k$-nearest neighbour approach with $k = 10$. Appendix C compares results across different values of $k$, showing that our conclusions are robust to this choice. In Pushak & Hoos (2018), step-wise improving paths were computed within the confidence intervals of each point to account for variability between runs. Since the width of these confidence intervals varies widely across benchmarks, we instead plot the reachability ratio using standardised relative performance intervals around each point. This is, for a specific configuration $c$ with mean performance $p_c$, the upper and lower bounds of the performance interval are defined as $p_c \pm \epsilon$ with $\epsilon \in [0, 1]$. For illustration, we consider relative intervals $\epsilon \in [0, 0.01, 0.02, 0.05, 0.1]$, this is, between 0% and 10% (see Figure 4).

We observe that for all performance intervals, including only traversing the mean at 0%, the RL benchmarks show median reachabilities considerably below the supervised ones. Above $\pm 2\%$, the supervised landscapes

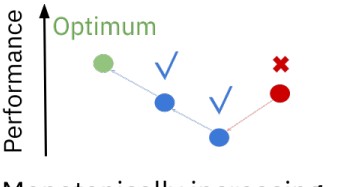

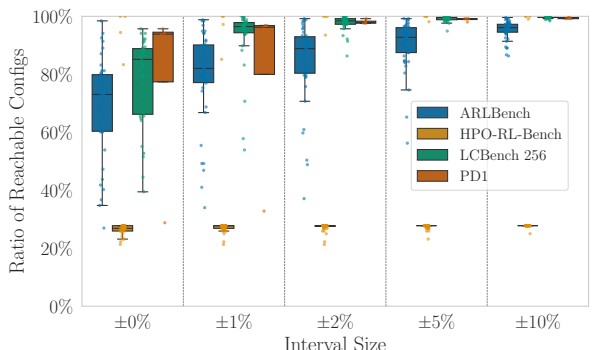

Figure 4: **Left:** One-dimensional illustration of our reachability analysis. **Right:** The percentage of reachable configurations via monotonically decreasing paths from the best-performing point within performance intervals. For intervals $> 0$, supervised learning landscapes show significantly greater unimodality.

appear to be almost fully unimodal; however, for the RL benchmarks, a fraction of configurations remains unreachable even at interval size $\pm 10\%$. This is in stark contrast to the results of Pushak & Hoos (2022), who observed near-perfect unimodality in the performance landscapes of the models they examined, including simple feed-forward neural network training, XGBoost, SVMs, and latent Dirichlet allocation (Hoffman et al., 2010). From an optimisation perspective, such multimodal landscapes pose challenges, as there is a risk of becoming trapped in local optima, and gradients tend to be less informative of the global optimum. In Appendix I, we present additional multidimensional scaling plots of the ARLBench and LCBench hyperparameter landscapes. These landscapes further support our conclusions. While LCBench shows distinguishable high- and low-performing regions, in ARLBench, both high-performing and low-performing configurations appear to be spread out substantially more uniformly across the search space.

> **Challenge 2** RL hyperparameter landscapes are rougher, i.e., contain more local optima, than supervised learning ones, rendering them more challenging for optimisation.

### 6.3 The Influence of Randomness

Large variability between different runs of a given configuration of an RL algorithm can pose serious challenges to predicting performance accurately. To better understand the uncertainty in the mean performance $p_c$ of a configuration $c$, we examine the size of the 95% bootstrapped performance confidence intervals (CIs), defined as $|\hat{p}_c - \check{p}_c|$ with $\hat{p}_c$ being the upper confidence bound and $\check{p}_c$ the lower confidence bound. Looking at the size of the intervals for the configuration included in ARLBench, HPO-RL-Bench and LCBench (see Figure 5), we noticed a strong tendency towards substantially larger CIs for the RL datasets and high variation in CI size. While LCBench shows about 90% of CIs smaller than 10% with a median of 0.4%, the CIs for HPO-RL-Bench and ARLBench extend up to 44% and 57% for three seeds, respectively, and show much higher medians (3% and 5%). This means there can be significant differences between runs of the same configuration in HPO-RL-Bench and especially ARLBench. It also makes dataset collection harder, since we expect to require a larger number of runs per configuration for accurate performance measurement and prediction.

Examining the distribution of CI size per ARLBench environment, as shown in Figure 5, we find that ARLBench also shows a more diverse and complex behaviour of the CIs with increasing performance values compared to LCBench. Some environments show a smooth progression of low CIs to high CIs with medium performance and back to low CIs for very well-performing configurations again. This, unfortunately, is not always the case; several environments show less benign relationships between CI size and performance, and there are even cases with a linear increase in CI as performance improves. Particularly, the latter case presents a challenge for training surrogates, as it makes predictions even more challenging in the well-performing regions that are only weakly covered by the HPO-RL-Bench and ARLBench datasets.

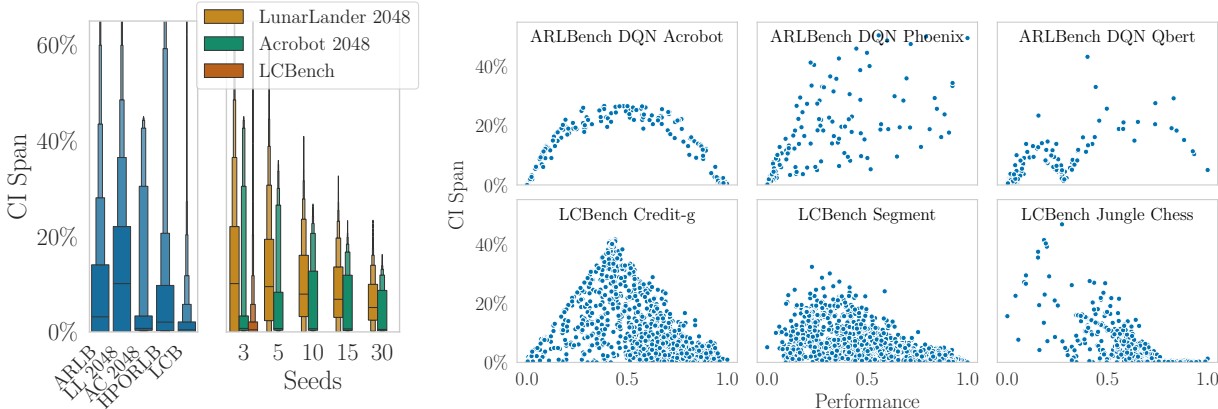

Figure 5: **Left:** Boxenplots aggregate the span of bootstrapped 95% performance CIs (i.e. upper performance confidence minus lower performance confidence) over all tasks and configurations for each benchmark suite. Further, for the Lunar Lander 2048 dataset, the box plots aggregate the CI span for each configuration trained with different seeds. **Right:** Whereas in LCBench, high-performing configurations tend to have smaller CI spans with better performance, for ARLBench, we observe tasks with benign behaviour, larger CI spans for higher performance or varying CI span sizes for different performance regions. Unlike ARLBench (top), LCBench (bottom) consistently exhibits very small CI spans at high performance levels.

If we could model the variability between runs, however, we might be able to considerably improve predictions. Therefore, we tested whether run-to-run variation is heteroscedastic, i.e., if noise differs across configurations. We assessed heteroscedasticity using Levene's test, which evaluates the null hypothesis that the variances across configurations are equal (i.e., the data is homoscedastic). The test was first applied across all configurations within each environment. To mitigate potential bias from extreme values, we repeated the test, considering only those configurations whose performance values did not coincide with the minimum or maximum observed performance for all their seeds in an RL environment. A standard significance level of 5% was used to determine statistical significance. Homoscedasticity was rejected in nearly all cases, across both testing setups, with p-values generally below $10^{-8}$, indicating strong evidence of unequal variance. The only exception was observed for the Mountain Car environment with PPO, within the reduced 256 configurations set, where the test fails to reject homoscedasticity. The likely reason for this is that only two of the 256 configurations do not lie on the performance boundary, and thus, the test was performed on only two configurations; when including all configurations, homoscedasticity was rejected as well. This result shows that we cannot expect to easily model the variations between seeds with surrogate models that can take randomness into account (such as BNNs (Goan & Fookes, 2020)), but would have to model the variation across independent runs on a per-configuration basis – a far more difficult modelling task.

> **Challenge 3** Performance variation between multiple runs of the same configuration of a given RL algorithm can be substantial, i.e. having large CIs, and behaves heteroscedastic, making it difficult to model performance variation.

## 7 Improving Surrogate Quality

After pinpointing specific challenges with the existing RL performance datasets, we decided to improve the training conditions for RL surrogates by extending the dataset in size, improving data quality and restricting the configuration space we consider.

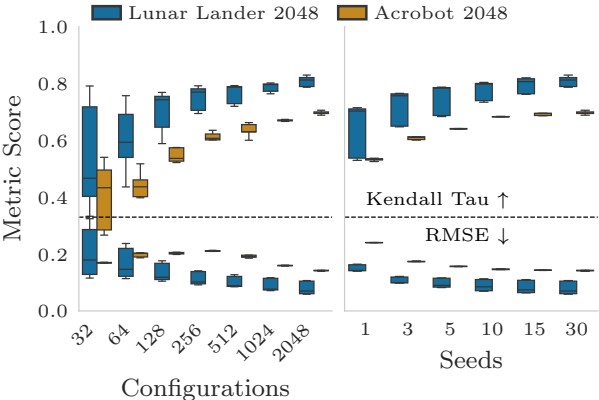 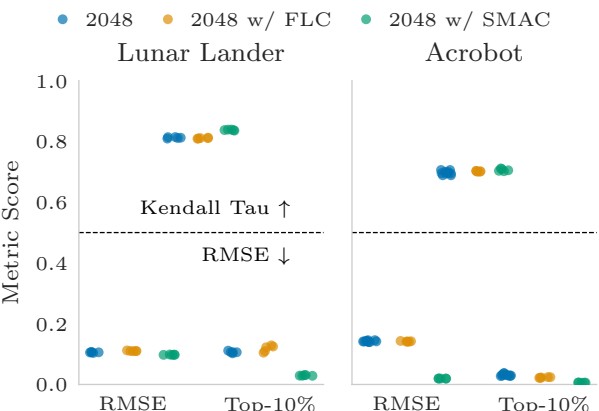

Figure 6: Box plots aggregating the RF surrogate model performances for the three algorithms and 5 model fits in the Lunar Lander and Acrobot 2048 dataset. **Left:** Test-set quality gains of surrogates with increasing number of configurations (30 seeds per configuration). **Right:** Test-set quality gains with increasing numbers of seeds (for 2048 configurations).

Figure 7: Comparison of RF surrogate test-set quality on Lunar Lander and Acrobot 2048 for PPO and all 5 model fits when training on the full learning curves (FLC) and with additional 256 configurations collected by a SMAC optimisation warmstarted with the random search landscapes.

### 7.1 Intervention 1: More Data

Given that we observed an improvement in surrogate metrics between LCBench 256 and LCBench and ARLBench subsampled to 3 seeds and ARLBench with 10 seeds, we used the extended datasets introduced in Section 4.1 to further study surrogate performance with increased numbers of configurations and seeds. Since we wanted to gather significantly more data to see the effects of data scaling, and this comes at substantial computational cost, we focus on the Lunar Lander and Acrobot environments in ARLBench for these experiments. We chose Lunar Lander as it showed the best surrogate scores with respect to the PPO algorithm out of all environments (PPO is the only algorithm with data from all environments), and we can collect data for all three algorithms, PPO, DQN and SAC. Additionally, we chose Acrobot trained with PPO, as it is among the worst environments in its Kendall's $\tau$ value ($\sim 0.4$) and also among the environments with the highest RMSE ($\sim 0.2$).

Figure 6 shows how surrogate quality develops with increasing numbers of configurations and seeds. More configurations lead to a higher Kendall's $\tau$ and more stable predictions, though the overall improvement from 256 configurations to 2048 configurations remains fairly limited. The biggest improvements already happen between 32 and 256 configurations; further data seems to yield diminishing returns. Increasing the number of configurations shows almost no overall improvement in the RMSE for Acrobot and even degrades performance between 32 and 256 configurations.

As with an increasing number of configurations, we observed a steady but decreasing improvement when using more seeds, but again, the difference between the 10 ARLBench seeds and our maximum of 30 is small. We therefore assume that the data requirements for well-performing surrogates, at least if they are generated by random sampling, are far greater than our $3\times$ extension of a dataset already containing a substantial number of seeds for RL tasks.

Repeating our LPI analysis with these surrogates trained on the Lunar Lander 2048 dataset, we observed that, interestingly, increasing the number of configurations for training the surrogates does not substantially improve accuracy for DQN and only improves top-3 and top-5 accuracy for PPO. For the ablation paths, increasing the number of configurations does not lead to consistent improvements and sometimes results in worse performance, which is counterintuitive. We used HyperShap to analyse what hyperparameters are

important in these surrogates and confirm that surrogate models trained on different numbers of seeds and configurations base their predictions on very similar hyperparameters and interactions (see Appendix D.1).

These results suggest that, while more data helps improve surrogate quality, improvements beyond the existing ARLBench dataset size are small and do not substantially improve the accuracy of hyperparameter analysis methods. Indeed, it seems that an increased amount of randomly sampled data, from 256 to 2048 configurations, is not necessarily worth gathering, even for a well-behaved environment such as Lunar Lander.

### 7.2 Intervention 2: Better Data

An alternative to simply gathering more data is to collect better data. Of course, it is not always clear what better data looks like, but in the case of HPO-RL-Bench and ARLBench, we know the data distribution is heavily skewed towards poorly performing configurations. This led us to believe that a simple improvement would be to gather data in the high-performance regions of the configuration space. Therefore, we extended our 2048 configuration datasets with an additional 256 configurations generated using the SMAC general-purpose algorithm configurator (Lindauer et al., 2022), following the recommendations of Eggensperger et al. (2015). SMAC is a dedicated hyperparameter optimisation algorithm that employs Bayesian optimisation to sequentially select promising configurations based on prior evaluations. We further investigated surrogate quality with higher training data density by training on full learning curves (FLC) of 10 intermediate evaluations, as commonly done in the literature (Pfisterer et al., 2022). This means that the surrogate model gets the number of training steps as an additional input to the hyperparameter configuration, normalised between 0 and 1, i.e. it extends the surrogate model to $\hat{f}_{\mathcal{A},\mathcal{M}} : \Lambda \times [0, 1] \rightarrow \mathbb{R}$, with $[0, 1]$ as the domain of the normalised training step.

As shown in Figure 7, training on full learning curves barely changes the surrogate metrics; it thus appears that the learning curves do not add relevant information in this case. For Lunar Lander, the SMAC data, however, improves the top 10% RMSE and slightly increases Kendall's $\tau$, while the overall RMSE remains unchanged. For Acrobot, the SMAC data improves both RMSE and top 10% RMSE while Kendall's $\tau$ remains unchanged. This suggests that the data sparsity at the high-performing end partly contributes to the degradation in surrogate accuracy for top configurations and adding high-performing data can help, particularly with obtaining improved RMSE and top 10% RMSE performance.

Furthermore, the additional SMAC data does not enhance LPI accuracy and does not substantially improve the correctness of the ablation paths for Lunar Lander. These findings suggest that the lack of data for high-performance configurations does not fully explain the observed limitations of the surrogates.

### 7.3 Intervention 3: Reduced Configuration Spaces

The dimensionality of the configuration space is obviously another factor causing challenges in training surrogate models. This can be seen in our HPO-RL-Bench results, which are based on similar algorithms and environments as those in ARLBench, indicating a hyperparameter landscape of similar complexity; yet, on the smaller configuration space of HPO-RL-Bench, we obtained much better surrogate scores.

We therefore manually reduced the configuration space of Lunar Lander and Acrobot for PPO and compared surrogate quality on this "expert-reduced" space induced by only three hyperparameters to the original space comprising nine hyperparameters. The hyperparameters not contained in the smaller configuration spaces are set to the respective environment baseline values as specified in ARLBench. We chose the learning rate, GAE $\lambda$ and clipping $\epsilon$, as these have been identified as important in the literature ((Eimer et al., 2021; Ceron et al., 2024) and clearly influence model updates, credit assignment and exploration. For Lunar Lander, repeating our surrogate training procedure on randomly sampled configurations from this configuration space, we obtained significant improvements with respect to all our metrics compared to the full Lunar Lander configuration space (see Figure 8, left). While Kendall's $\tau$ is similar to that of the full search space with 32 configurations, the reduced version shows better RMSE and, most importantly, improved scaling with the number of configurations. This indicates that there is an easier-to-model subspace of the full configuration space that still captures important performance information. These scores are better than those on the HPO-RL-Bench data and are comparable to LCBench, indicating that we have achieved our goal of training

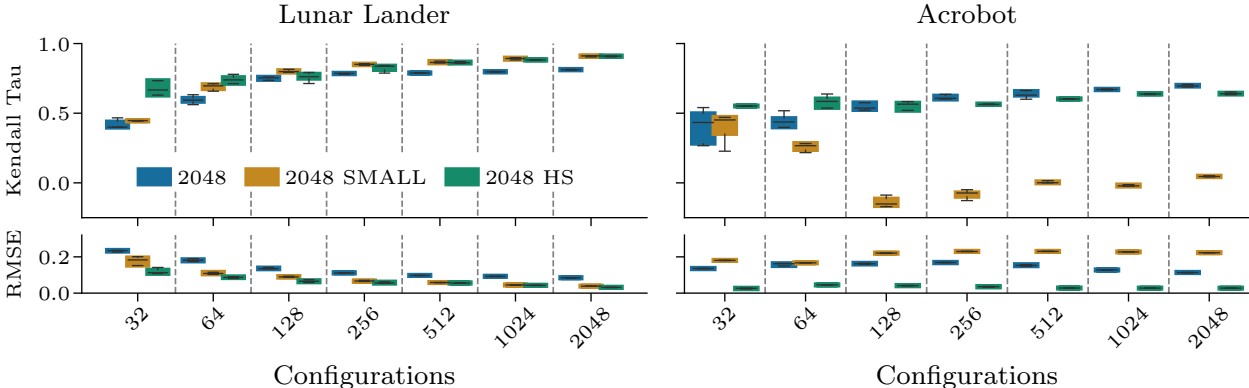

Figure 8: Boxplots aggregate the test-set performance of RF surrogate models across the 5 fits trained with growing numbers of configurations on reduced configuration spaces: expert reduced "SMALL" and HyperShap reduced "HS". For Lunar Lander, results are shown for all three algorithms and for Acrobot only for PPO.

a surrogate model fit for benchmarking. However, for Acrobot, we observe very different behaviour (see Figure 8, right). The manual search space reduction results in a higher RMSE and substantially decreased Kendall's $\tau$. This shows that manual configuration space reduction can also exacerbate the difficulty in landscape modelling if not done carefully.

We therefore propose an alternative for reducing configuration spaces for surrogate models. Specifically, we used HyperShap (Wever et al., 2026) to identify the most important effects in the predictions of a surrogate. Focusing on the hyperparameters with the highest importance scores will ensure that the resulting configuration space includes the most important effects *that can be captured by the surrogate* while excluding both unimportant hyperparameters and hyperparameters that the surrogate fails to model well.

Using our Lunar Lander and Acrobot surrogate trained on 2048 configurations and 30 seeds, the three most important hyperparameters for both environments are the GAE $\lambda$, the entropy coefficient and normalisation of the advantage estimation. In fact, these three and their interactions make up all top-5 hyperparameter effects. These top 5 are approximately consistent both across different subsamples of the 2048 configurations (see Appendix D.1) as well as for different numbers of configurations and seeds (see Appendix D.2 and D.3). In these two domains, therefore, HyperShap allows us to identify a fairly stable core of hyperparameters relevant for the construction of surrogates. This automatically reduced configuration space is easier to model than the full dataset and achieves much better results with only 32 and 64 configurations (see Figure 8). While for Acrobot, the Kendall's $\tau$ is slightly lower for more than 256 configurations, it consistently achieves a substantial improvement in RMSE. Therefore, our HyperSHAP-based configuration space reduction is an easy-to-apply alternative to manual configuration space reduction that, in our use case, yielded better results.

However, configuration space reduction is no silver bullet. Effects that the surrogate cannot fit well are likely also important for training performance; therefore, we may be missing some of the most relevant hyperparameter configurations. When analysing the properties of the reduced landscapes regarding reachability and noise, we observe reduced noise but lower reachability, which indicates no overall improvement for the landscape; detailed results can be found in Appendix D.4.

Furthermore, environments likely scale differently in the number of important hyperparameters: some may need only two hyperparameters, others up to five, as observed by Eimer et al. (2023). Lastly, for Lunar Lander and Acrobot, the most important subsets of hyperparameters are very stable across all our surrogates. Even 64 configurations would be sufficient to yield the same reduced configuration space as 2048 configurations. This does not necessarily hold for more difficult environments, where a surrogate trained on substantially more than 64 configurations might be required for obtaining a well-performing reduced configuration space.

The use of reduced configuration spaces for obtaining insights on hyperparameter importance and interactions is obviously limited, since we exclude potentially very important hyperparameters. Nevertheless, our LPI analysis for the reduced hyperparameter spaces yields overall much better fits to the ground truth, with RMSEs comparable to the full 2048 model, even with 32 configurations and high top-1 and top-2 accuracy (see Appendix E.2). For the ablation paths, the reduced configuration spaces also improve the accuracy with respect to the ground truth.

Overall, these results validate our configuration space reduction method, but cannot replace a full-scale analysis. Therefore, while automatic configuration space reduction appears to be a promising approach, surrogate-based hyperparameter analysis currently does not seem practical for RL algorithms.

## 8    Surrogates in RL: Practical Recommendations

Based on our findings, we give seven concrete recommendations for surrogate modelling techniques in RL.

**Surrogate Benchmarks:**    Given our results, sufficiently accurate surrogate benchmarks in RL are possible, but full configuration spaces will require immense amounts of performance data. We therefore recommend a workflow for benchmark construction with reduced configuration spaces.

---

**Surrogate Benchmarking**

1. **Baseline Surrogate**: Fit a surrogate used for configuration space reduction. This can be done using existing datasets, e.g. the ARLBench data. An alternative is gathering an initial random search dataset covering the search space of potentially relevant hyperparameters.

2. **Configuration Space Reduction**: Use HyperShap to reduce the number of hyperparameters in the configuration space. As the size of the ideal reduced space will likely vary, we recommend finding a good balance between covering as many subsets of important hyperparameters as possible while keeping the absolute number of hyperparameters low.

3. **Data Collection**: Collect the training data for the surrogate based on the reduced configuration space. We recommend using both random sampling and optimiser data for improved coverage of high-performance regions.

4. **Surrogate Training**: Use the collected data to fit and tune a surrogate. Note that both model selection and HPO of the surrogates resulted in significant predictive quality gains (see Appendix B). We therefore recommend surrogate tuning to be included in any training pipeline.

---

**Surrogate-based Insights:**    Our results on using surrogates to gain insights on RL algorithms warrant caution overall. Therefore, our biggest recommendation is *to carefully assess surrogate performance for individual tasks* and additionally *to limit analysis on high-level trends in hyperparameter importance and impact* using the surrogate model. We further detail this below.

---

**Surrogate-Based Interpretability**

5. **Rough trends hold** generally, even though surrogate-based analysis may not accurately reflect the ground truth. Insights such as identification of the three most important hyperparameters may be approximately correct, but it is not advisable to trust exact scores and rankings of less important hyperparameters. Therefore, analysis methods like ablation paths that compound errors should only be used if the surrogate model is evidently reliable, which was not the case in our RL landscapes, even for Lunar Lander with PPO, a landscape with benign behaviour.

6. **Question analysis tools** on their surrogate use. Hyperparameter insights are offered as part of several tools, such as Weights & Biases (Biewald, 2020), Optuna (Akiba et al., 2019) or DeepCave (Segel et al., 2025), although we do not know of any tools that make their surrogates

---

accessible (e.g. for tuning) or even show the surrogate fit. While this is convenient, it may also hide potential reliability issues.

7. **Run evaluations** where certainty is needed, even if it is more computationally expensive.

## 9 Ugly Landscapes and Bad Quality?

Indeed, our analysis shows that surrogate modelling may be very difficult for RL in general and possibly also for more complex deep learning problems. The optimisation landscapes we have characterised are not unimodal, RL tasks induce large and heteroscedastic noise patterns we cannot consistently model, surrogates have large prediction errors even with tuning and additional data collection, and surrogate-based HPO insights are unreliable, even when trained on large amounts of performance data. In short, the training of useful surrogate models for RL is rather surprisingly more challenging than for supervised learning, and their use for AutoML purposes therefore requires substantial adjustments.

Nonetheless, we do not believe that performance prediction in RL is completely infeasible; in fact, we have seen that reducing the number of hyperparameters in the configuration space results in surrogate models suitable for benchmarking, and we have proposed an automated manner for finding such reduced configuration spaces. However, for the moment, we advise against using performance prediction in the context of RL when precision is required. Even for environments with seemingly low error rates within the ARLBench tasks, such as Lunar Lander, surrogate predictions are often faulty and will not reliably match characteristics of the underlying performance landscape. Therefore, HPO insights should be gathered with additional function evaluations instead of fully surrogate-based methods if the details of the result are important (e.g. the ranking of important hyperparameters should be correct). Where rough impressions or insights on reduced search spaces are sufficient, surrogates remain a useful tool.

The current paradigm appears unlikely to yield comprehensive insights into hyperparameter landscapes and configuration spaces given realistic training data volumes. Since optimal RL configurations can shift significantly during training (Mohan et al., 2023), adapting learning curve prediction methods, which utilise partial evaluations to forecast final performance (Adriaensen et al., 2024), may mitigate this issue. Furthermore, addressing the extreme variance between independent runs through explicit probabilistic modelling could substantially enhance surrogate model quality. It seems, therefore, that RL is an ideal testing ground for new iterations of surrogate modelling techniques that take variance and algorithm dynamics into account.

The question of what these findings imply about surrogate-based HPO methods remains open. Some previous work has found that BO performs similarly to random search on RL environments (Becktepe et al., 2024), but evidence from the literature for this remains inconsistent (Shala et al., 2024). On tasks comparable to the PD1 data, BO has also not performed well for learning rate optimisation, with significantly worse results than on simpler tasks with a much more uniform landscape (Henheik et al., 2025). While the relationship between the hard-to-fit surrogate models and at times suboptimal HPO is, as of yet, not fully clear and requires further research, the information we gathered about RL hyperparameter landscapes should nevertheless be useful in designing improved HPO strategies that better capture these traits.

Therefore, we do not see our results as purely negative; they actually point towards various potential causes for the difficulties in tuning RL algorithms experienced by the RL community. Quantifying the underlying issues in the HPO landscape points towards what better surrogate models would need to capture, alternative data modalities (e.g. dynamic configurations or learning curves) for fitting them, possible improvements for optimizers to still arrive at very good results, and even at how RL algorithms themselves can be improved for more predictable performance. While currently most studies of RL algorithms compare performance over time only, our results suggest that higher unimodality in the performance landscape, better performance distributions and more consistency between runs of good configurations can serve as impactful metrics for assessing progress in the development of better RL methods.

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

# Appendix

## Table of Contents

# A  Datasets

All datasets used to train the surrogate models were generated via random search within the respective configuration spaces of each dataset. To ensure comparability across datasets, all performance values were normalised to the range $[0, 1]$.

The data processing procedure for each dataset is as follows:

1. All columns that do not represent hyperparameters, test performance, or the seed are removed.

2. All *NaN* values are imputed with the minimum observed performance value.

3. Each performance value is normalised to $[0, 1]$ using appropriate performance bounds.

4. For each configuration and seed, we compute the following metrics from the normalised training performance learning curve: final performance, maximum performance, and the AUC.

5. These performance metrics are then averaged across training runs for each configuration. Additionally, 95% confidence intervals are computed to quantify the uncertainty associated with each average.

The following sections provide further details on the characteristics of the landscapes represented in our datasets, their configuration spaces, and their respective normalisation procedures.

## A.1  ARLBench

*ARLBench* (Becktepe et al., 2024) is a benchmark for comparing the performance of HPO methods for RL algorithms. It contains 21 different environments depicted in Table 4. The set of environments consists of 5 ALE environments (Bellemare et al., 2013; Machado et al., 2018), 3 Box2D and 5 classic control environments from Gymnasium (Towers et al., 2024), 4 continuous control environments from Google Brax (Freeman et al., 2021) and 4 XLand-MiniGrid environments (Nikulin et al., 2024). ARLBench comes with implementations of PPO (Schulman et al., 2017), DQN (Mnih et al., 2013) and SAC (Haarnoja et al., 2018). As part of the benchmark, the authors have published a dataset consisting of performance landscapes of each algorithm and environment combination, with DQN and SAC being applied only to the respective environments with discrete and continuous action spaces. Each landscape consists of 256 configurations that have been randomly sampled in configuration spaces specific to each combination. Each configuration is evaluated using 10 independent runs. The configuration spaces of PPO, DQN and SAC for random sampling used in ARLBench are given in Table 1, 2 and 3, respectively.

| Hyperparameter | Box2D | XLand | ALE | CC | Brax |
|---|---|---|---|---|---|
| batch size | $\{32, 64, 128\}$ | | $\{128, 256, 512\}$ | | $\{512, 1024, 2048\}$ |
| learning rate | $\log([10^{-6}, 10^{-1}])$ | | | | |
| entropy coefficient | $[0.0, 0.5]$ | | | | |
| gae lambda | $[0.8, 0.9999]$ | | | | |
| policy clipping | $[0.0, 0.5]$ | | | | |
| value clipping | $[0.0, 0.5]$ | | | | |
| normalize advantages | $\{Yes, No\}$ | | | | |
| value function coefficient | $[0.0, 1.0]$ | | | | |
| max gradient norm | $[0.0, 1.0]$ | | | | |

Table 1: The hyperparameter configuration space of ARLBench for PPO. The first column specifies the different classes of tasks in the benchmark that use different configuration spaces.

| Hyperparameter | ALE | Box2D | CC | XLand |
|---|---|---|---|---|
| batch size | $\{16, 32, 64\}$ | $\{64, 128, 256\}$ | | $\{32, 64, 128\}$ |
| buffer priority sampling | | $\{Yes, No\}$ | | |
| buffer $\alpha$ | | $[0.01, 1.0]$ | | |
| buffer $\beta$ | | $[0.01, 1.0]$ | | |
| buffer $\epsilon$ | | $\log([10^{-7}, 10^{-3}])$ | | |
| buffer size | | $[1024, 10^6]$ | | |
| initial epsilon | | $[0.5, 1.0]$ | | |
| target epsilon | | $[0.001, 0.2]$ | | |
| learning rate | | $\log([10^{-6}, 10^{-1}])$ | | |
| learning starts | | $[1, 2048]$ | | |
| use target network | | $\{Yes, No\}$ | | |
| target update interval | | $[1, 2000]$ | | |

Table 2: The hyperparameter configuration space of ARLBench for DQN. The target update interval is a conditional hyperparameter that is only optimised when a target network is used. Similarly, $\alpha$, $\beta$ and $\epsilon$ of the buffer are only optimised when priority sampling is used. The first column specifies the different classes of tasks in the benchmark that use different configuration spaces.

| Hyperparameter | Box2D | CC | Brax |
|---|---|---|---|
| batch size | $\{128, 256, 512\}$ | $\{256, 512, 1024\}$ | $\{512, 1024, 2048\}$ |
| buffer priority sampling | | $\{Yes, No\}$ | |
| buffer $\alpha$ | | $[0.01, 1.0]$ | |
| buffer $\beta$ | | $[0.01, 1.0]$ | |
| buffer $\epsilon$ | | $\log([10^{-7}, 10^{-3}])$ | |
| buffer size | | $[1024, 10^6]$ | |
| learning rate | | $\log([10^{-6}, 10^{-1}])$ | |
| learning starts | | $[1, 2048]$ | |
| use target network | | $\{Yes, No\}$ | |
| tau | | $[0.01, 1.0]$ | |
| reward scale | | $\log([0.1, 10])$ | |

Table 3: The hyperparameter configuration space of ARLBench for SAC. The hyperparameter configuration space for SAC. The hyperparameter $\tau$ is a conditional parameter that is only optimised when a target network is used. Similarly, $\alpha$, $\beta$ and $\epsilon$ of the buffer are only optimised when priority sampling is used. The first column specifies the different classes of tasks in the benchmark that use different configuration spaces.

The performances of different RL environments are not straightforward to normalise due to differences in the scale of their returns across environments. Furthermore, often, no clear minimal or maximal performance values are available. All environments with clearly defined minimal and maximal performances have been normalised as such. For environments without clearly defined performance boundaries, we followed an approach similar to the human-normalised scores of the ALE (Mnih et al., 2015), where we used the performance of a random policy as the minimal value and a high-performing baseline from the literature as the maximal value. For environments from the ALE, we used the random and human scores presented by Mnih et al. (2015). For the maximum scores in Box2D Lunar Lander and Brax Ant and Humanoid, we used the best results from Dierkes et al. (2024), while for Brax Hopper and HalfCheetah, we relied on the top performances reported by Zheng et al. (2025). The lower values of $-200$ and $-2000$ for the Box2D and Brax environments have been chosen similarly to ARLBench due to reported numerical instabilities. If the performance value in a landscape was greater or smaller than the normalisation performance, the respective landscape maximal performance was used for normalisation.

| Environment | Performance | |
|---|---|---|
| | Minimal | Maximal |
| ALE BattleZone-v5 | 2360 | 37187 |
| ALE DoubleDunk-v5 | -18.6 | -16.4 |
| ALE NameThisGame-v5 | 2292 | 8049 |
| ALE Phoenix-v5 | 761 | 7242 |
| ALE Qbert-v5 | 163 | 13455 |
| Box2D BipedalWalker-v3 | -200 | 300 |
| Box2D LunarLander-v2 | -200 | 287 |
| Box2D LunarLanderCont-v2 | -200 | 287 |
| Brax Ant | -2000 | 8379 |
| Brax Halfcheetah | -2000 | 15000 |
| Brax Hopper | -2000 | 3300 |
| Brax Humanoid | -2000 | 12292 |
| CC Acrobot-v1 | -1000 | -100 |
| CC CartPole-v1 | 0 | 500 |
| CC MountainCar-v0 | -200 | -110 |
| CC MountainCarCont-v0 | -200 | -110 |
| CC Pendulum-v1 | -1300 | -200 |
| MiniGrid DoorKey-5x5 | 0 | 1 |
| MiniGrid EmptyRandom-5x5 | 0 | 1 |
| MiniGrid FourRooms | 0 | 1 |
| MiniGrid Unlock | 0 | 1 |

| Environment (ALE) | Performance | |
|---|---|---|
| | Minimal | Maximal |
| Alien-v0 | 227.8 | 6875 |
| Asteroids-v0 | 719.1 | 13157 |
| BankHeist-v0 | 14.2 | 753.1 |
| BeamRider-v0 | 363.9 | 16926.5 |
| Bowling-v0 | 23.1 | 160.7 |
| Boxing-v0 | 0.1 | 12.1 |
| Breakout-v0 | 1.7 | 4.3 |
| Enduro-v0 | 0 | 309.6 |
| Pong-v0 | -20.7 | 9.3 |
| Riverraid-v0 | 1339 | 13513 |
| Seaquest-v0 | 68.4 | 20182 |
| Skiing-v0 | -17098.1 | -4336.9 |
| SpaceInvaders-v0 | 148 | 1668.7 |
| Tennis-v0 | -23.8 | -8.3 |

Table 4: Minimal and maximal performance values used for normalising the RL performance landscapes. If the performance value in a landscape was greater or smaller than the normalisation performance, the minimal or maximal performance value of the landscape was used for normalisation. **Left:** The normalisation values for ARLBench and Lunar Lander 2048. **Right:** The normalisation values for HPO-RL-Bench if not already included in ARLBench.

## A.2 LunarLander 2048

The landscapes in ARLBench consist of 256 configurations, each with 10 individual training runs. While this is already a large number of configurations compared to previous works like HPO-RL-Bench (Shala et al., 2024) with 108 configurations, it does not yet allow for studying the effects of the number of configurations and random seeds at scale. Therefore, we used ARLBench to compute new landscapes for Lunar Lander using PPO, DQN and SAC, with each landscape consisting of 2048 different configurations and 30 individual runs per configuration. Furthermore, to study the effect of fewer hyperparameters, we collected two landscapes of the same size for PPO, but with only three hyperparameters. The first reduced configuration space was manually designed using the hyperparameters learning rate, GAE $\lambda$ and clipping $\epsilon$. The second reduced configuration space consists of the hyperparameters GAE $\lambda$, entropy coefficient and advantage normalisation. The selection of the second subset was conducted using the most important coalition of hyperparameters as assessed by HyperShap (Wever et al., 2026). The landscapes have been computed with the same configuration space as the ARLBench landscapes and will be made publicly available.

The Lunar Lander environment was chosen, since in our experiments, it has shown benign behaviour with regards to its hyperparameters for both PPO and SAC, making it a good candidate to study the impact of scaling while avoiding the high variety of results of less benign landscapes. Furthermore, the environment comes with both a discrete and continuous version, allowing generation of landscapes for PPO, DQN and SAC.

## A.3 HPO-RL-Bench

*HPO-RL-Bench* (Shala et al., 2024) is a zero-cost benchmark for HPO in RL. In contrast to ARLBench, it is based on pre-computed grids of hyperparameter configurations and tabular lookups of performance values in the grid. Due to the curse of dimensionality, the grids in HPO-RL-Bench consist of only 2–3 hyperparameters

and a few distinct values per hyperparameter. Among others, the benchmark contains the algorithms PPO, DQN and SAC and is evaluated on 15 tasks from the ALE, 4 Classic Control tasks from Gymnasium and 3 MuJoCo tasks. The tabular data allows for training surrogates to predict the performance of a configuration. Each such landscape in HPO-RL-Bench consists of a $6 \times 6 \times 3$ grid, resulting in 108 configurations, each evaluated using 10 independent runs. The hyperparameters of the different algorithms and the respective configuration spaces are shown in Table 5.

| Hyperparameter | PPO | DQN | SAC |
|---|---|---|---|
| learning rate | $\{10^{-6}, 10^{-5}, 10^{-4}, 10^{-3}, 10^{-2}, 10^{-1}\}$ | | |
| discount | $\{0.8, 0.9, 0.95, 0.98, 0.99, 1.0\}$ | | |
| clipping | $\{0.1, 0.2, 03\}$ | - | |
| epsilon | - | $\{0.1, 0.2, 03\}$ | - |
| tau | - | | $\{0.001, 0.005, 0.01\}$ |

Table 5: The hyperparameter configuration space of HPO-RL-Bench for PPO, DQN and SAC. The learning rate and discount factor are hyperparameters shared between all three algorithms. All other hyperparameters are only specified for their respective algorithms.

For normalising the performance values in HPO-RL-Bench, we used the random policy performance for the minimum and an appropriate baseline from the literature as the maximum performance. The corresponding values are depicted in Table 4.

## A.4  LCBench

The landscapes in *LCBench* (Zimmer et al., 2021) have been collected across 35 tabular datasets in the OpenML benchmark suite (Vanschoren et al., 2014). Each landscape comes with 2000 randomly sampled configurations, with each configuration evaluated using 3 independent runs. All runs have been done with funnel-shaped multilayer perceptrons optimised with SGD and cosine annealing without restarts. Overall, 7 hyperparameters were sampled. The hyperparameters with their respective configuration spaces are depicted in Table 9.

Similar to existing LCBench surrogate benchmarks like YAHPOGym (Pfisterer et al., 2022), we used the *test balanced accuracy* as the performance metric. Since performance was thereby measured as classification accuracy, it is already normalised between 0 and 1.

| Hyperparameter | LCBench |
|---|---|
| batch size | $\log([16, 512])$ |
| learning rate | $\log([10^{-4}, 0.1])$ |
| momentum | $[0.1, 0.99]$ |
| weight decay | $[10^{-5}, 0.1]$ |
| num layers | $\{1, 2, 3, 4\}$ |
| max units | $\log([64, 1024])$ |
| max dropout | $[0.0, 1.0]$ |

Figure 9: The hyperparameter configuration space of the landscapes in LCBench.

## A.5  PD1 (HyperBO)

The *PD1* landscapes (Wang et al., 2024) have been computed on large architectures spanning both natural language and computer vision tasks. Following Mallik et al. (2023), we picked 4 out of the available 13 different tasks, including the batch size as a hyperparameter. Tasks have been selected for variety and to favour larger models. The landscapes consist of randomly sampled configurations, each evaluated with a single training run. Landscapes consisting of 2 different batch sizes consist of 800 configurations and landscapes with a single batch size consist of 400 configurations. The selected tasks with their respective configuration spaces are shown in Table 6.

The performance in PD1 was measured in *validation error rate* and is therefore already normalised between 0 and 1.

| Hyperparameter | Transformer | Wide-ResNet | XFormer |
|---|---|---|---|
| batch size | 2048 | $\{256, 2048\}$ | 64 |
| learning rate decay | | $[0.010543, 0.9885653]$ | |
| learning rate initial | | $\log([10^{-5}, 9.986256])$ | |
| learning rate power | | $[0.100811, 1.999659]$ | |
| momentum | | $[5.9 \cdot 10^{-5}, 0.9989986]$ | |

Table 6: The hyperparameter configuration space of the landscapes in PD1. The *ResNet* architectures have been sampled with two categorical choices for the batch size, whereas the batch size for the *Transformer* and *XFormer* was fixed to a single value.

## B  Surrogate Training

Table 7 summarises the configuration spaces used to tune the RF, XGB, GP, and SVM surrogate models. The configuration spaces for RF, XGB, and SVM have been adopted from NAS-Bench-301 Zela et al. (2022),

| Hyperparameter | Gaussian Process |
|---|---|
| kernel | $\{$RBF, Matern-0.5, Matern-1.5, Matern-2.5$\}$ |
| alpha | $[10^{-12}, 10^{-5}]$ |

| Hyperparameter | Random Forest |
|---|---|
| number estimators | $\log([16, 128])$ |
| minimal samples split | $[2, 20]$ |
| minimal samples leaf | $[1, 20]$ |
| maximal features | $[0.1, 1.0]$ |
| bootstrap | True, False |

| Hyperparameter | SVM |
|---|---|
| C | $\log([1, 20])$ |
| coef0 | $[-0.5, 0.5]$ |
| degree | $\log([1, 128])$ |
| epsilon | $\log([0.1, 0.99])$ |
| gamma | $\{$scale, auto$\}$ |
| kernel | $\{$linear, ref, poly, sigmoid$\}$ |
| shrinking | $\{$True, False$\}$ |
| tol | $\log([10^{-4}, 0.01])$ |

| Hyperparameter | XGB |
|---|---|
| max depth | $[1, 20]$ |
| minimal child weight | $\log([1, 100])$ |
| colsample byte | $[0.0, 1.0]$ |
| colsample bilevel | $[0.0, 1.0]$ |
| lambda | $\log([0.001, 1000])$ |
| alpha | $\log([0.001, 1000])$ |
| learning rate | $\log([0.001, 0.1])$ |

Table 7: The configuration spaces used for tuning our RFs, XGB, GPs and SVMs. We used Scikit-learn (Pedregosa et al., 2011) for our RF, GP and SVM surrogates and XGBoost (Chen & Guestrin, 2016) for XGB surrogates. The hyperparameter names correspond to the respective algorithm implementations.

while those for GPs were designed specifically for this work. Due to the substantially smaller size of the GP configuration space, random search tuning was performed over only 50 configurations, compared to 200 configurations for the other model classes.

### B.1  Surrogate Comparison

In Figure 10, we depict the performance of the four surrogates that have been fitted on our datasets. It can be observed that both RFs and XGB consistently achieve the highest median prediction performance across all tasks, with minimal differences between them. In contrast, prediction accuracy deteriorates substantially for GPs and SVMs. This is expected in the case of SVMs, given their relatively limited model capacity. GPs, however, are widely used as surrogate models in Bayesian optimisation for HPO, making their lower performance more surprising. Notably, this degradation in predictive accuracy is not uniform across all datasets but is particularly pronounced on RL tasks. For the supervised learning benchmarks LCBench and PD1, GPs perform comparably to RFs, whereas their performance drops significantly on ARLBench and

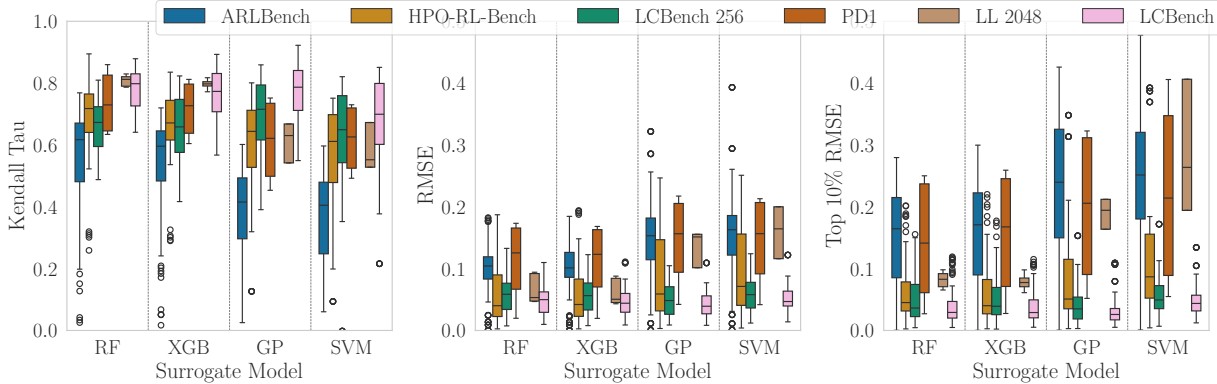

Figure 10: Prediction performance of RFs, XGB, GPs and SVMs. The top 10% RMSE is the RMSE computed on only the 10% best-performing configurations in the landscape. The landscapes of ARLBench, LunarLander 2048 and HPO-RL-Bench have been subsampled to three seeds per configiguration to be comparable to LCBench.

LunarLander 2048. These findings suggest that the choice of surrogate model plays a particularly critical role in the context of RL tasks.

## B.2   Surrogate Validation

To validate the correctness of our tuned surrogate models and to assess whether their performance is comparable to established alternatives, we conducted a comparison against the official surrogate models provided by YAHPOGym (Pfisterer et al., 2022) for the LCBench benchmark. Specifically, we evaluated our RF and XGB surrogate models on the prediction of *balanced accuracy*, using the same set of 34 tasks supported by YAHPOGym. We selected RFs and XGB due to their improved performance compared to GPs and SVMs.

The YAHPOGym surrogates have been trained on full learning curves, and thus we compare their predictive performance in two settings: (i) predictions restricted to the final epoch, which is directly comparable to the targets used in our surrogates, and (ii) predictions across the full learning curve. For consistency, we employed the official configuration set released as part of the *data_2k_lw.zip* dataset from LCBench, which was also used to train the YAHPOGym surrogates.

To prevent data leakage, our models were assessed using the mean performance across all test folds in a 10-fold cross-validation setup. The comparative results are presented in Figure 11.

The results show that our surrogate models achieve Spearman and Kendall's $\tau$ correlations comparable to those of YAHPOGym, validating the correctness of our implementations.

## B.3   Performance Metric Comparison

The most common performance objective in RL is the final performance of a training run. However, RL is known to exhibit substantial variability across evaluation checkpoints and is prone to catastrophic forgetting. These factors make predicting the final performance more sensitive to outliers, reducing the robustness of such predictions.

Alternative metrics, such as the maximal performance or AUC over the entire training run, can provide more robust assessments of training performance by mitigating the influence of these detrimental effects. To better understand how the choice of performance metric affects surrogate model accuracy, we trained surrogate models for each metric. The comparison for ARLBench and LunarLander 2048 is shown in Figure 12.

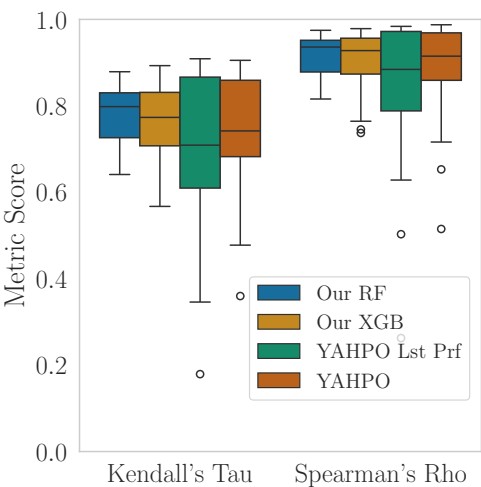

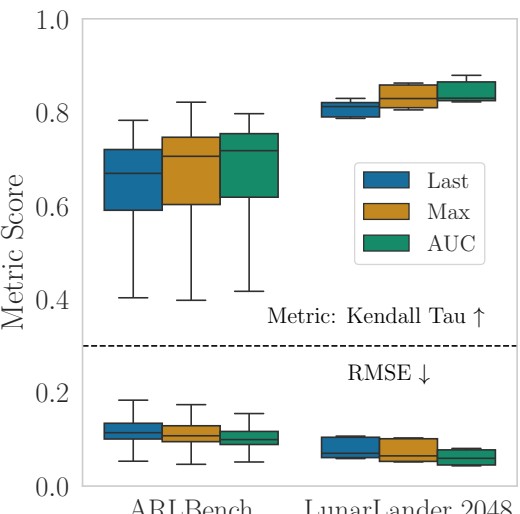

Figure 11: Comparison of the performance of our RF and XGB surrogate models with YAHPOGym on LCBench. For YAHPOGym, we report results for both the final epoch and the full learning curve. Our models show comparable accuracy, demonstrating the correctness of our approach.

Figure 12: The Boxplots compare the quality of surrogate models predicting different performance objectives of RL trainings. Surrogate performance is measured in Kendall's $\tau$ and RMSE of its predictions. We consistently observe that the AUC results in the best surrogate models, followed by the maximal performance and the final performance.

We consistently observe that using the AUC produces the most accurate surrogate models, followed by maximal performance and then final performance. This aligns with the intuition that maximal performance and AUC are less sensitive to performance outliers, simplifying performance prediction. However, the differences are not substantial. In particular, for ARLBench, we still observe a considerable performance spread across the various environments.

### B.4 Tunability

To better understand the importance of tuning the hyperparameters of surrogate models, we analysed their tunability (Probst et al., 2019). Tunability is the average performance difference between the single-best configuration on all tasks and the best achieved performance for configurations tuned for each task. For a model $M$ with $N$ different tasks $t_1, \cdots, t_n$ and function $h_i(c)$ mapping configurations $c$ to the respective fitted surrogate accuracy performance on task $t_i$, we define

$$c^* \in \underset{c}{\mathrm{argmax}} \ \frac{1}{N} \cdot \sum_{i \in [1,N]} h_i(c) \qquad (1)$$

as the configuration performing best overall. The tunability of model $M$ is then defined as

$$Tun(M) = \frac{1}{N} \cdot \sum_{i \in [1,N]} \max_{c} [h_i(c)] - h_i(c^*). \qquad (2)$$

We depict the tunability measured in RMSE for different surrogate models and datasets in Table 8. We observe that for RFs, XGB, and GPs, no single configuration consistently performs well across all tasks within a dataset, with an average RMSE between 0.023 and 0.13. In contrast, SVMs show perfect tunability (0.0 across all datasets), but their substantially lower predictive performance does not make them practically relevant. Among RFs, XGB, and GPs, tunability varies notably. GPs are the least sensitive to per-instance tuning, while RFs show slightly worse tunability. XGB models are the most tuning-dependent, with an

| Model | Dataset | | | | |
| --- | --- | --- | --- | --- | --- |
| | ARLBench | Lunar Lander 2048 | HPO-RL-Bench | LCBench | PD1 |
| RF | $-0.032$ | $-0.023$ | $-0.092$ | $-0.023$ | $-0.048$ |
| XGB | $-0.077$ | $-0.084$ | $-0.13$ | $-0.078$ | $-0.098$ |
| GP | $0.019$ | $-0.009$ | $-0.045$ | $-0.011$ | $-0.05$ |
| SVM | 0.0 | | | | |

Table 8: The tunability of different surrogate models and datasets measured in the loss of RMSE.

average tunability of at least $-0.077$. Given their comparable predictive accuracy, RFs may thus be a more robust choice than XGB models.

Overall, these results indicate that per-instance tuning is necessary to achieve optimal surrogate performance. The sensitivity of surrogate models to their hyperparameters also highlights the importance of reviewing default settings in hyperparameter analysis tools such as Weights & Biases (Biewald, 2020), DeepCAVE (Sass et al., 2022) and Optuna (Akiba et al., 2019). For instance, DeepCAVE uses a default of 16 trees in RFs for LPI or fANOVA, whereas our tuned RFs comprise on average 59 trees.

## C   Unimodality Analysis

In Figure 13, we depict the comparison of different values of $k$ in the $k$-nearest neighbour computation of the neighbourhood relation.

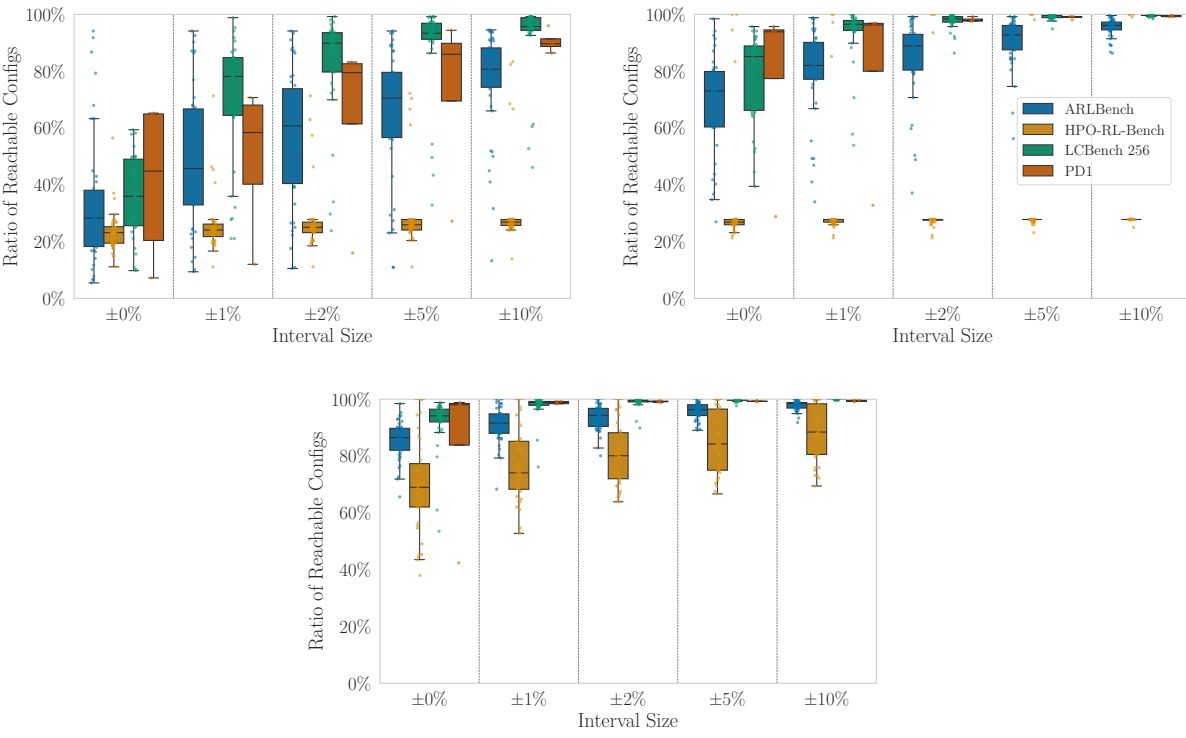

Figure 13: Comparison of the unimodality analysis for different values of neighbours in the $k$-nearest neighbour algorithm. The Figures depict the results from left to right and top to bottom for the values 5, 10 and 15. While the rate of convergence to full unimodality differs, the overall trends remain similar.

# D   HyperShap Analysis

*HyperShap* (Wever et al., 2026) uses game theory to assess hyperparameter importance, tunability and more. For our analysis, we are interested in hyperparameter importance and thus use the Ablation Game, where a configuration of interest $\lambda^*$ (in our case the best overall configuration in our dataset) should be reached from a default configuration $\lambda^0$. The value of each hyperparameter or hyperparameter coalition in moving toward $\lambda^0$ is quantified using Shapley values (Shapley, 1953). To approximate a global importance score less tied to a specific $\lambda^0$, we perform this analysis across 100 randomly sampled configurations.

For the evaluation, we use the full analytical evaluation of all coalitions (instead of an approximation). We show the values from the Moebius transform of order $n$ where $n$ is the number of hyperparameters so we can include all possible interaction effects.

## D.1   Consistency Across 2048 Configurations

To verify the values returned by HyperShap are consistent enough for configuration space reduction, we sample ten subsets for 1024 configurations each from our full 2048 Lunar Lander configurations and fit a surrogate for each one. Table 9 shows that the highest valued hyperparameters in the full 2048 configuration surrogate are still overwhelmingly the most important ones in the subsampled surrogates. The weakest coalition of these, entropy coefficient and normalised advantage, does not appear in the top-5 of every model, but only three. It is replaced by coalitions of the GAE $\lambda$ with the value function coefficient (3 times) or learning rate (3 times) as well as once with a coalition of GAE $\lambda$, normalise advantage and the minibatch size. Overall, however, this still indicates high consistency between the surrogates trained on the subsamples and thus that our HyperShap analysis yields reliable insights into our surrogates.

## D.2   Results with Fewer Configurations

For the automatic configuration space reduction, the consistency of results across dataset sizes is obviously of interest. Therefore, we include an overview of the top-5 hyperparameter coalitions for each dataset size we tested for Lunar Lander in Table 10. Again, we colour the hyperparameters making up the top-5 effects in the 2048 configuration surrogate for comparison. We observe a strong trend from the 32 configuration dataset to the full 2048 configurations. The GAE $\lambda$ is always part of the most important coalition and the entropy coefficient is either in the most important or the second most important one. In every surrogate, there is one more hyperparameter in the top-5 coalitions. For 64, 256, 512 and 1024 configurations, normalise advantage is included just like for the full-sized set. For 256 and 512 configurations, there is also a fourth hyperparameter, the learning rate and value function coefficient, respectively. The learning rate also appears in the 128 configuration dataset. The smallest 32 configuration dataset includes the clipping epsilon instead.

This considerable overlap across dataset sizes points towards a strong trend in the data, which is present already with few configurations on Lunar Lander. We should not assume that this is the case for each environment, however. These results support our finding of surrogate performance growing only slowly, since the importance of different hyperparameters shifts only gradually. It seems that this is since surrogates can identify broad trends in the landscape very early on, but that it takes a lot of additional data for more complex relationships to arise. This is consistent with our LPI results as well.

## D.3   Results with Fewer Seeds

Finally, we also include the HyperShap analysis of the 2048 configuration dataset surrogates trained on smaller numbers of random seeds. Differences in importance between these variations could indicate strong biases in some evaluation seeds. Table 11 shows a high level of consistency across seeds, similar to that in dataset sizes. The biggest difference is the inclusion of the value coefficient in the top-5 effects for one through 15 seeds. We see this hyperparameter in some of the subsamples of the full dataset as well, so it is not completely new, but it is not as consistently included. The three most important hyperparameters, however, stay the same across variations in seed number. As Lunar Lander is not one of the more extreme

| Hyperparameter Coalition | Model 1 | Model 2 | Model 3 | Model 4 | Model 5 |
|---|---|---|---|---|---|
| GAE λ | −0.05 | −4% | −4% | −4% | −0.05 |
| Entropy Coefficient, GAE λ, Normalise Advantage | 4% | 3% | 3% | 4% | 3% |
| Entropy Coefficient, GAE λ | −4% | −2% | −3% | −3% | −3% |
| GAE λ, Normalise Advantage | −4% | −3% | −3% | −3% | −3% |
| GAE λ, Value Function Coefficient | −2% | −2% | −3% | - | - |
| GAE λ, Learning Rate | - | - | - | −2% | - |
| Entropy Coefficient, Normalise Advantage | - | - | - | - | −2% |
| GAE λ, Normalise Advantage, Minibatch Size | - | - | - | - | - |
| Hyperparameter Coalition | Model 6 | Model 7 | Model 8 | Model 9 | Model 10 |
| GAE λ | −4% | −4% | −4% | −4% | −0.05 |
| Entropy Coefficient, GAE λ, Normalise Advantage | 4% | 3% | 3% | 3% | 4% |
| Entropy Coefficient, GAE λ | −3% | −2% | −2% | −2% | −2% |
| GAE λ, Normalise Advantage | −3% | −3% | −3% | −3% | −4% |
| GAE λ, Value Function Coefficient | - | - | - | - | - |
| GAE λ, Learning Rate | - | −2% | −2% | - | - |
| Entropy Coefficient, Normalise Advantage | −2% | - | - | - | −2% |
| GAE λ, Normalise Advantage, Minibatch Size | - | - | - | 2% | - |

Table 9: Top-5 interaction values in surrogates trained on subsamples of the full 2048 lunar lander dataset. Colored hyperparameters are part of the overall top-5 highest valued coalitions. Interaction values measure improvement over the mean performance in percentage.

| 32 Configs | 64 Configs | 128 Configs | 256 Configs | 512 Configs | 1024 Configs | 2048 Configs |
|---|---|---|---|---|---|---|
| Ent. Coef., GAE λ | Ent. Coef., GAE λ | GAE λ | GAE λ | GAE λ, Norm. Adv. | GAE λ | Ent. Coef., GAE λ, Norm. Adv. |
| GAE λ | GAE λ | Ent. Coef., GAE λ | Ent. Coef., GAE λ | Ent. Coef., GAE λ, Norm. Adv. | Ent. Coef., GAE λ, Norm. Adv. | GAE λ |
| Clip. Eps. | Ent. Coef. | Ent. Coef. | Ent. Coef. | GAE λ | Ent. Coef., GAE λ | GAE λ, Norm. Adv. |
| Ent. Coef. | GAE λ, Norm. Adv. | Learning Rate, GAE λ | Learning Rate, GAE λ | Ent. Coef., GAE λ | GAE λ, Norm. Adv. | Ent. Coef., GAE λ |
| Clip. Eps., GAE λ | Ent. Coef., GAE λ, Norm. Adv. | Learning Rate | GAE λ, Norm. Adv. | GAE λ, Value Coef. | Ent. Coef. | Ent. Coef., Norm. Adv. |

Table 10: Top-5 interaction values in surrogates trained on Lunar Lander datasets of different sizes. Colored hyperparameters are part of the top-5 highest valued coalitions in the full 2048 configuration dataset.

| 1 Seed | 3 Seeds | 5 Seeds | 10 Seeds | 15 Seeds | 30 Seeds |
|---|---|---|---|---|---|
| GAE λ | GAE λ | GAE λ | GAE λ | GAE λ | Ent. Coef., GAE λ, Norm. Adv. |
| GAE λ, Norm. Adv. | GAE λ, Norm. Adv. | GAE λ, Norm. Adv. | GAE λ, Norm. Adv. | GAE λ, Norm. Adv. | GAE λ |
| Ent. Coef., GAE λ, Norm. Adv. | Ent. Coef., GAE λ, Norm. Adv. | Ent. Coef., GAE λ, Norm. Adv. | Ent. Coef., GAE λ, Norm. Adv. | Ent. Coef., GAE λ, Norm. Adv. | GAE λ, Norm. Adv. |
| GAE λ, Value Coef. | Ent. Coef., GAE λ | Ent. Coef., GAE λ | Ent. Coef., GAE λ | Ent. Coef., GAE λ | Ent. Coef., GAE λ |
| Learning Rate, GAE λ | GAE λ, Value Coef. | GAE λ, Value Coef. | GAE λ, Value Coef. | GAE λ, Value Coef. | Ent. Coef., Norm. Adv. |

Table 11: Top-5 interaction values in surrogates trained on Lunar Lander datasets evaluated across different amounts of seeds. Colored hyperparameters are part of the top-5 highest valued coalitions in the full 2048 configuration dataset.

environments when it comes to noise, this is a good sign that on domains that have a somewhat predictable noise structure, the surrogate predictions do not heavily depend on the seeding in the training data.

## D.4 Landscape Properties of Reduced Configuration spaces

In Figure 14, we depict the unimodality analysis and noise comparison between the Lunar Lander 2048 landscape and the reduced landscapes. While both reduced landscapes show less benign unimodal behaviour, their noise distribution improves compared to the full landscape.

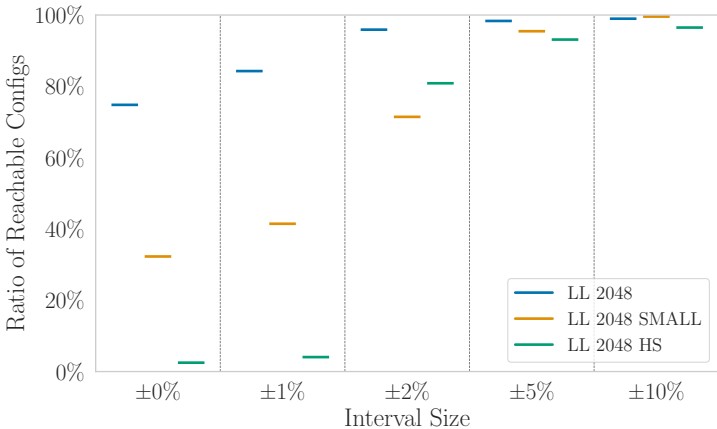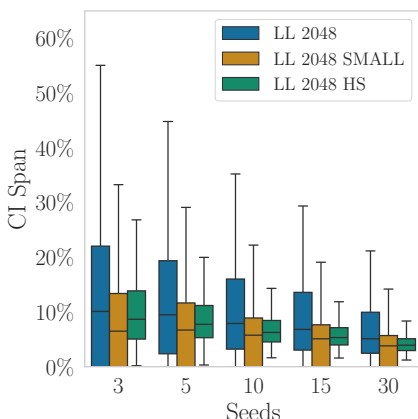

Figure 14: The left shows the comparison of the unimodality between the Lunar Lander 2048 performance landscapes and the expert reduced and HyperShap reduced configuration spaces. The right shows the distribution of the confidence intervals in the performance landscapes.

## E  Local Parameter Importance

*Local Parameter Importance* (LPI) estimates the influence of individual hyperparameters on performance in the context of a specific configuration (Biedenkapp et al., 2018). It is computed by quantifying the variance in performance when varying each hyperparameter individually, normalised by the total variance across all hyperparameters.

Let a hyperparameter configuration $c$ consist of a set of hyperparameters $\mathcal{H}$, and let $\Theta$ denote the domain of hyperparameter values $h \in \mathcal{H}$. We write $c[c_h = v]$ to denote the configuration where hyperparameter $h$ is set to value $v$, while all other hyperparameters remain as in $c$. Given a function $f : \Theta \to \mathbb{R}$ that maps a configuration to its training performance, the LPI of hyperparameter $h$ at configuration $c$ is defined as:

$$LPI(h \mid c) = \frac{\mathrm{Var}_{v \in \Theta_h} f(c[c_h = v])}{\sum_{h' \in \mathcal{H}} \mathrm{Var}_{v \in \Theta_{h'}} f(c[c_{h'} = v])}. \tag{3}$$

The resulting LPI values represent the proportion of performance variation attributable to each hyperparameter. Higher values indicate greater local influence on performance.

In practice, a surrogate model is commonly used as $f$ in Equation 3, with $n_h$ configurations sampled per hyperparameter $h$ to estimate the variance. The surrogate is trained on previously collected configurations and their associated performance values. This allows efficient LPI computation but may introduce inaccuracies due to surrogate prediction errors. Alternatively, $f$ can be evaluated by directly executing the training task for each sampled configuration, typically over multiple random seeds.

### E.1  Full Configuration Space LPI Plots

In this section, we present the extended LPI results comparing ground-truth values of performance with surrogate predictions for the full configuration space for PPO and DQN. Figure 15, 16 and 17 depict the importance of the full configuration space for both PPO and DQN for the performance metrics last, max and AUC, respectively.

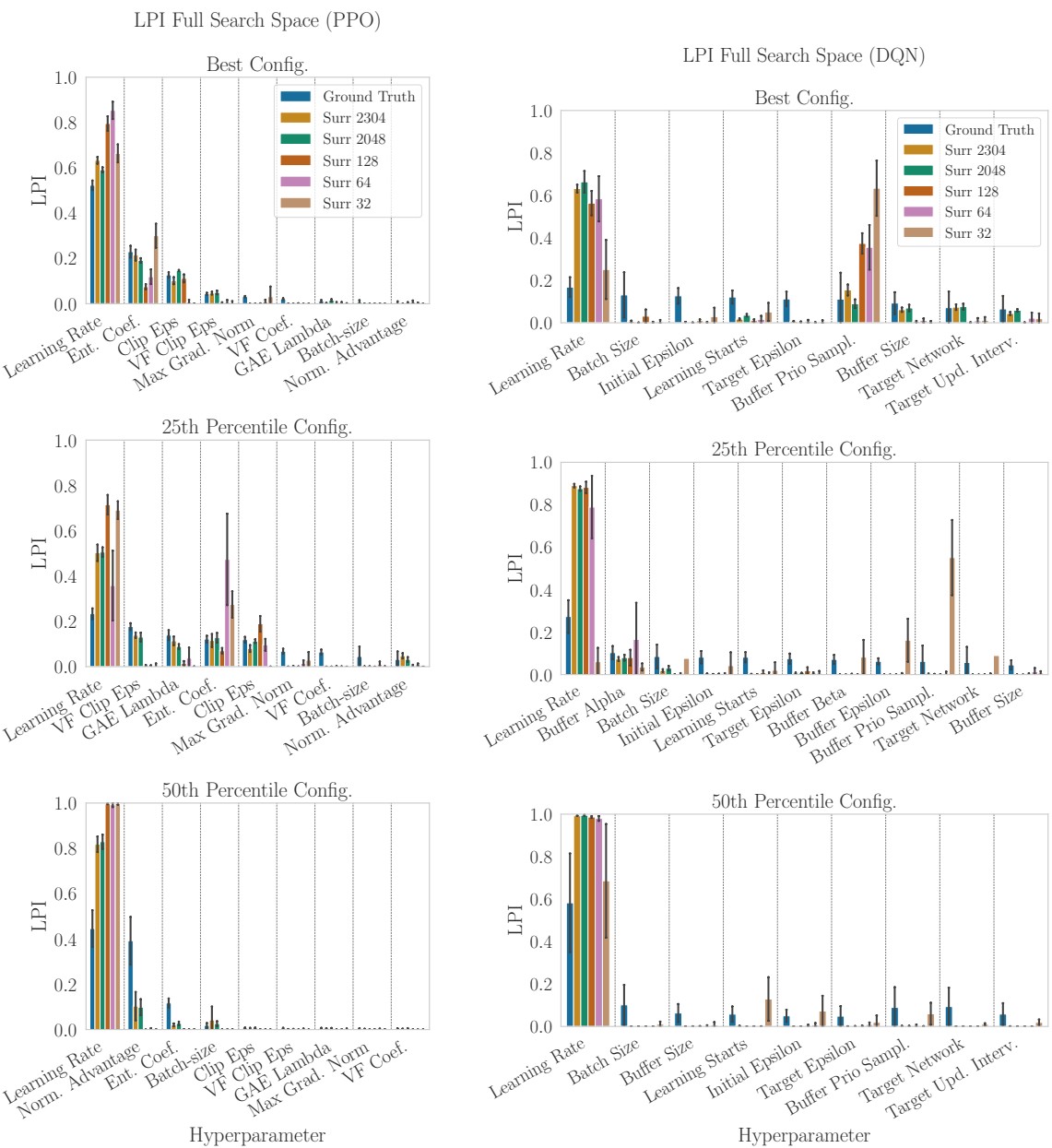

Figure 15: Summary of LPIs of the ground-truth performance values and surrogate predictions for the full ARLBench configuration space and performance metric last. For DQN, the number of hyperparameters differs between different configurations due to inactive conditional hyperparameters dependent on the buffer properties and target network. The surrogate with 2304 configurations is the surrogate trained on the combined random search and configurations found by SMAC.

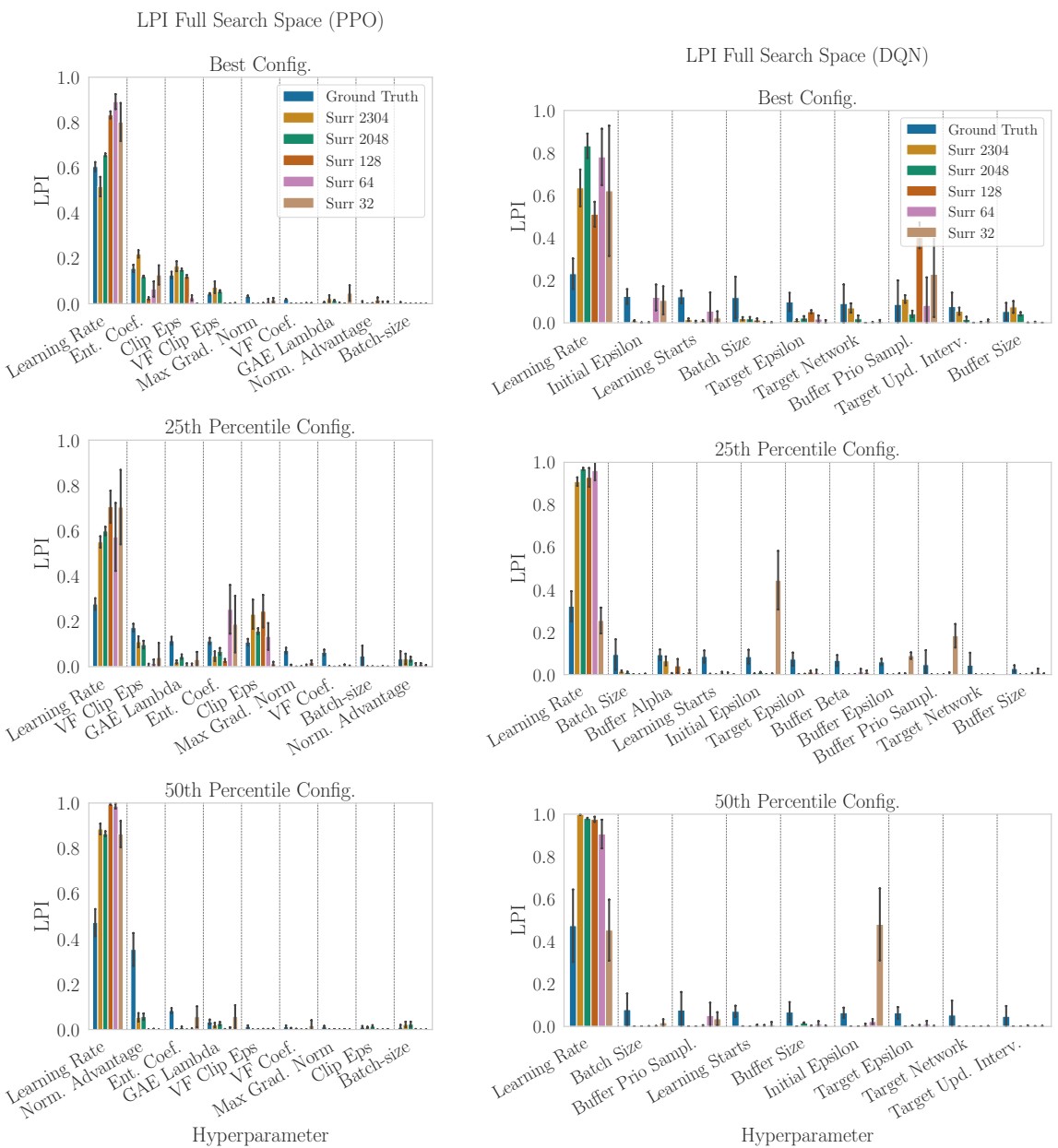

Figure 16: Summary of LPIs of the ground-truth performance values and surrogate predictions for the full ARLBench configuration space and performance metric max. For DQN, the number of hyperparameters differs between different configurations due to inactive conditional hyperparameters dependent on the buffer properties and target network. The surrogate with 2304 configurations is the surrogate trained on the combined random search and configurations found by SMAC.

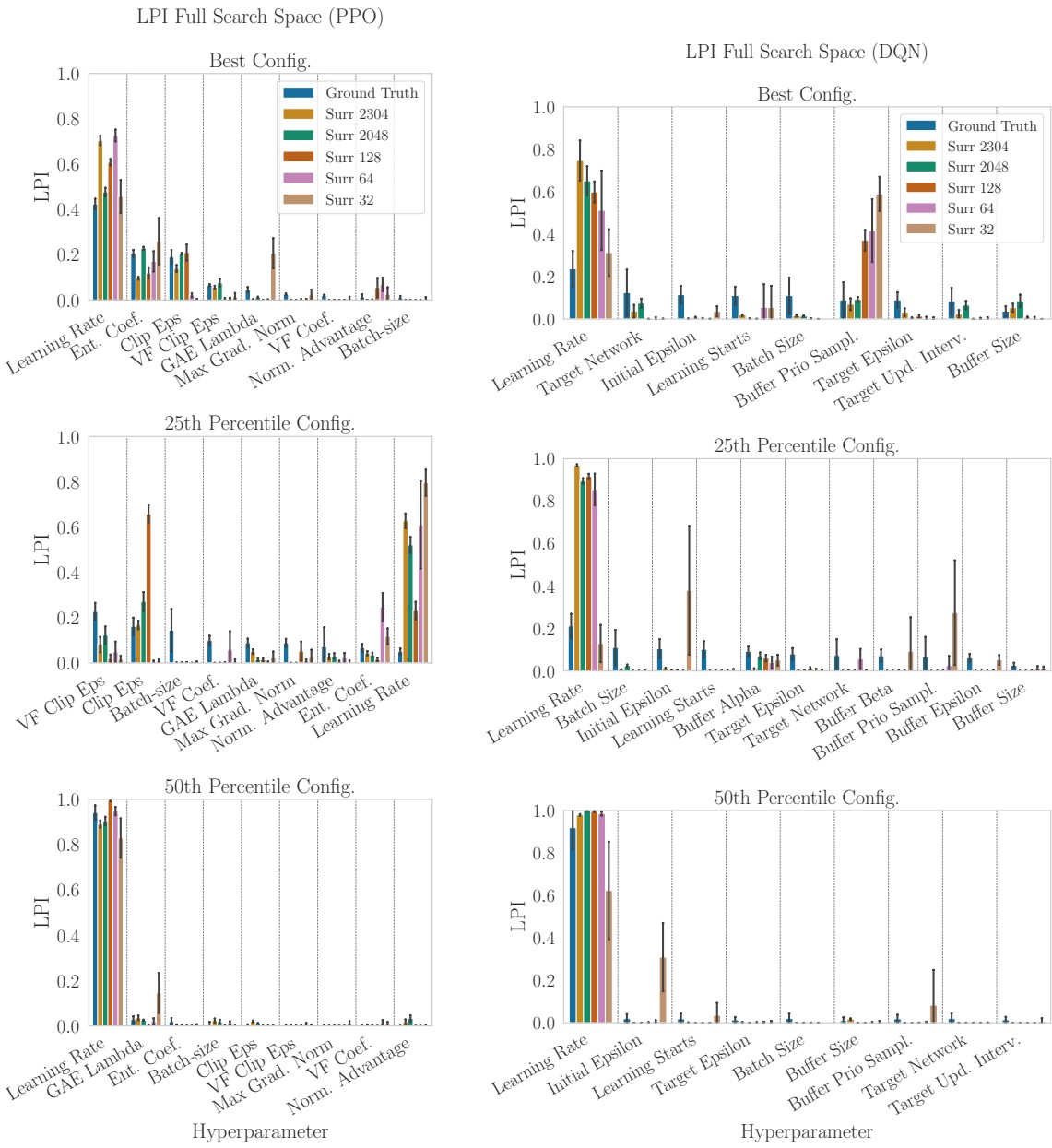

Figure 17: Summary of LPIs of the ground-truth performance values and surrogate predictions for the full ARLBench configuration space and performance metric AUC. For DQN, the number of hyperparameters differs between different configurations due to inactive conditional hyperparameters dependent on the buffer properties and target network. The surrogate with 2304 configurations is the surrogate trained on the combined random search and configurations found by SMAC.

Figure 18 presents the RMSE when comparing the surrogate predictions to the ground-truth values.

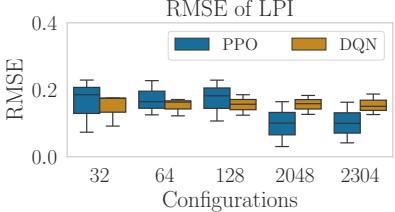

Figure 18: The RMSE of the surrogate predictions compared to the ground-truth performance values computed over all three performance metrics last, max and AUC.

We find that for PPO, the surrogate predictions become more precise with an increased number of configurations. For DQN, this improvement is not evident, which could explain why we also do not observe improved LPI accuracy for DQN when training surrogates with more configurations. A possible reason for this behaviour is that the DQN configuration spaces contain several conditional hyperparameters, whereas PPO contains no conditional dependencies. Such conditional dependencies are not explicitly modelled in the surrogates and might result in performance degradation.

### E.2 Reduced Configuration Space LPI Plots

In this section, we present the LPI results for the manually and HyperShap reduced configuration spaces. Similarly, to the main LPI results, we picked the best, 25th and 50th percentiles configurations for each reduced space, respectively. The raw results are depicted in Figure 19, 20 and 21 for the performance metrics last, max and AUC, respectively.

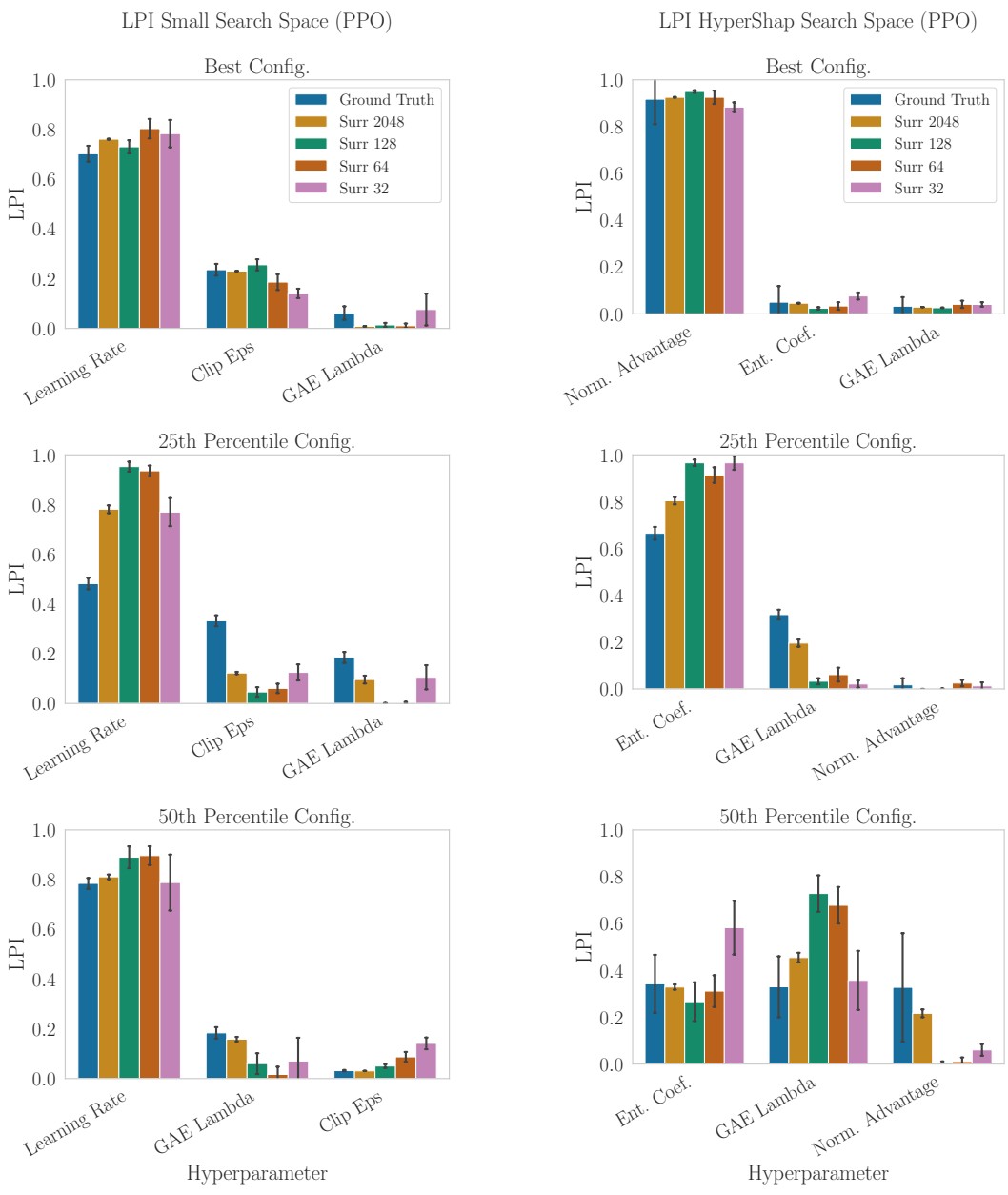

Figure 19: Summary of LPIs of the ground-truth performance values and surrogate predictions for the reduced configuration spaces and performance metric last.

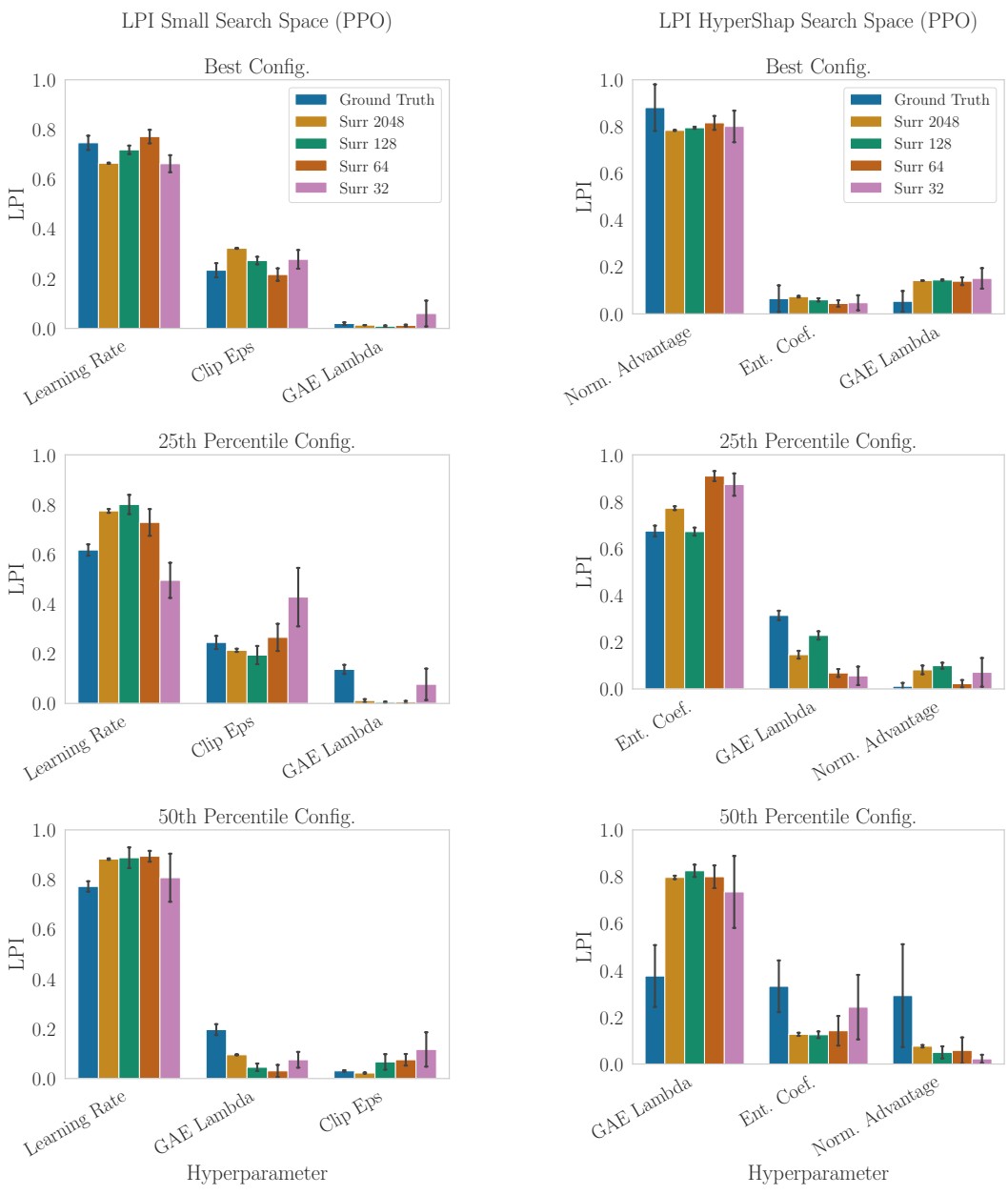

Figure 20: Summary of LPIs of the ground-truth performance values and surrogate predictions for the reduced configuration spaces and performance metric max.

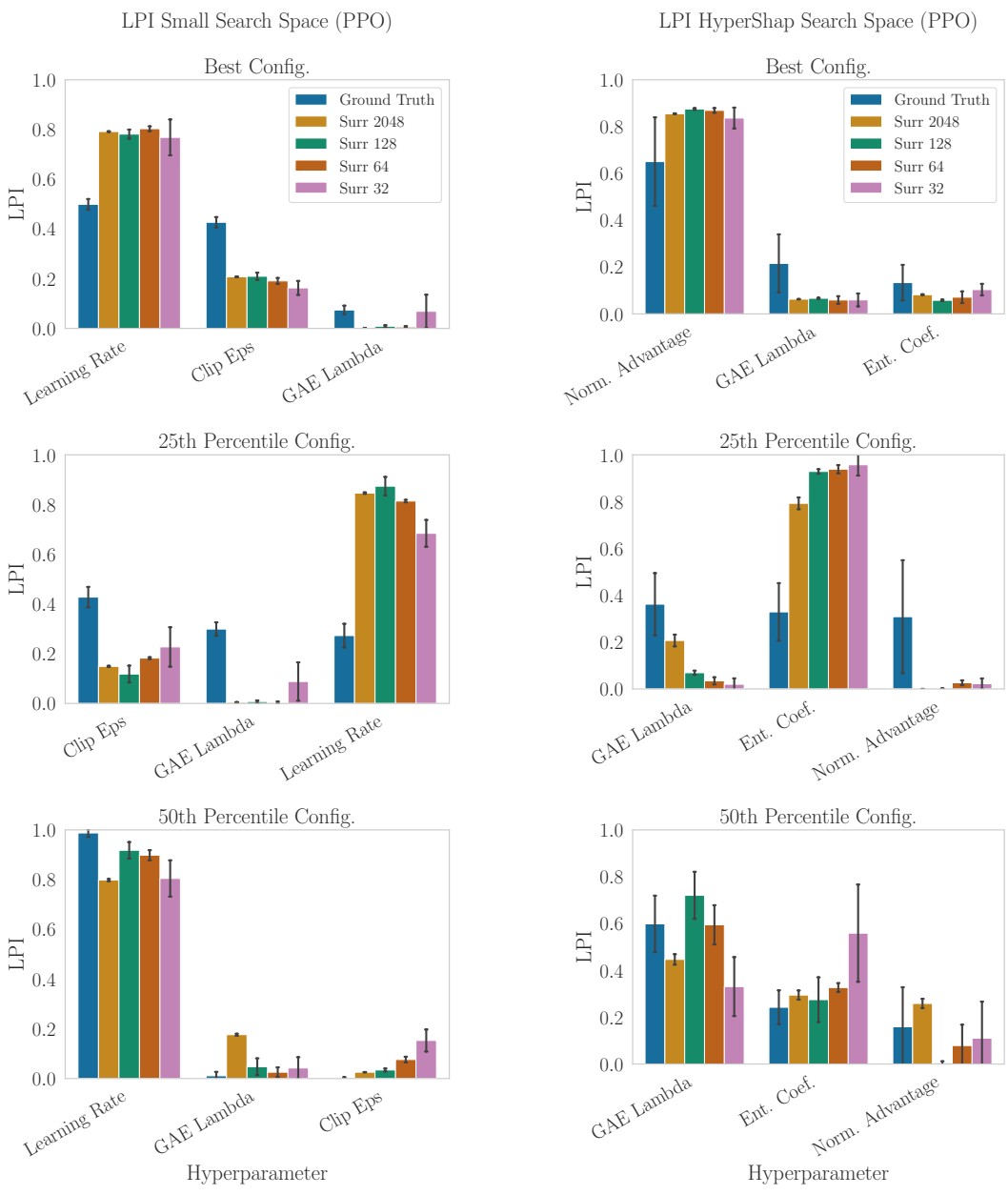

Figure 21: Summary of LPIs of the ground-truth performance values and surrogate predictions for the reduced configuration spaces and performance metric AUC.

Figure 22 presents the RMSE when comparing the surrogate predictions to the ground-truth values.

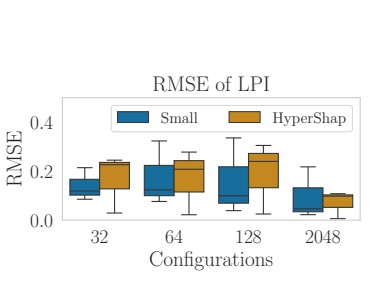 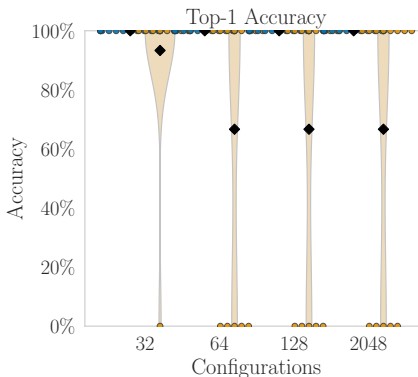 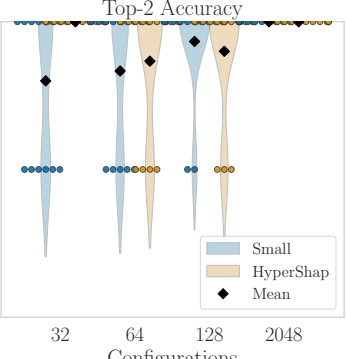

Figure 22: **Left:** The RMSE of the surrogate predictions compared to the ground-truth performance values. **Right:** The top-1 and top-2 accuracy of the resulting importance estimates of the surrogate models. All values are computed over all three performance metrics: last, max and the AUC.

The RMSE of the prediction accuracy does substantially improve when training surrogates with more configurations, whereas for the manually reduced configuration space, the improvement already stagnates for more than 64 configurations.

## F  Ablation Paths

An *Ablation Path* is a sequence of configurations in which hyperparameters are incrementally set to values from a reference configuration (e.g., the best-performing one). Starting from a source configuration (e.g., random configuration), one hyperparameter at a time is changed to a target configuration, and its impact on performance is recorded. At each step, the hyperparameter whose change yields the highest performance improvement is selected. This greedy process continues until the full target configuration is reconstructed. Ablation paths help to understand the contribution of individual hyperparameters and their interactions along a trajectory toward optimal performance. The resulting ranking of hyperparameters can be interpreted as an importance ranking, with hyperparameters that need to be adjusted first being more important than the latter ones.

To evaluate the performance of our surrogate models on ablation paths, we measured their ability to identify the most influential hyperparameter whenever the ground-truth ablation path exhibits a performance change greater than 10% of the performance spread between the source and target configuration. This choice does not fully capture the quality differences between surrogate and ground-truth ablation paths, since predicting no substantial change in all hyperparameters correctly is an important component of ablation as well, not captured by our metric. However, predicting substantial performance shifts is commonly important at the beginning of ablation paths, where the correctness matters most.

### F.1  Full configuration space Ablation Paths

In Figure 23, 24 and 25, we present the ablation paths for the performance metrics last, max and AUC, respectively. For each ablation path, we depict the ground-truth ablation path with the respective performance changes predicted by our surrogate model. Further, we depict the ablation path under the surrogate performance predictions for each hyperparameter change.

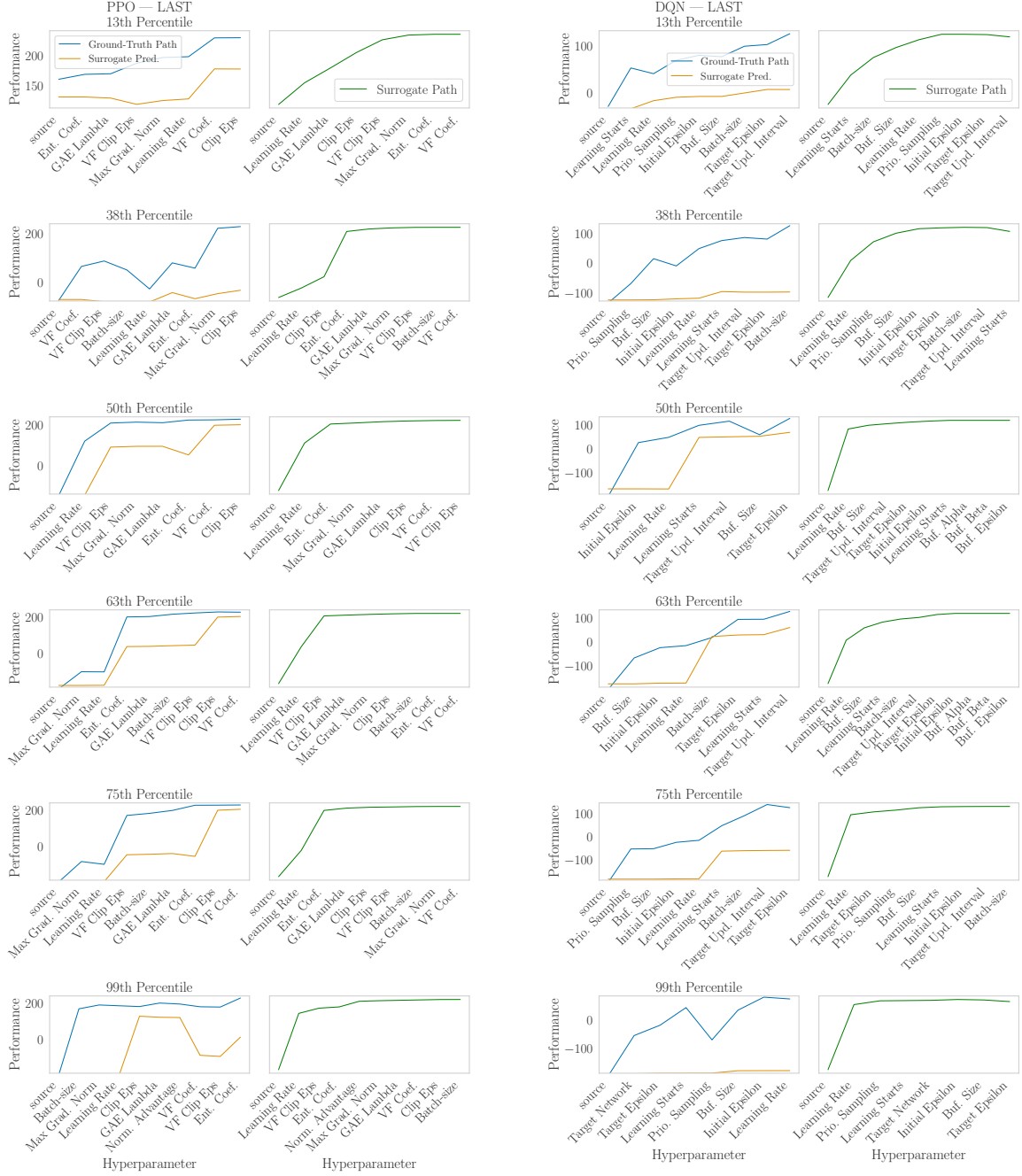

Figure 23: Ablation paths for PPO and DQN on the full Lunar Lander configuration space with the following subplots. **Left:** The ground-truth ablation paths for the performance metric last depicted in blue, with the respective surrogate performance predictions for each hyperparameter in orange. **Right:** The resulting ablation path under the surrogate model predictions in green. For the path construction, the Lunar Lander surrogate model fitted on 2048 configurations was used.

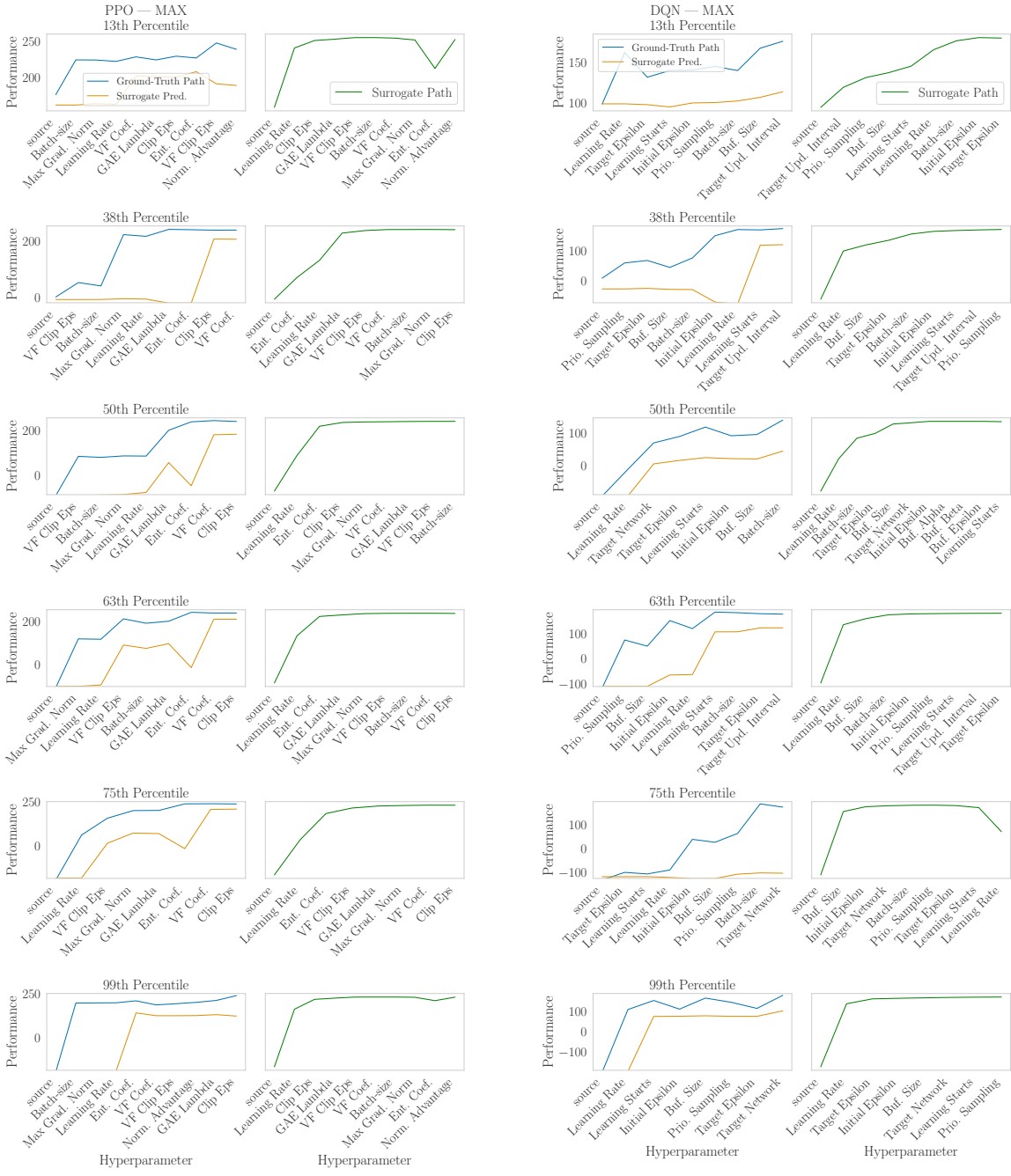

Figure 24: Ablation paths for PPO and DQN on the full Lunar Lander configuration space with the following subplots. **Left:** The ground-truth ablation paths for the performance metric max depicted in blue, with the respective surrogate performance predictions for each hyperparameter in orange. **Right:** The resulting ablation path under the surrogate model predictions in green. For the path construction, the Lunar Lander surrogate model fitted on 2048 configurations was used.

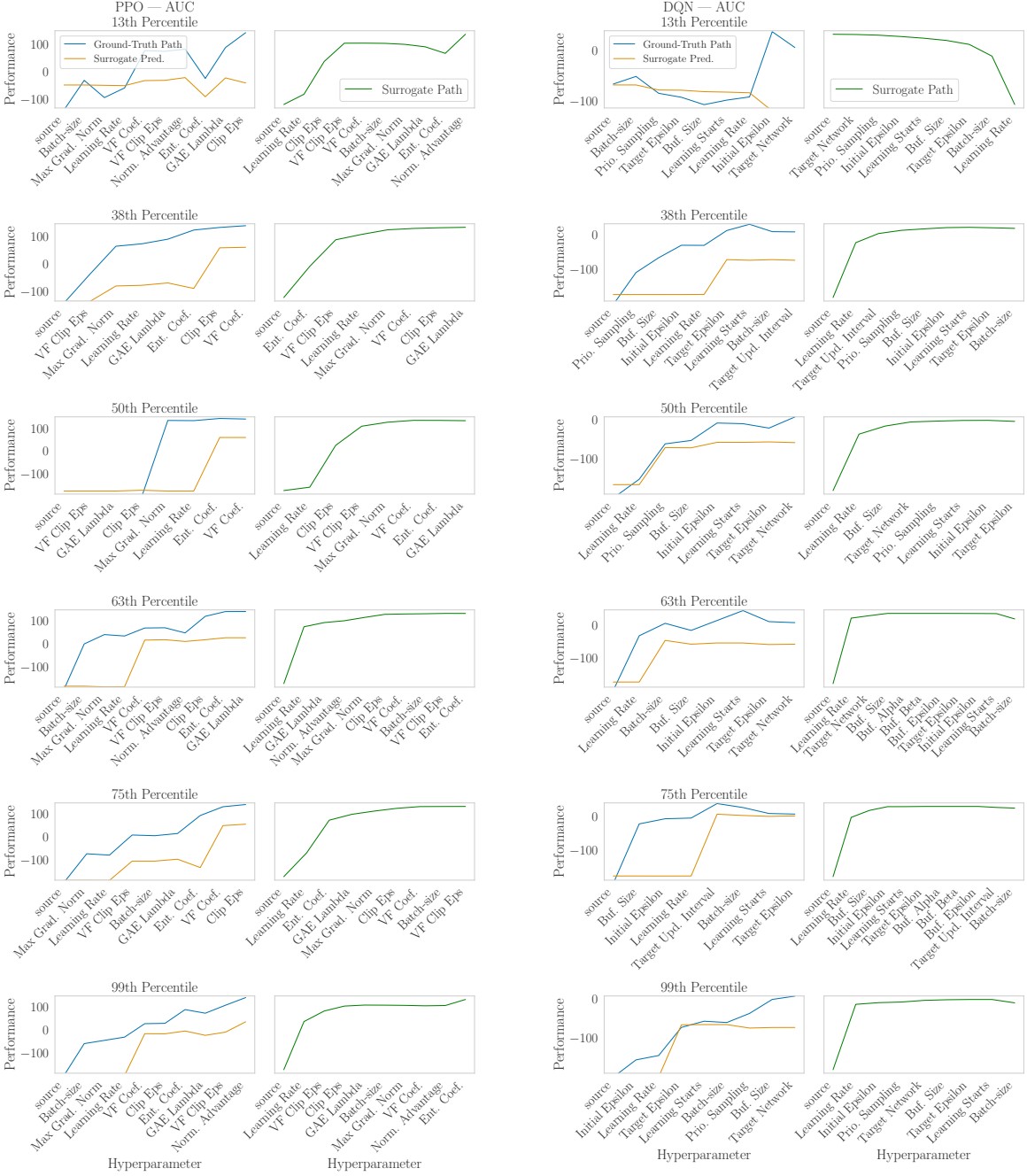

Figure 25: Ablation paths for PPO and DQN on the full Lunar Lander configuration space with the following subplots. **Left:** The ground-truth ablation paths for the performance metric AUC depicted in blue, with the respective surrogate performance predictions for each hyperparameter in orange. **Right:** The resulting ablation path under the surrogate model predictions in green. For the path construction, the Lunar Lander surrogate model fitted on 2048 configurations was used.

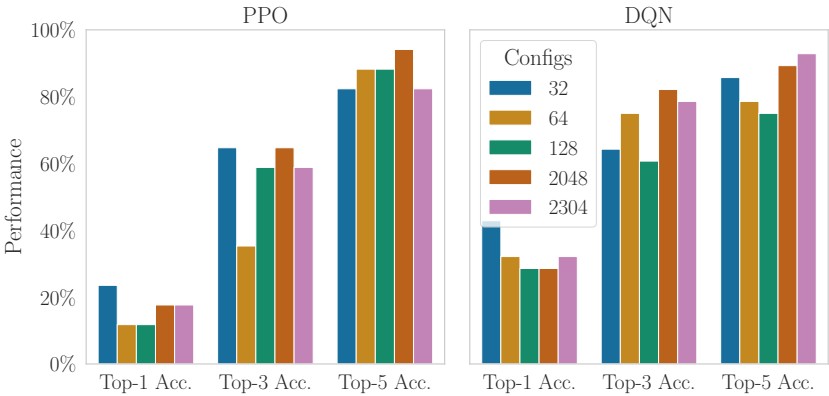

Figure 26: The accuracy of the surrogate model of predicting the most significant hyperparameter whenever a performance shift or more than 10% occurs in the ground-truth ablation paths.

Neither the surrogate predictions nor the ablation paths constructed by the surrogate are good estimates of the ground-truth ablation paths. The surrogate ablations show significantly different shapes from the ground-truth and do not reliably predict changes in performance correctly. Similarly, the hyperparameter rankings of the surrogate deviate substantially from the ground-truth rankings.

In Figure 26, we present the accuracy of the surrogate ablation paths to correctly predict significant hyperparameter changes of more than 10% of the performance spread between source and target configurations. A top-1 accuracy of 20% means that in only 20% of the cases where the ground-truth shows a significant performance shift, the surrogate correctly identifies the responsible hyperparameter. This is a notably low accuracy, especially considering that altering a single hyperparameter in an ablation path can drastically change subsequent steps. Such results suggest that the surrogate models are not reliably capturing the underlying dependencies. Furthermore, we can not observe an increased performance when using more configurations to train the surrogate models. Ablation paths commonly consist of many high-performing configurations and we previously observed that our surrogate models perform especially badly in high-performing regions of the landscape, which are sampled scarly. Therefore, we hypothesise that, in particular, more high-performing configurations are necessary to train well-performing surrogate models.

To verify that the low top-1 accuracy is not merely due to noise in the configurations, we report the average rank distance between the surrogate prediction and the true most important hyperparameter in Figure 27. Additionally, Figure 27 presents the average absolute performance difference between the surrogate prediction and the best-performing configuration. For PPO, we observe a median rank distance of approximately 2 and an average absolute performance difference of around 20%. In particular, the latter value indicates that the worse performance of the surrogate models can not be explained merely by noise. In the case of DQN, these differences appear much smaller; however, further analysis revealed that the surrogate models assign uniformly low performance scores to most configurations. This behaviour, while yielding low numerical differences, is also undesirable and indicates a lack of meaningful discrimination across configurations.

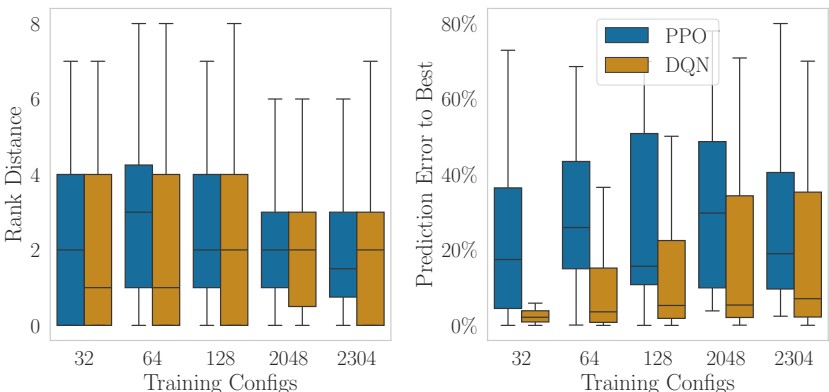

Figure 27: **Left:** The rank distance between the most important hyperparameter of the ground-truth and the top-ranked prediction of the surrogate model. **Right:** The difference between the performance of the most important hyperparameter and its surrogate prediction at each step of the ablation path. The percentage is calculated according to the performance spread between the source and target configurations.

## F.2 Reduced Configuration space Ablation Paths

In Figure 28, 29, and 30, we present the ablation paths of the configuration spaces reduced manually and by HyperShap.

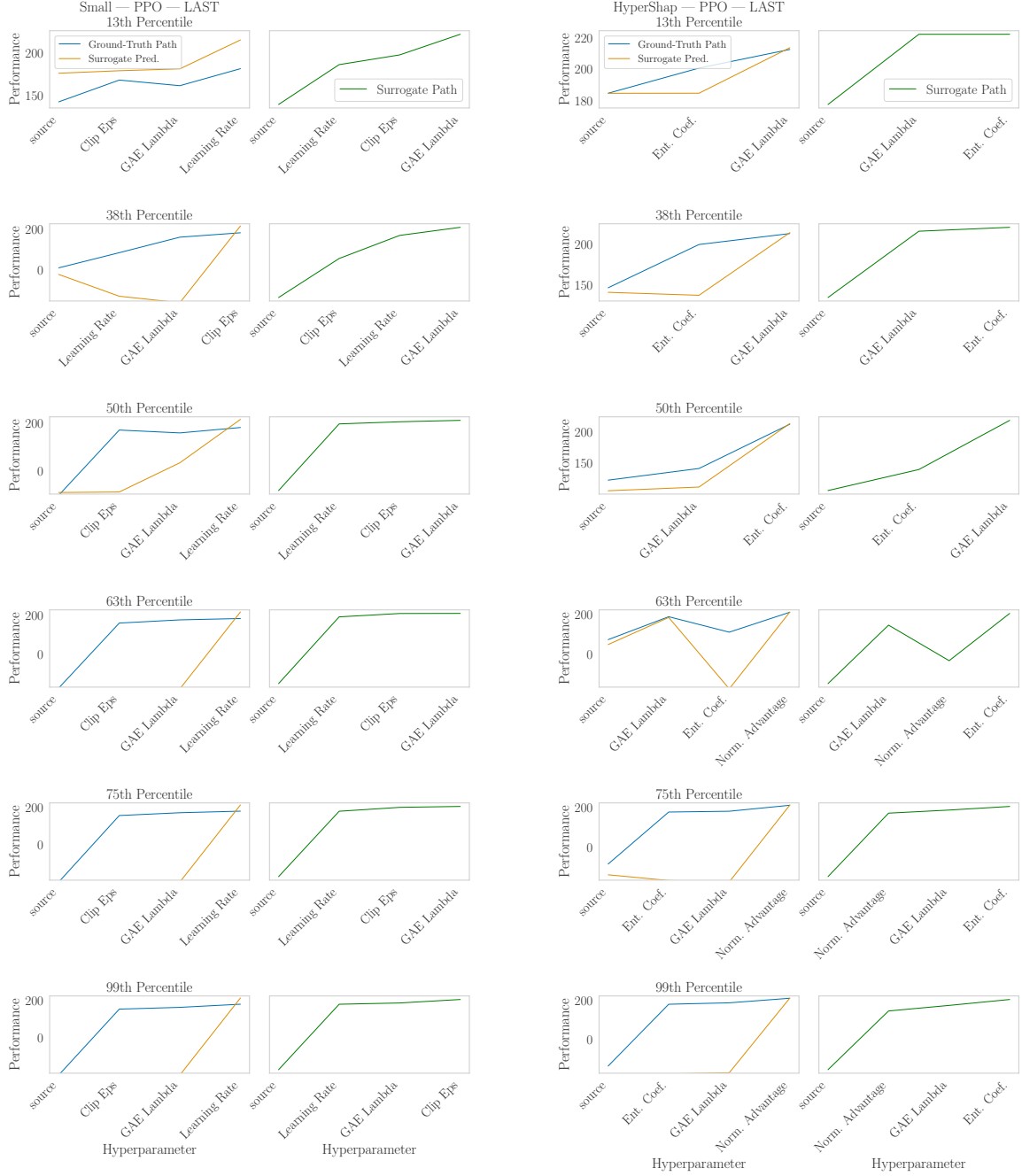

Figure 28: Ablation paths for the manually and HyperShap reduced configuration space for Lunar Lander with the following subplots. **Left:** The ground-truth ablation paths for the performance metric last depicted in blue, with the respective surrogate performance predictions for each hyperparameter in orange. **Right:** The resulting ablation path under the surrogate model predictions in green. For the path construction, the Lunar Lander surrogate model fitted on 2048 configurations was used.

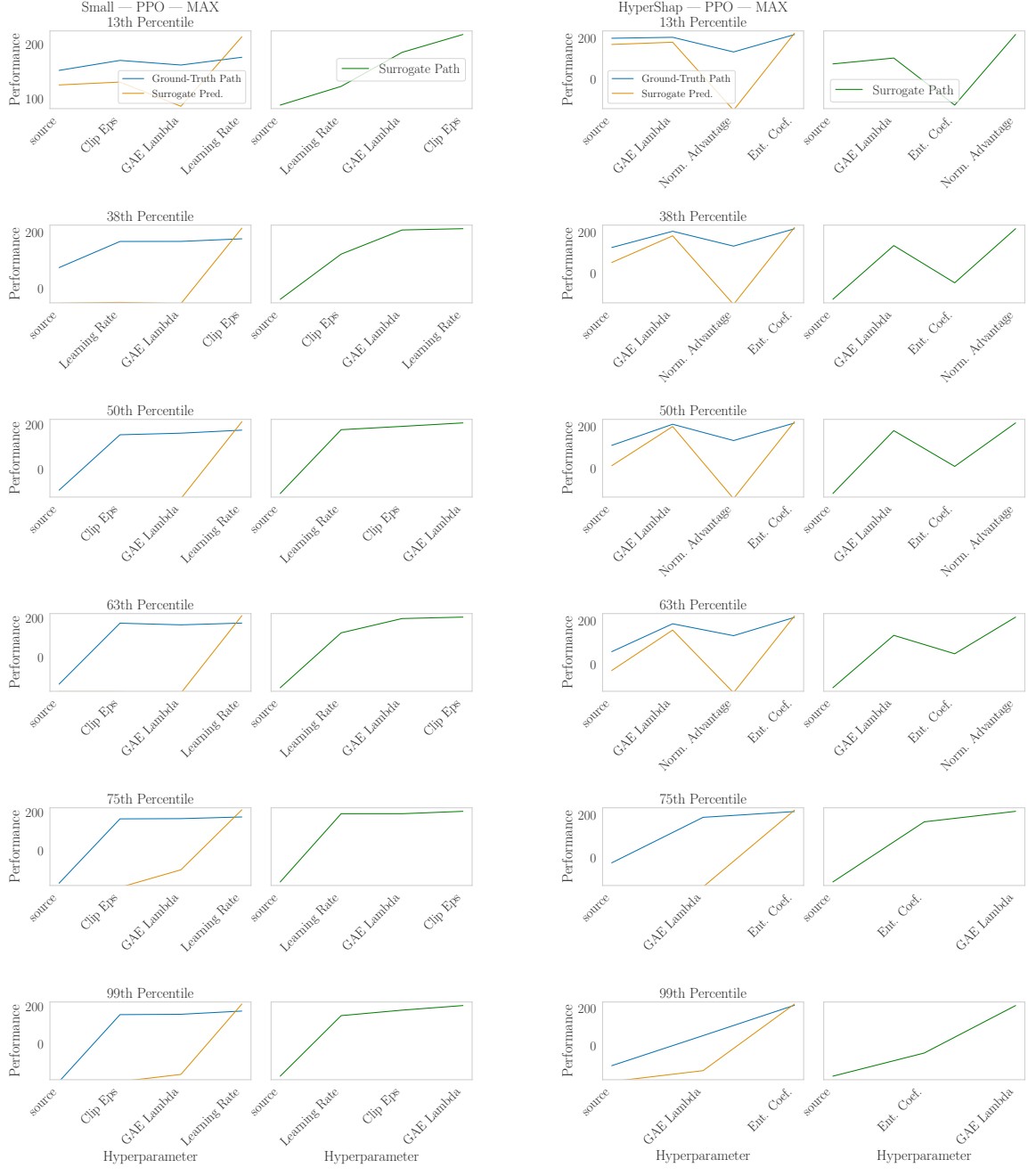

Figure 29: Ablation paths for the hand-crafted and HyperShap reduced configuration space for Lunar Lander with the following subplots. **Left:** The ground-truth ablation paths for the performance metric max depicted in blue, with the respective surrogate performance predictions for each hyperparameter in orange. **Right:** The resulting ablation path under the surrogate model predictions in green. For the path construction, the Lunar Lander surrogate model fitted on 2048 configurations was used.

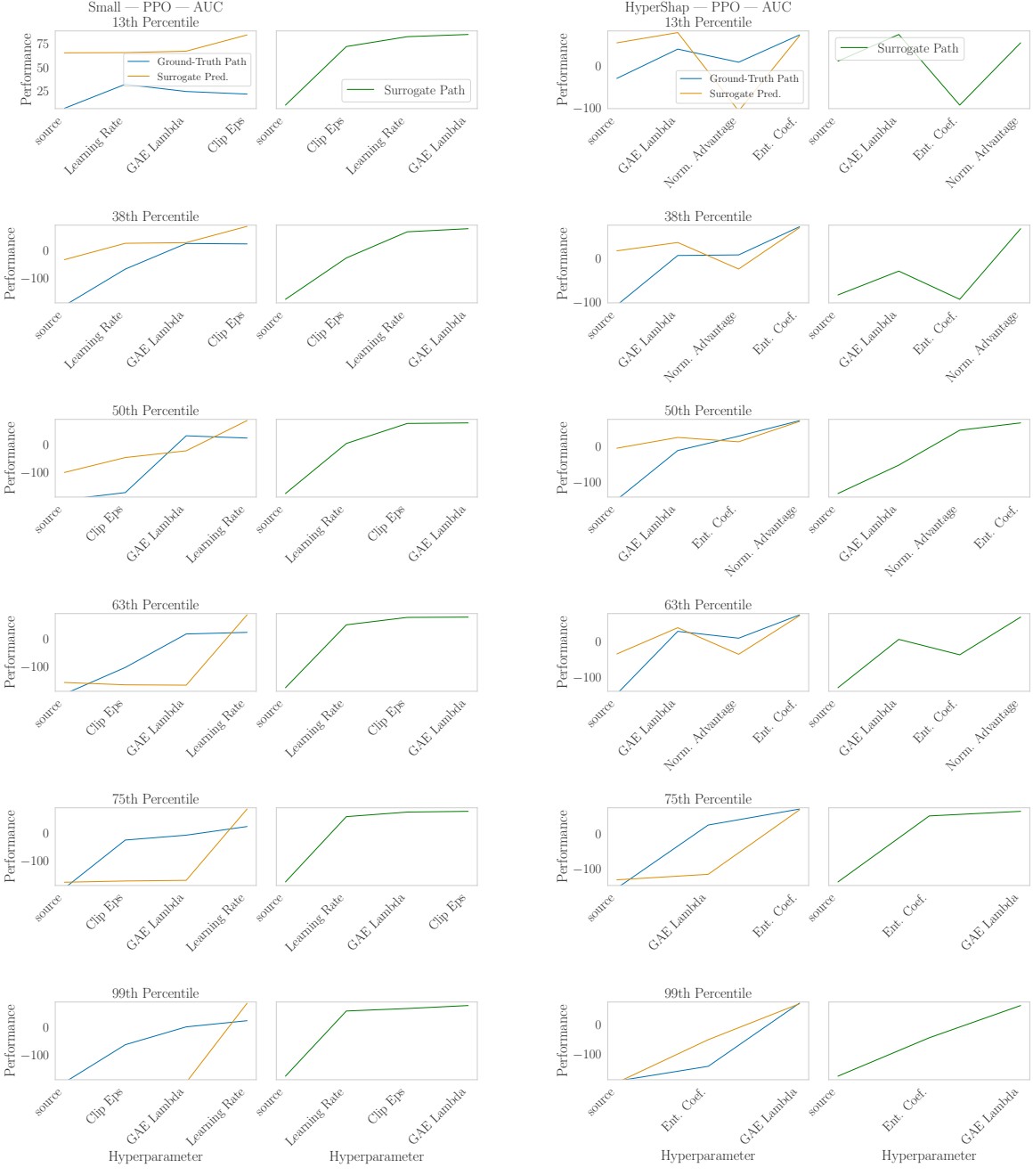

Figure 30: Ablation paths for the hand-crafted and HyperShap reduced configuration space for Lunar Lander with the following subplots. **Left:** The ground-truth ablation paths for the performance metric AUC depicted in blue, with the respective surrogate performance predictions for each hyperparameter in orange. **Right:** The resulting ablation path under the surrogate model predictions in green. For the path construction, the Lunar Lander surrogate model fitted on 2048 configurations was used.

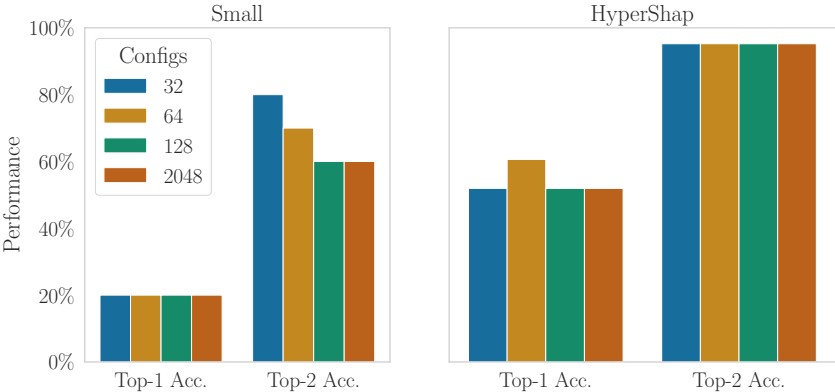

Figure 31: The accuracy of the surrogate model of predicting the most significant hyperparameter whenever a performance shift or more than 10% occurs in the ground-truth ablation paths.

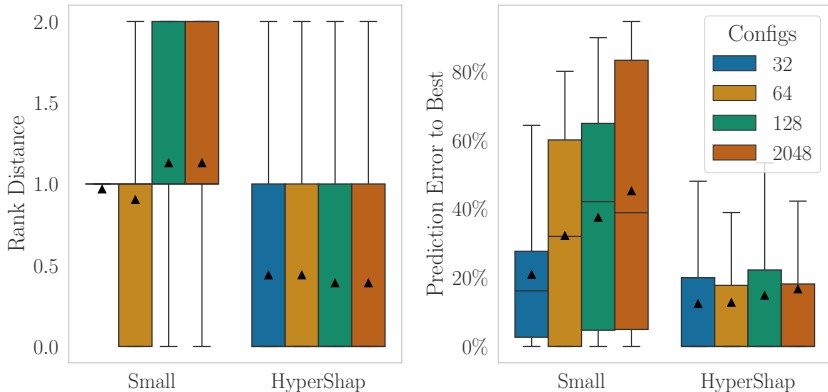

Figure 32: **Left:** The rank distance between the most important hyperparameter of the ground-truth and the top-ranked prediction of the surrogate model. **Right:** The difference between the performance of the most important hyperparameter and its surrogate prediction at each step of the ablation path. The percentage is calculated according to the performance spread between the source and target configurations.

Similarly to the full configuration space, we again observe significant differences in shape and ranking to the ground-truth hyperparameters. For a more detailed analysis, Figure 31 and 32 depict the accuracy in predicting significant changes in an ablation path and the margin of the prediction errors of the surrogates, respectively. Notably, there is a substantial difference in accuracy between the hand-crafted reduced configuration space and the HyperShap reduction. While the accuracy of the hand-crafted configuration space is around 20%, it rises to 50% for the HyperShap reduction and has perfect prediction for the top-2 configurations. This shows the potential importance of an informed reduction of the configuration space. However, we again observe that fitting the surrogate with more configurations does not result in improved ablation path accuracy. Therefore, even the reduced configuration spaces appear to require better configuration performance distributions to fit accurate surrogate models.

## G   Performance Spreads

The following Figures depict the performance distributions per task in our different benchmarks. All performance spreads are normalised between 0 and 1.

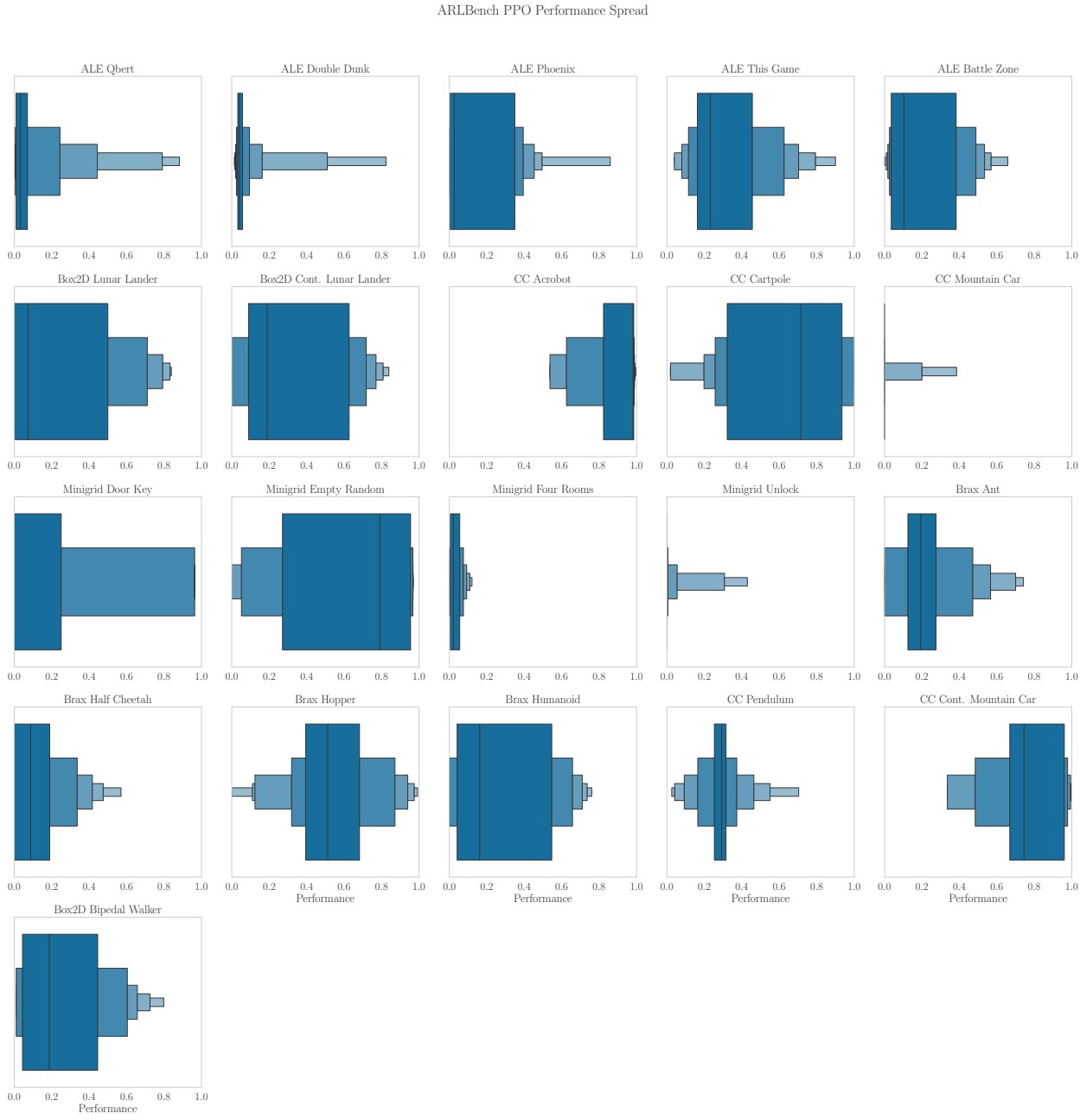

Figure 33: Performance spread of the ARLBench PPO environments.

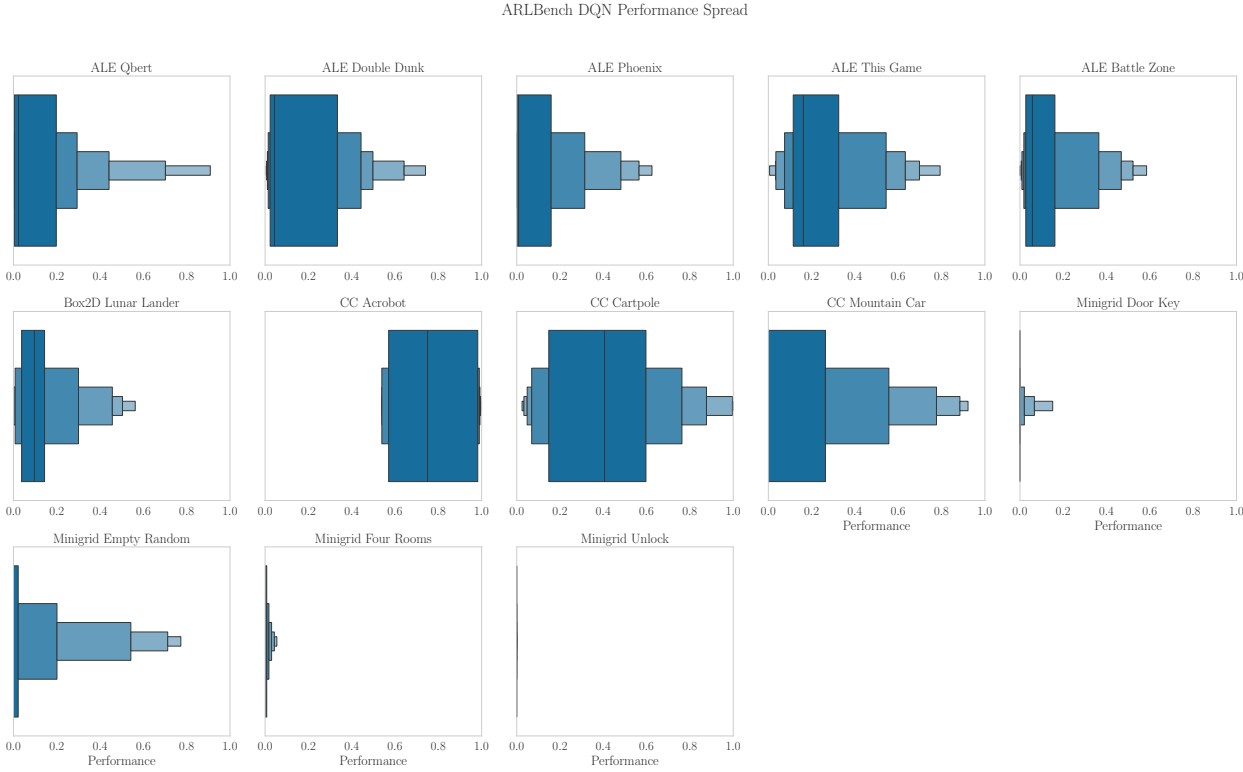

Figure 34: Performance spread of the ARLBench DQN environments.

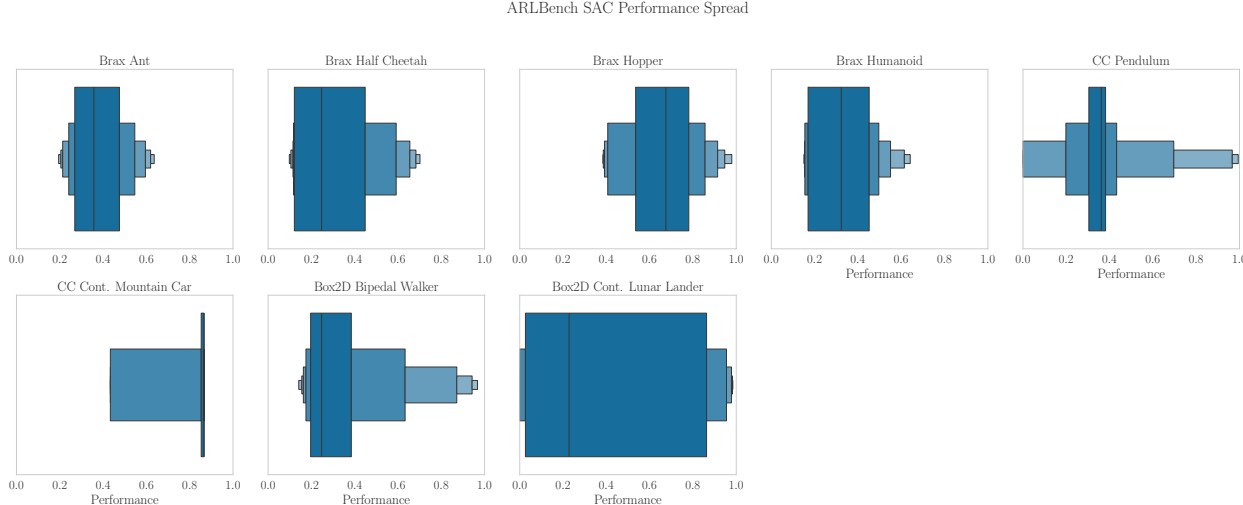

Figure 35: Performance spread of the ARLBench SAC environments.

Lunar Lander Performance Spread

Figure 36: Performance spread of the LunarLander 2048 environments.

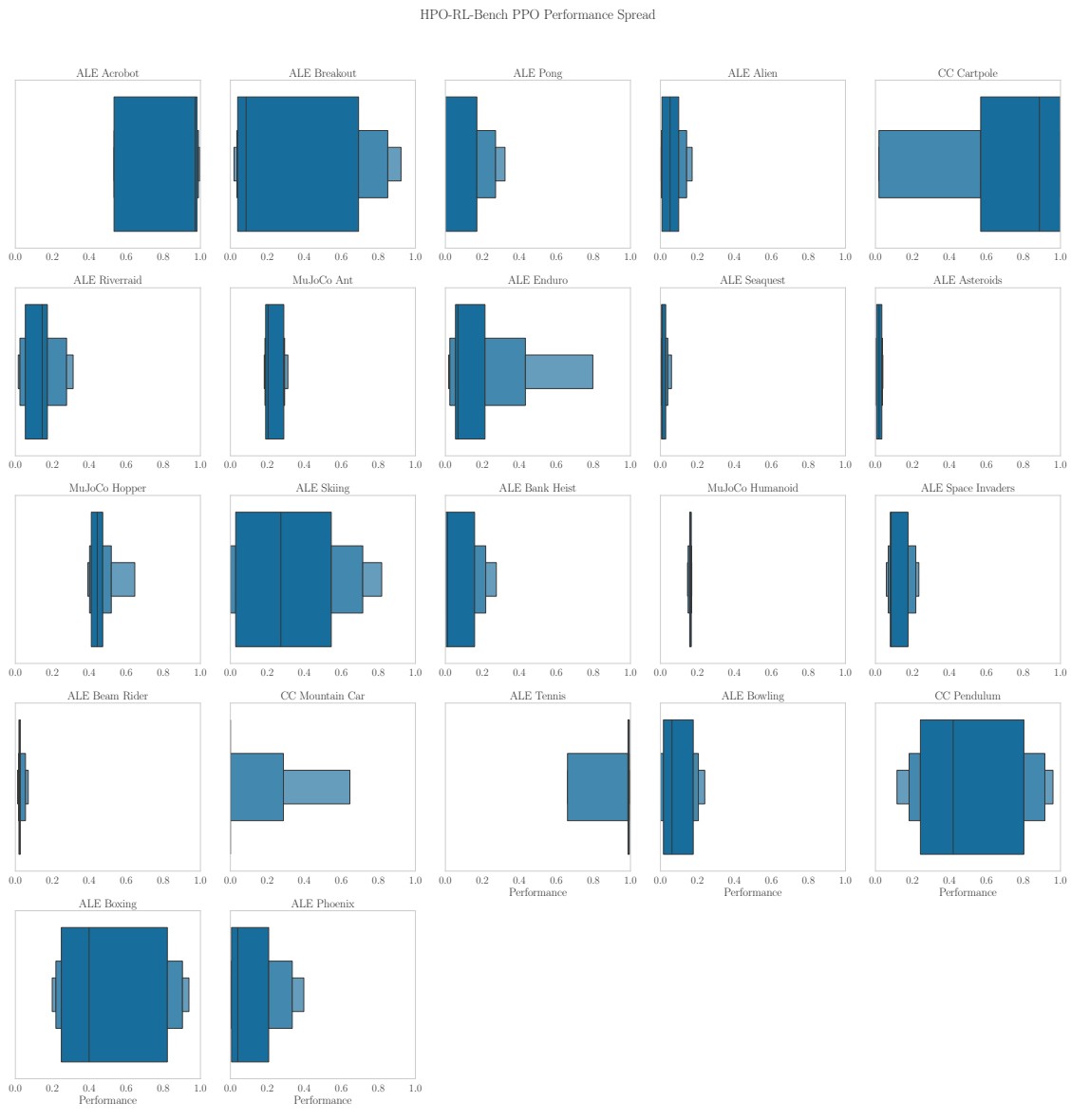

Figure 37: Performance spread of the HPO-RL-Bench PPO environments.

HPO-RL-Bench DQN Performance Spread

Figure 38: Performance spread of the HPO-RL-Bench DQN environments.

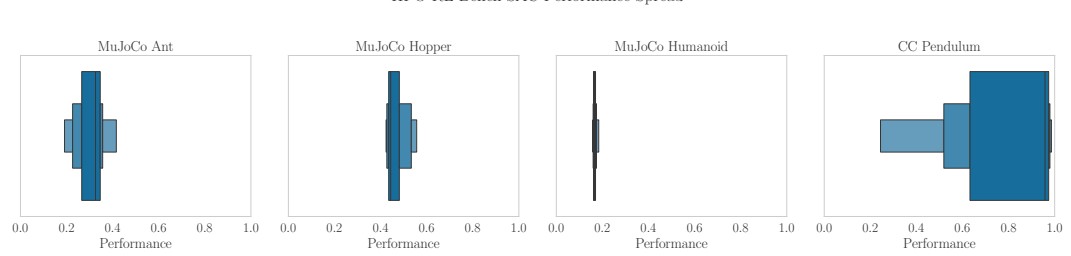

Figure 39: Performance spread of the HPO-RL-Bench SAC environments.

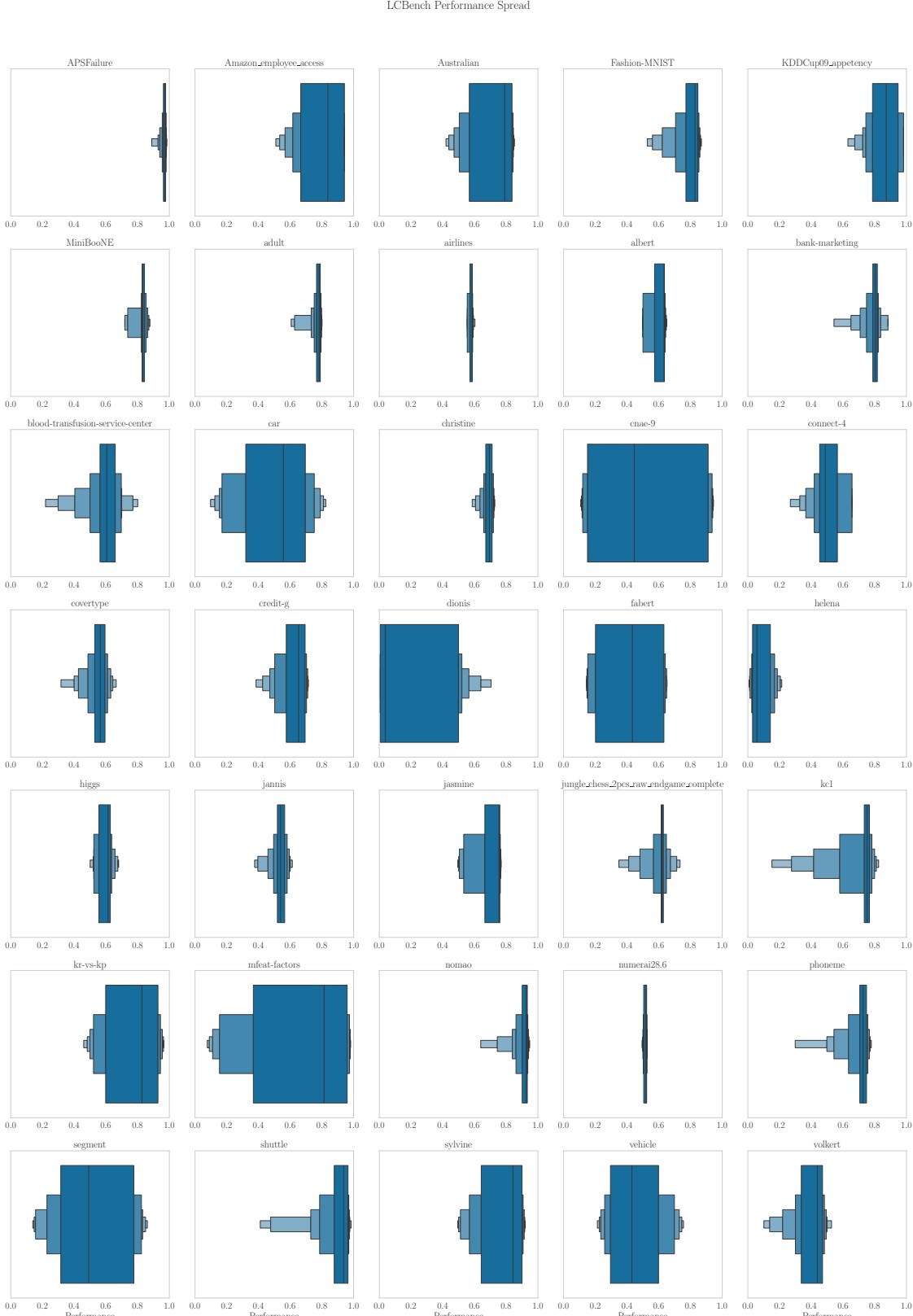

Figure 40: Performance spread of the LCBench environments.

Figure 41: Performance spread of the PD1 environments.

## H CI Scatter Plots

In Figure 42, 43, 44 and 45, we present the scatter plots between performance and confidence intervals over all the different tasks in ARLBench PPO, DQN and SAC and LCBench, respectively.

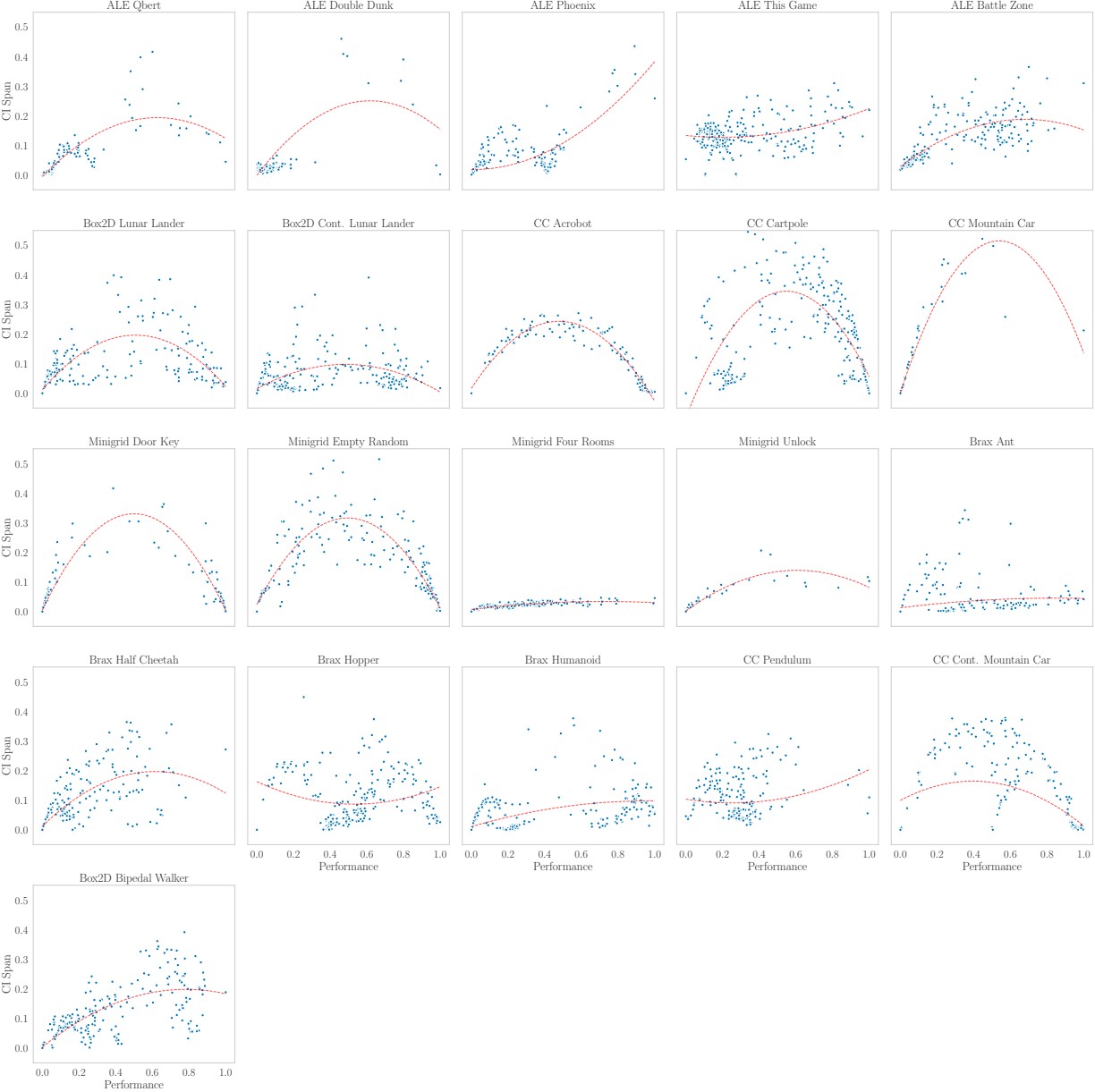

Figure 42: ARLBench PPO CI/Performance scatter plots. The red line shows the best fit of a polynomial of degree two.

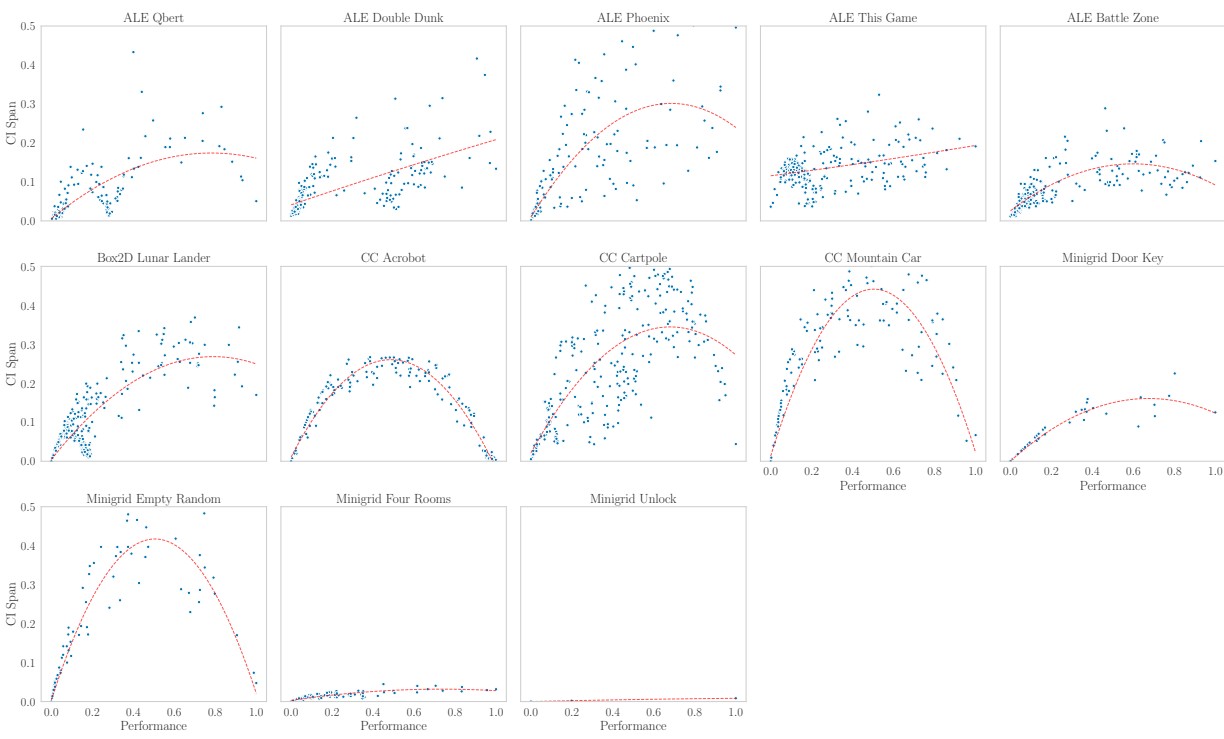

Figure 43: ARLBench DQN CI/Performance scatter plots. The red line shows the best fit of a polynomial of degree two.

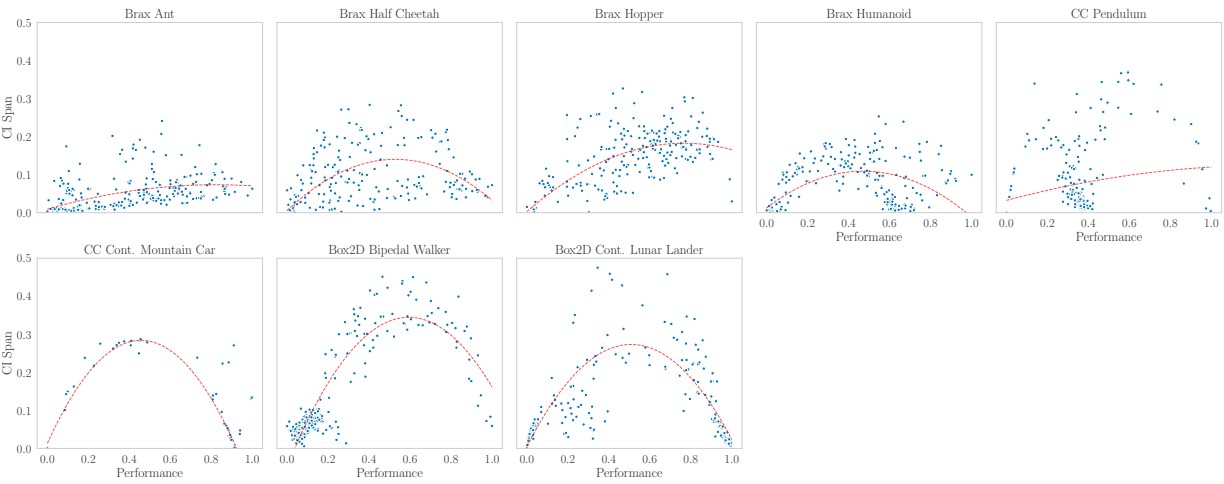

Figure 44: ARLBench SAC CI/Performance scatter plots. The red line shows the best fit of a polynomial of degree two.

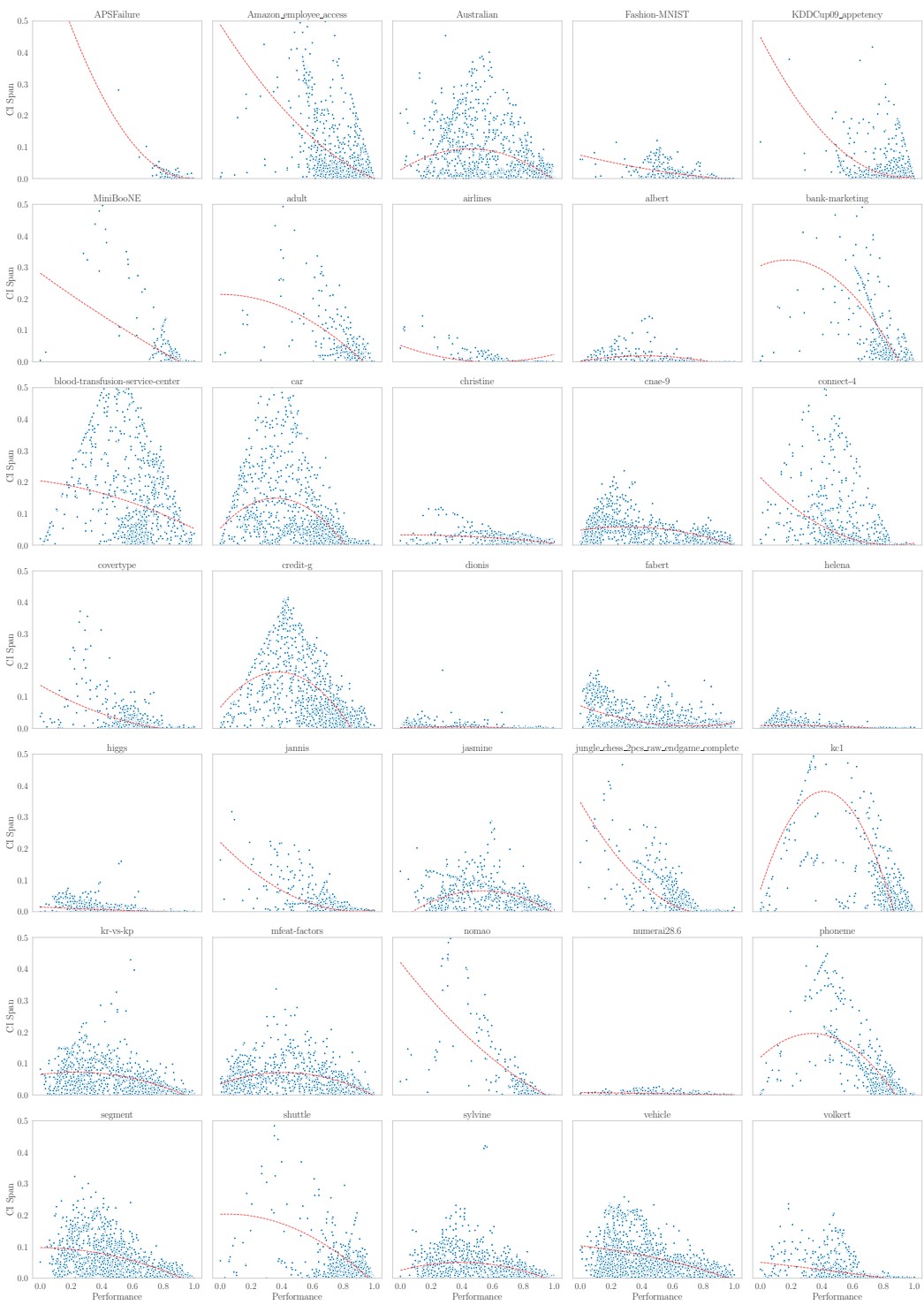

Figure 45: LCBench CI/Performance scatter plots. The red line shows the best fit of a polynomial of degree two.

## I    Landscape MDS Analysis

In Figure 46, 47, 48 and 49, we present a multidimensional scaling (MDS) analysis of the hyperparameter landscapes in ARLBench PPO, DQN and SAC and LCBench, respectively. MDS aims to preserve distances as accurately as possible when mapping from a high-dimensional space to a lower-dimensional one and is used similarly in the AutoML analysis visualisation tool DeepCAVE (Segel et al., 2025), LCBench shows an overall substantially more unimodal behaviour with clearly distinguishable regions between high-performing and low-performing regions. In contrast, the ARLBench landscapes show a strong overlap where low-performing configurations occur close to high-performing ones and no clear overall structure.

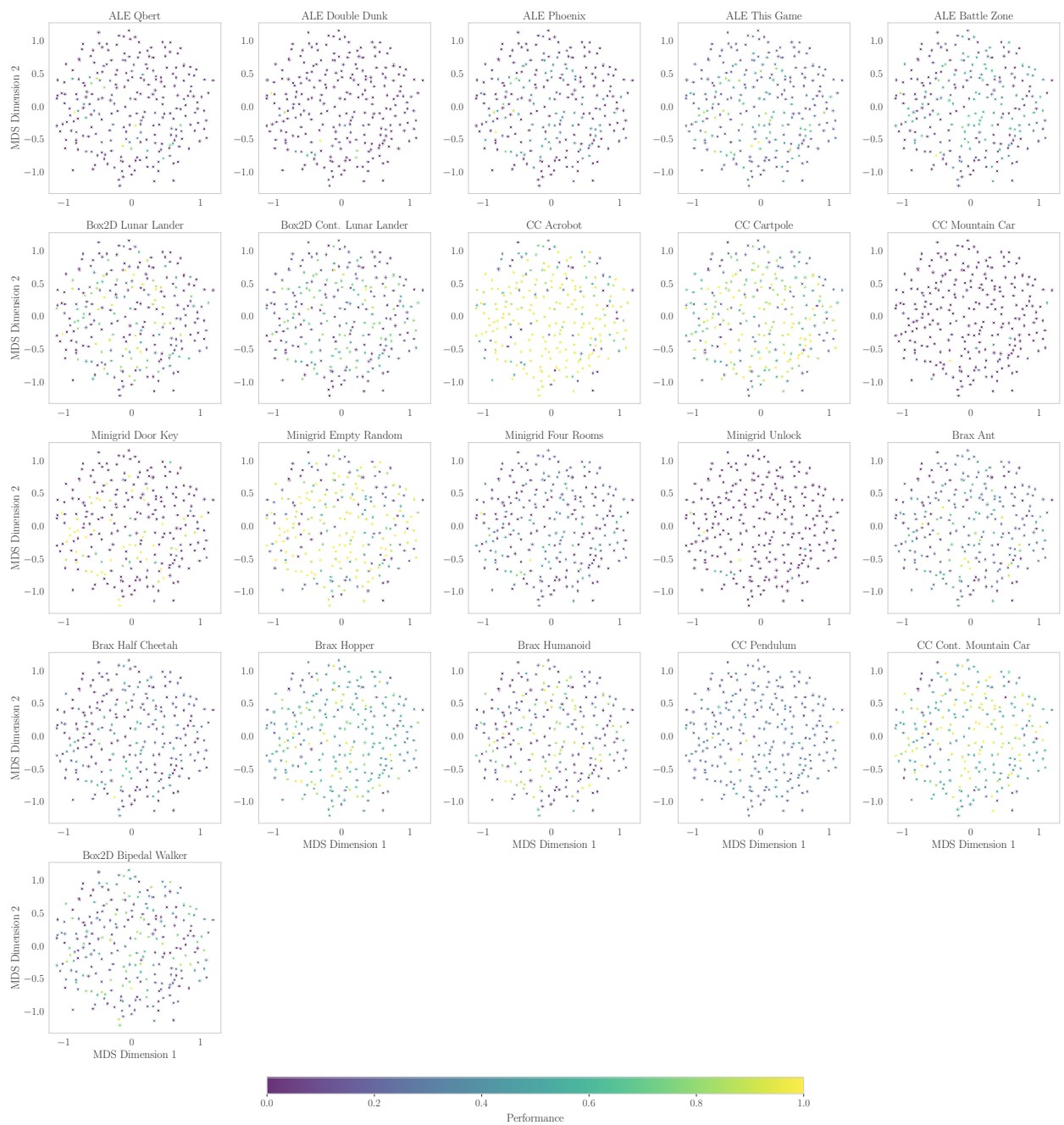

Figure 46: ARLBench PPO MDS analysis.

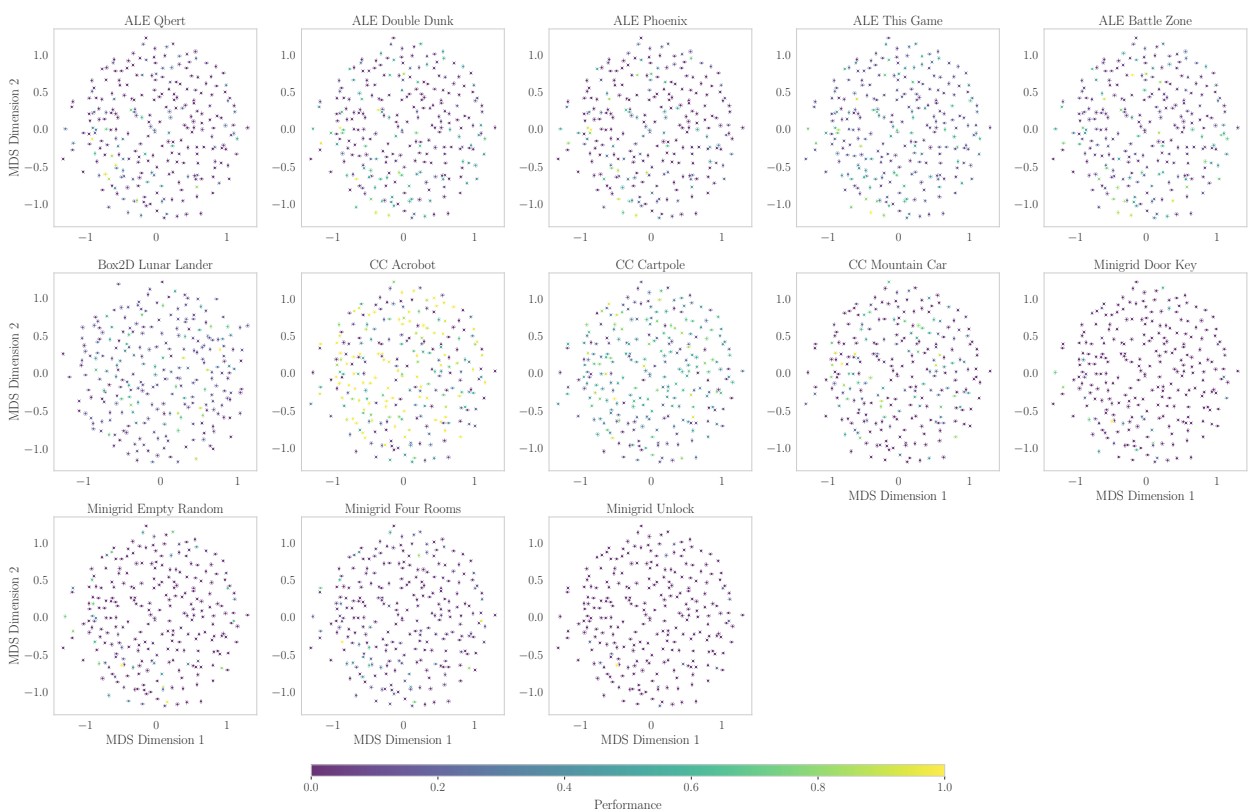

Figure 47: ARLBench DQN MDS analysis.

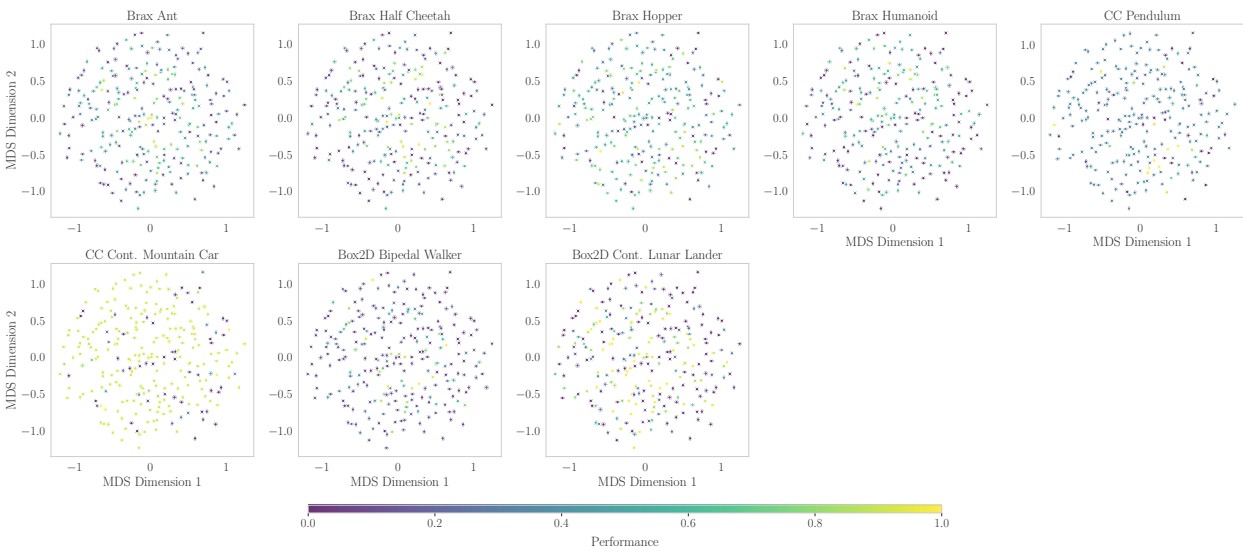

Figure 48: ARLBench SAC MDS analysis.

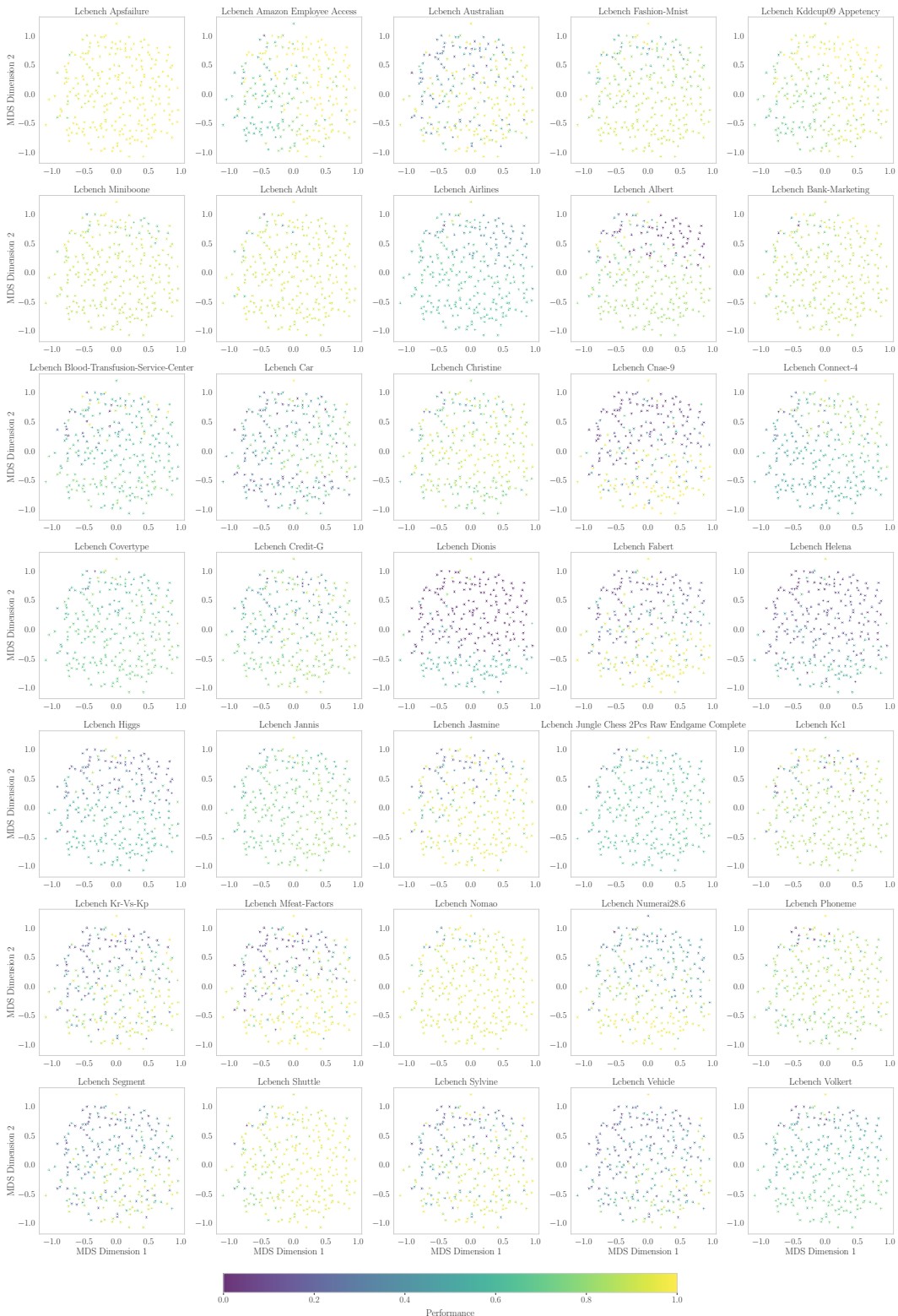

Figure 49: LCBench MDS analysis.

