# OpenReview forum: "Performance Prediction In Reinforcement Learning: The Good, The Bad And The Ugly"
_TMLR — Rejected by TMLR_

### Review · Reviewer_eCRf · 2026-04-23

**Summary Of Contributions:**

This paper consists of a thorough empirical investigation of models, called surrogates, which predict the performance of hyperparameters for existing reinforcement learning algorithms. Experiments demonstrate that surrogate models are for the most part inaccurate predictors of true hyperparameter performance. The paper then produces recommendations for practitioners about how to use surrogates, as well as an analysis as to why surrogate models perform worse in RL than in supervised domains. Overall this paper clearly demonstrates the challenges of surrogate models for RL with lots of empirical evidence, and also provides well supported analysis as to why this may be the case.


**Strengths**

1. The results about the poor performance of surrogate models are well supported by the provided experimental results.

2. The practical recommendations are very clear and useful for practitioners working in this space.

3. The questions about why surrogate models struggle are clearly defined and answered very strongly throughout the paper.

4. The analysis on unimodality is very compelling and links nicely to the surrogate performance results.

**Weaknesses**

Some of the results are not sufficiently clear, and missing important details. I list these specifically under requested changes because I feel they can likely be resolved with small changes.

**Audience:**

Yes

**Audience Explanation:**

I believe that RL researchers and practitioners may be interested in knowing that surrogate modeling for RL hyperparameter analysis is ill-advised. Existing literature has established that hyperparameters present a significant challenge in RL [1,2,3]. Surrogate models present a possible solution to this noted problem, and showing this will not work is useful for those interested in this area. Additionally RL practitioners, particularly those working in applied areas, may benefit from the recommendations and general advice to stay clear of building surrogate models as a way to bypass hyperparameter tuning.


[1] Adkins, J., Bowling, M., & White, A. (2024). A Method for Evaluating Hyperparameter Sensitivity in Reinforcement Learning. In Neural Information Processing Systems

[2] Patterson, A., Neumann, S., White, M., & White, A. (2024). Empirical design in reinforcement learning. In Journal of Machine Learning Research

[3] Theresa Eimer, Marius Lindauer, and Roberta Raileanu. 2023. Hyperparameters in reinforcement learning and how to tune them. In the
International Conference on Machine Learning.

**Broader Impact Concerns:**

Nothing about this work suggests a significant potential for harm. This work seeks to greater understanding of hyperparameters in RL which if anything would lead to less need for hyperparameter tuning which is an energy inefficient and arguably wasteful process.

**Claims And Evidence:**

Yes

**Claims Explanation:**

This paper makes three main claims each of which are backed by sufficient empirical evidence. For all these claims, multiple datasets, and multiple surrogate models are used for the experiments which makes the conclusions more likely to generalize to other settings. These claims are then synthesized into the recommendations for practitioners.

1. The first claim is that surrogate models deviate substantially from the ground truth, which is established by Figures 1 and 2.

2. The second main claim is that RL HPO landscapes and datasets contain more local optima, have worse performance distributions, and have higher variance than the same datasets/landscapes in supervised learning. Each part of this claim is well supported by the experiments in section 5.

3. The last claim is that the proposed practice of reducing the configuration space of existing RL algorithms via HyperSHAP will lead to better but still far from satisfactory performance for surrogate modelling. This is shown conclusively in figure 8, and table 1.

**Requested Changes:**

The requested changes for this work entirely concern clarity of the presented results. To me the most important two changes are to the clarify the following two points of confusion,
1. A statement explaining what is meant by top-K in each sub-section on page 6 would be beneficial.  To my understanding, under the heading “Local Hyperparameter Importance” top-K is used to refer to the surrogate's ability to predict the K most important hyperparameters. This would explain why the top-K accuracy decreases as K increases. On the same page under the heading "Ablation Paths” it is used, to my understanding, to refer to the accuracy of the top-K predictions of the model. I found this confusing and would appreciate a statement clarifying this matter.
2. In figure 3, it is unclear to me what LL refers to. Four benchmarks are introduced on page 3, and four datasets are used in figure1, it was therefore surprising to see 5 datasets in figure 3. I assume LL means Lunar Lander but this is not listed along with the other datasets, and is introduced only later in the work on page 10 for figure 6.

I have two additional less crucial requested changes which are minor points of clarification.

3. When reducing the configuration space, on page 11, it would be useful to briefly clarify what happens to the hyperparameters that are not included in the induced space. I assume they are set to the optimal value?

4. In Figure 4 it is challenging to visually delineate between the HPO-RL and PDI boxes without guessing based on the context from the analysis. I would appreciate another choice of colour or symbol to delineate these boxes.

---

> ### Author Response · Authors · 2026-05-28
>
> Dear Reviewer,
>
> Thank you for engaging with our work and for your assessment. Below, we would like to clarify the requested changes and explain how we plan to incorporate them into the revised version of our paper. We will publish our revised paper by June 12th at the latest.
>
> 1. >A statement explaining what is meant by top-K in each sub-section on page 6 would be beneficial. To my understanding, under the heading “Local Hyperparameter Importance” top-K is used to refer to the surrogate's ability to predict the K most important hyperparameters. This would explain why the top-K accuracy decreases as K increases. On the same page under the heading "Ablation Paths” it is used, to my understanding, to refer to the accuracy of the top-K predictions of the model. I found this confusing and would appreciate a statement clarifying this matter.
>
> The reviewer raises a valid point here, as the top-K accuracy is indeed defined differently for LPI and ablation paths. For LPI, it is defined as the number of true K most important hyperparameters that are included in the predicted top-K set, regardless of the correctness of their internal order.
>
> For ablation paths, the top-K accuracy of the surrogate is defined as the accuracy of the surrogate model of predicting the correct hyperparameter that is responsible for a shift. For example, for top-1 accuracy of ablation paths, predicting the correct hyperparameter. For top-3 accuracy, predicting a hyperparameter that is included in the set of the three hyperparameters causing the biggest performance shifts. Therefore, top-K accuracy for ablation paths is easier to achieve with larger K values than with LPI.
>
> We will add a clarification to the paper, clearly defining and distinguishing between the top-K consistency for LPI and top-K prediction accuracy for ablation paths.
>
> 2. >In figure 3, it is unclear to me what LL refers to. Four benchmarks are introduced on page 3, and four datasets are used in figure1, it was therefore surprising to see 5 datasets in figure 3. I assume LL means Lunar Lander but this is not listed along with the other datasets, and is introduced only later in the work on page 10 for figure 6.
>
> Lunar Lander 2048 (LL) was indeed not supposed to be contained yet in this analysis, because the Lunar Lander environment is already contained in ARLBench for 256 configurations and all three algorithms. We will remove it now from the figure and will define it in the (LL) acronym used in some of the later figures in the body text.
>
> 3. >When reducing the configuration space, on page 11, it would be useful to briefly clarify what happens to the hyperparameters that are not included in the induced space. I assume they are set to the optimal value?
>
> The hyperparameters not used in the reduced hyperparameter spaces are set to the default baseline values used in ARLBench for the LunarLander environment. We will make this explicit in the paper.
>
> 4. >In Figure 4 it is challenging to visually delineate between the HPO-RL and PDI boxes without guessing based on the context from the analysis. I would appreciate another choice of colour or symbol to delineate these boxes.
>
> Thank you for raising this point. We will adjust the plot to not only show barplots but also the individual data points. This will make the colours more clearly distinguishable and additionally adds more context to the plots.
>
> Again, we want to thank the reviewer for the effort in assessing our work and for their valuable feedback.

---

### Review · Reviewer_Tct9 · 2026-04-26

**Summary Of Contributions:**

This paper studies performance prediction of RL algorithms. While there is no formal definition of the problem at hand in the paper, my understanding of performance prediction for RL is the following: given an algorithm (e.g. PPO, DQN, ...), a hyperparameter configuration (e.g. learninig rate = 0.8, batch size =32, ...), and an MDP (e.g. Lunar Lander or Acrobot), predict the average performance of the trained agent.

Overall, the topic is well motivated and the experiments are interesting, but the paper lacks formalism which in my opinion did damage the contributions. We shall detail this later on.

The paper contains three main contributions.

In section 4, the authors do a comprehensive empirical study of RL performance prediction and compare it to supervised learning performance prediction (given a supervised learning task e.g. the titanic dataset, predict the performance of e.g. MLPs with 2 layers and learning rate = 0.2). They also study hyperparamter importance analyses in RL, i.e. identifying which hyperaprameters of an RL algorithms are important to look at to predict its performance. The author's conclusion is that performance prediction and hyperparameter importance analyses are hareder for RL than for SL.

In section 5, the authors study RL performance landscapes. They claim that the performance of an RL algorithm as a function of the hyperparameters is non-smooth with multiple local minima. This claim would help support the first contribtion as supervised learning performance landscapes are known to be smooth.

In section 6, the authors study the impact of 3 distinct tricks on RL performance prediction (more training runs, sample more hard to predict runs, vary important hyperparameters only). The authors claim that reducing the number of hyperparameters to vary helps performance prediction.

My overall impression of the contributions is that they are interesting but necessitate more work. Again , I will detail this later on, but I feel  like this paper is baseically two not quite yet finished papers. The first one could be written with sections 4 and 5 and answer the question: how hard is RL performance prediction and why is it hard ?   The second paper could be answering: how to make good predictions of RL performances?

**Audience:**

Yes

**Audience Explanation:**

This paper is of interest to both the RL and the HPO/AutoML communities. Both are major communities inside TMLR readers.

**Broader Impact Concerns:**

No concerns

**Claims And Evidence:**

No

**Claims Explanation:**

## General comment
First of all, I would like to command the amout of experiments in this paper and say that we can really feel the authors put a lot of work in this paper.
The key problem I have with the contributions is that **they are hard to interpret because the paper lacks formalism**.
What the authors are tackling is essentially a supervised learning problem where the training set is either HPO-RL or ARLBench, This leads the reader to ask natural questions that are left unanswered by the authors: are the RMSE presented in the paper obtained from a test or train set? This is essential as the whole premise of the paper is that if one has good RL performance predictors, one can save time in evaluation: no need to actually run the RL training we can predict it. Well, if the all the RMSE presented in the paper are from train set, then one cannot conclude anything about the potential use of RL performance prediction in practice. Indeed, if the data are in the train set, the RL training already happened: no time to to save. What would really be an extraordinary contribution, is to predict the performance of an RL training not done previously! All the above makes the case for a clear formalism of what is the problem at hand: how is RL performance defined? How are the metric values on figures 1-5-6-8 computed and what are the uncertainty presented? This would  help clarify later on what the authors mean by RL landscapes. I also believe it would simplify the hard-to-digest writing with many different terminologies 'surrogate model performances', 'RL performance predictions' , see the first 4 lines of section 4.

I understand that because the end goal of the authors is to improve HPO/AutoRL, the writing (terminology and experiments) of the paper is anchored in the HPO literature. However since no HPO or AutoRL experiments are included in the paper, I recommend writing and presenting the paper as what it really is: an interesting study of a specific supervised lerarning problem.

## Contrib 1: performance prediction (section 4)
- You mention LCBench RMSE from figure 1 but LCBench is absent from the figure.
- Why use Kendall Tau correlation rather than a statistical rank test with a clear hypothesis?
- Why is it surprising that RMSE top 10 <= RMSE for RL performance prediction?
- Why talk about interpretablility in section 4.2 rather than simple features importance?
- Why use Local Hyperparameter Importance rather than the straightforward random forest features importance since you already use RFs as your predictor?
- LunarLander 2048 is first mentioned in 4.2 but defined only later in the paper.
- When you say PPO and DQN have n hyperparams with a certain importance, it would be good if you tell us which ones.

## Contrib 2: RL performance landscape (section 5)
- The term RL landscape is very hard to understand.
- I think the term unimodal landscape is also a weird choice. I think it should be concave/convex since we are talking about a function values (RL performances) over a domain (hyperparams configs) rather than talking about distributions,
- In 5.1, similarly to the previous contrib, it is not clear to me why we would be intersted in having better predictions only for high RL performances.
- I found section 5.2 **very interesting**. It would have deserved more details and the graal would have been a surface plot of a performance landscape (fixing some hyperparamters and varying only e.g. the 2 most important ones, or do a PCA).
- In section 4 you had observaation boexes, why introduce challenges boxes?
- I think figure 7 is referenced before figures 5 and 6.
- I did not see (or did not understand) how challenge 3 is supported
## Contrib 3: tips for predicting RL performances (section 6)
Overall this section is more sound than the other 2 even though the results are restricted to the LL2048 dataset only.
- A sentence to explain how SMAC works would have been helpful.
- In which literature were $\lambda$ and $\epsilon$ were identified as important HPs for PPO? Why not re-use the most important HPs you found in section 4.2?
- I did not find Figure 42 :)
-

**Requested Changes:**

## Minor
- Please Include RL landscape plots
- ''obstained''
- Please answer the questions I asked in the previous sections.
- Please share your code asap
## Major
- Please give clear defnition of what is presented in the figures (uncertainty and train or test rmse)
- Please give clear definition of the RL performance prediction problem
- Please use better terminology or justify yours (RL landscape, surrogate models, unimodality...)

---

> ### Author Response · Authors · 2026-05-28
>
> Dear Reviewer,
>
> Thank you very much for engaging with our work so carefully and raising many important points. We believe that by addressing these, we will be able to substantially improve the paper. Please find below our clarifications on the questions you raised and on the requested changes. We will publish our revised paper by June 12th at the latest.
>
> We will first discuss the reviewer’s major concerns.
>
> ### 1. Clear definitions of what is presented in the figures.
>
> >This leads the reader to ask natural questions that are left unanswered by the authors: are the RMSE presented in the paper obtained from a test or train set?
>
> We agree that this should be clearly defined in the paper. We are reworking the section about the surrogate training to better explain how the surrogate models are tuned, selected and how the resulting test metrics are computed:
>
> All surrogate models were optimised via random search comprising 200 trials, with the exception of GPs, which were allocated 50 trials owing to their substantially smaller hyperparameter space.
> Each configuration was assessed using 10-fold cross-validation.
> For each fold, the data was partitioned into a 90% training set, a 5% validation set (utilised to identify the optimal configuration from the random search), and a 5% test set.
> The optimal configuration was determined by the maximum mean performance across all 10 validation folds.
> Subsequently, the generalisation capability of this selected configuration was computed as its mean performance across the 10 test folds when trained on the respective train folds.
> All surrogate performance metrics reported in our paper reflect these test set evaluations.
> To obtain an uncertainty value of the surrogate optimisation, random search was repeated with five different pseudo-random seeds, each yielding one surrogate model.
>
> Crucially, this means that all surrogate models have been evaluated in the use case that the reviewer proposes: using the surrogate models to evaluate RL configurations that have been unseen before.
>
> >How are the metric values on figures 1-5-6-8 computed and what are the uncertainty presented?
>
> As stated above, the metric values are computed on unconfounded test sets. For each of the figures, we will give additional context in our revised paper.
>
> **Figure 1:** Boxplots aggregating the test-set prediction performance of the RF surrogate models across all tasks, algorithms and surrogate model fits in each benchmark suite.
>
> **Figure 5:** Boxplots aggregating the test-set prediction performance of the RF surrogate models across all three algorithms and surrogate model fits in Lunar Lander 2048 landscapes.
>
> **Figure 6:** Boxplots aggregating the test-set prediction performance of the RF models across all three algorithms and surrogate model fits in the Lunar Lander 2048 landscapes.
>
> **Figure 8:** Boxplots aggregate performance of RF surrogate models for PPO across the 5 surrogate model fits trained with growing numbers of configurations. The LunarLander landscape is trained with PPO and given for the full configuration space and on reduced configuration spaces: expert reduced “LL SMALL” and HyperShap reduced “LL HS”.

---

> ### Author Response · Authors · 2026-05-28
>
> ### 2. Clear definition of RL performance prediction problem and (3.)  better RL terminology
>
> >I also believe it would simplify the hard-to-digest writing with many different terminologies 'surrogate model performances', 'RL performance predictions' , see the first 4 lines of section 4.
>
> We agree that the paper would benefit from improved formalism, especially to enhance readability for members of the RL community who are not necessarily familiar with concepts from AutoML. We will add the missing definitions similar to the following description to the revised paper.
>
> ---
>
> We formulate the task of *RL performance prediction* as a supervised regression problem.
>
> Given an environment $\mathcal{M}$, let $\mathcal{A}_\lambda$ denote an RL algorithm parametrised by a hyperparameter configuration $\lambda \in \Lambda$, where $\Lambda$ is the configuration space.
>
> The true performance of the algorithm $\mathcal{A}$ for an environment $\mathcal{M}$ is a function $f_{\mathcal{A},\mathcal{M}}: \Lambda \to \mathbb{R}$, which yields a scalar performance metric $p$,  for RL typically the expected undiscounted return over a set of random seeds.
>
> We refer to the mapping $f_{\mathcal{A},\mathcal{M}}: \Lambda \to \mathbb{R}$ as the *hyperparameter performance landscape* or short *hyperparameter landscape* for a fixed environment $\mathcal{M}$ and algorithm $\mathcal{A}$.
>
> Because evaluating $f_{\mathcal{A},\mathcal{M}}(\lambda)$ requires running the complete training process for multiple seeds, it is computationally expensive.
>
> A *surrogate model* $\hat{f}_{\mathcal{A},\mathcal{M}}: \Lambda \to \mathbb{R}$ is a computationally cheap approximation of the true performance function for a target algorithm and environment combination.
>
> ---
>
> >The term RL landscape is very hard to understand.
>
> This term is indeed used inconsistently in the paper. It is meant to refer to the RL hyperparameter performance landscape. This will be clarified in our revised version with the proper term (i.e. RL hyperparameter performance landscape or short RL hyperparameter landscape)
>
> >I think the term unimodal landscape is also a weird choice. / I think it should be concave/convex since we are talking about a function values
>
> The unimodality term is an already established term in the AutoML literature, as used in “AutoML Loss Landscapes” by Pushak et al. Convexity is defined for continuous functions; however, hyperparameter configuration spaces often consist of discrete or categorical hyperparameters (e.g. batch-size or activation functions), making convexity not applicable.

---

> ### Author Response · Authors · 2026-05-28
>
> Additionally, the reviewer commented on multiple minor issues, which we will address in the following.
>
> ### Section 4:
>
> >You mention LCBench RMSE from figure 1 but LCBench is absent from the figure.
>
> It is correct that the body text of the paper used LCBench instead of LCBench 256. We will correct this in the revised paper.
>
> >Why use Kendall Tau correlation rather than a statistical rank test with a clear hypothesis?
>
> We used Kendall's tau because our objective is to measure rank correlation (i.e. how well the surrogate preserves the order of configurations), whereas statistical rank tests typically evaluate the significance of differences between medians.
>
> Kendall tau offers a direct probabilistic interpretation, which is the probability that a randomly selected pair of configurations is ranked incorrectly. This is the standard metric in AutoML literature (e.g., NAS-Bench-301) for assessing surrogate utility in downstream optimisation.
>
> >Why is it surprising that RMSE top 10 <= RMSE for RL performance prediction?
>
> We considered this again and reached the conclusion that we can not really make a good argument why top 10 <= RMSE is surprising without already making non-trivial assumptions about the landscapes (e.g. the landscaping having many configurations with close to 0 performance), which are examined only later in the paper. We will therefore remove the statement.
>
> >Why talk about interpretablility in section 4.2 rather than simple features importance?
>
> We use the term interpretability rather than simple feature importance because hyperparameter importance metrics and other surrogate based analysis methods in the AutoML literature aim to explain the complex, non-linear mechanisms of the hyperparameter landscape, rather than just ranking variables. Therefore, we believe that interpretability captures the holistic use case of surrogate models better.
>
> >Why use Local Hyperparameter Importance rather than the straightforward random forest features importance since you already use RFs as your predictor?
>
> Random Forest feature importance is a global metric that aggregates variance across the entire dataset. For such a metric it is infeasible to compute ground-truth values with a justifiable computational effort. We do not see an effective approach on how to obtain the ground-truth values except simply collecting more empirical data. For the same reason we did not choose to examine the global hyperparameter metric fANOVA in our work. For local metrics like ablation paths and LPI, it is straight-forward to compute ground-truth values that can be compared to the surrogate model’s performance.
>
> >LunarLander 2048 is first mentioned in 4.2 but defined only later in the paper.
>
> This is a good observation, we will fix this in the revised paper.
>
> >When you say PPO and DQN have n hyperparams with a certain importance, it would be good if you tell us which ones.
>
> We are presenting the LPI importance plots in the Appendix E of the paper from which the important hyperparameters can be inferred. However, the important hyperparameters are not consistent across environments and can vary substantially. This is consistent with related literature, for example “Reinforcement Learning Hyperparameters and How to Tune Them” by Eimer et al. We will include a better reference to the importance plots in our revised paper.

---

> ### Author Response · Authors · 2026-05-28
>
> ### Section 5:
>
> >The term RL landscape is very hard to understand.
>
> We agree. As noted above the term was inconsistently used. We will revise this in the updated paper.
>
> >I think the term unimodal landscape is also a weird choice. I think it should be concave/convex since we are talking about a function values (RL performances) over a domain (hyperparams configs) rather than talking about distributions,
>
> Convexity is not defined for discrete or categorical hyperparameters. To stay consistent with the related literature we use the term unimodality instead.
>
> >In 5.1, similarly to the previous contrib, it is not clear to me why we would be interested in having better predictions only for high RL performances.
>
> Hyperparameter importance methods and landscape analysis methods are very often applied to high-performing configurations or regions in the hyperparameter landscape. Therefore, to ensure accuracy results, the surrogate models need to be accurate in these regions. This is particularly important, as we have observed in Section 5.1, that the RL hyperparameter landscapes are bottom heavy (i.e. having substantially more configurations with close to 0 performance than high performance) and thus surrogate models have less training data for accurately predicting high-performing configurations correctly. We are of course not interested in having better performance only for high-performing RL configurations, but ensuring high performance on high-performing configurations is crucial for the downstream applications of surrogate models
>
> >I found section 5.2 very interesting. It would have deserved more details and the graal would have been a surface plot of a performance landscape (fixing some hyperparamters and varying only e.g. the 2 most important ones, or do a PCA).
>
> Thank you! We also believe it is an interesting analysis. We have generated multidimensional scaling (MDS) plots of the ARLBench and LCBench landscapes and will include them in the Appendix of the revised papers and refer to them properly in the body text. MDS is chosen as it aims to preserve distances as accurately as possible when mapping from a high-dimensional space to a lower-dimensional one and is used similarly in the AutoML analysis visualisation tool DeepCAVE Segel et al.
> Overall, the MDS plots support our analysis. While ARLBench shows substantially more erratic behaviour with low- and high-performing configurations closely clustered, LCBench shows more continuous performance changes with clear well-performing regions.
>
> >In section 4 you had observaation boexes, why introduce challenges boxes?
>
> This is linked to the story we want to tell with the paper. First, we check the performance of surrogate models and observe their performance specifically compared to supervised learning. Next, after observing that surrogate models struggle for RL landscapes, we examine the RL hyperparameter landscapes in-depth and conclude the specific “challenges” that RL hyperparameter landscapes yield.
>
> >I think figure 7 is referenced before figures 5 and 6.
>
> We will fix this in the revised version of our paper.
>
> >I did not see (or did not understand) how challenge 3 is supported
>
> Challenge 3 is mainly supported by our results in Figure 7 and our test on heteroscedasticity. This is, “Performance variation between multiple runs of the same configuration of a given RL algorithm can be substantial”. How substantial is depicted in Figure 7 (left), where the span of the confidence intervals (upper confidence bound - lower confidence bound) is large even when training with 30 random seeds.
> “... and is difficult to model” is shown by our test on heteroscedasticity, showing that different configurations have different variance, making it difficult to assume a uniform noise distribution or model the performance variation using standard probabilistic surrogates without explicitly fitting the variance on a per-configuration basis.
>
> “Predicting a single performance value for a set of diverse runs can be exceptionally difficult”.Figure 7 (left) illustrates this behaviour through the wider dispersion of the boxplots compared to LCBench. Although the CIs remain narrow for select configurations, they are substantial for others, exceeding 10% in at least one-quarter of the cases, even when using 30 seeds.
>
> We will make this connection more precise in our revised paper.

---

> ### Author Response · Authors · 2026-05-28
>
> ### Section 6:
>
> >A sentence to explain how SMAC works would have been helpful.
>
> SMAC is a dedicated hyperparameter optimisation algorithm that employs Bayesian optimisation to sequentially select promising configurations based on prior evaluations.
>
> We will add a sentence similar to the one above to the revised paper.
>
> >In which literature were lambda and epsilon identified as important HPs for PPO? Why not re-use the most important HPs you found in section 4.2?
>
> In “On the consistency of hyper-parameter selection in value-based deep reinforcement learning” by Obando-Ceron et al. the update horizon was identified as one of the most important hyperparameters. In “Hyperparameters in Reinforcement Learning and How to Tune Them” by Eimer et al., the clipping epsilon was identified as a hyperparameter that can be a decisive factor between learning success or failure.
>
> We can not use the import hyperparameter in Section 4.2 because they are based on local hyperparameter importance scores, which differ from configuration to configuration in the landscape. HyperShap instead infers hyperparameter scores for the full landscape.
>
> >I did not find Figure 42 :)
>
> Figure 42 is given in Appendix H of the paper. We will make this reference more explicit in the revised paper.
>
> We again want to thank the reviewer for all their helpful feedback and their effort in improving our work.

---

> ### Author Response · Authors · 2026-05-28
>
> Our code can be found in the following anonymous code repository: https://anonymous.4open.science/r/arlbench_surrogates_tmlr-1AF8/README.md

---

### Review · Reviewer_WVaE · 2026-05-14

**Summary Of Contributions:**

This paper investigates whether surrogate models, can be applied to reinforcement learning. The authors train four standard surrogate model classes (random forests, Gaussian processes, support vector machines, XGBoost) on two RL hyperparameter optimisation benchmarks (ARLBench and HPO-RL-Bench) and contrast their predictive quality with surrogates trained on two supervised-learning benchmarks (LCBench and PD1). The headline empirical finding is that the RL surrogates are noticeably weaker, particularly on top-performing configurations, and that this degradation propagates into surrogate-based hyperparameter-importance methods (Local Parameter Importance and ablation paths), whose rankings only match the ground truth at a high level.

The paper then characterises three properties of RL hyperparameter landscapes intended to explain the observed degradation: empirical performance distributions are skewed toward low-performing configurations (Challenge 1), reachability ratios indicate higher multimodality than corresponding supervised-learning landscapes (Challenge 2), and run-to-run variability is large and heteroscedastic (Challenge 3). Finally, the authors propose a HyperShap-based automatic configuration-space reduction method, evaluate it together with data-scaling and warm-start interventions on the LunarLander environment, and conclude with a set of practical recommendations for surrogate-model use in RL.

The paper's strengths. The motivating question is important: AutoML surrogates are increasingly applied to RL settings, and there is a genuine gap in the literature regarding their fidelity in this domain. The HyperShap-based configuration-space reduction is a sensible methodological contribution.

The paper's weaknesses. Several of the paper's substantive claims are not supported by the figures they reference: Observation 1 and Observation 2 are presented as findings about "RL" while the underlying data shows the pattern in only one of the two RL benchmarks, and the supervised-learning controls show comparable behaviour. The interventions in section 6 (more data, more seeds, full learning curves, SMAC-warmstart, HyperShap-based reduction) are tested only on LunarLander, which the authors themselves chose because it was the environment where their surrogates already worked best.   The empirical methodology is unclear. Several = terms ("intervals," "covered performance range," "FLC," "appropriately chosen bounds," the surrogate-tuning protocol) are undefined, making the figures hard to interpret. Finally, the section 7 recommendations exhibit a noticeable tonal whiplash between a confident four-step "Surrogate Benchmarking" workflow and an immediately following italicised recommendation *not to rely* on surrogate-based insights, both based on the same single-environment evidence.

**Additional Comments:**

The reviewer's overall assessment is that the paper's premise is interesting and the topic is one that the AutoML and AutoRL communities need a systematic empirical investigation of. The reviewer agrees with the authors' high-level intuition that surrogate models built for AutoML hyperparameter optimisation are unlikely to transfer cleanly to RL HPO data. The reservations expressed in this review are about the gap between the paper's claims and the evidence presented,  not about the value of the question or the broad direction of the conclusions. The reviewer would be receptive to revising this assessment upward following a major revision that rescopes claims to match the evidence and greater clarity on the empirical setup.  The reviewer thanks the authors for  engaging with a topic where careful empirical analysis is much needed.

**Audience:**

Yes

**Audience Explanation:**

There is a clear audience for this work.  The AutoML/HPO research community, the AutoRL community, and practitioners building benchmarks for RL hyperparameter optimisation would all benefit from a systematic empirical study of surrogate fidelity on RL landscapes. The motivating observation that AutoML surrogate-modelling techniques have not been validated on RL HPO data before being assumed transferable  is a real and important point, and one that the community needs to engage with regardless of the specific conclusions one draws from the experiments. The HyperShap-based configuration-space reduction method, the landscape-property analyses, and the heteroscedasticity findings each provide pieces that would be valuable to readers in this area even if the paper's takeaway claims are recalibrated.

**Broader Impact Concerns:**

No broader-impact concerns. The work studies methodology for automated reinforcement learning research and does not raise additional ethical considerations.

**Claims And Evidence:**

No

**Claims Explanation:**

The paper has the right instinct about its topic but the evidence base does not support the strength of its claims. Three patterns drive this assessment.

** The "Observations" generalise from a subset of the data.** The paper distils its section 4 findings into three boxed Observations. Two of these are unsupported by the underlying figures.

Observation 1 states that performance prediction for RL is harder than for the simple supervised-learning tasks commonly found in surrogate benchmarks and "comparable to deep learning problems, e.g. from computer vision." Reading the table in Figure 1: HPO-RL-Bench (RL) achieves a Kendall τ of 0.77 and RMSE of 0.05 — better RMSE than both supervised-learning benchmarks, and Kendall τ essentially tied with PD1 (the best SL benchmark). PD1 (SL) has Kendall τ of 0.78 and RMSE of 0.13 — worse RMSE than HPO-RL-Bench (RL) and ARLBench (RL). The within-category variation between ARLBench and HPO-RL-Bench, and between LCBench and PD1, is at least as large as the between-category variation between RL and SL. No statistical test accompanies the box-plots in Figure 1; the conclusion is drawn from overlapping distributions without inferential support.

Observation 2 states that prediction quality for RL is lower for better-performing configurations. The supporting prose immediately admits the pattern holds only for ARLBench: "HPO-RL-Bench shows similar RMSE on both sets, while the ARLBench surrogates look significantly worse." The Observation as boxed is therefore a description of ARLBench's behaviour, not RL's. Additionally, PD1 (SL) shows comparable RMSE degradation on top-10% configurations (0.13 → 0.15) — undermining the supervised learning versus-reinforcement learning framing the Observation invokes.

**The "Improving Surrogate Quality" experiments in Section6 use a confounded test environment.** The authors choose LunarLander for all data-scaling, FLC, SMAC-warmstart, and HyperShap-reduction experiments. The stated reasons for this choice:  "showed the best surrogate scores with respect to the PPO algorithm out of all environments" and "the confidence intervals between runs narrow as performance increases, making Lunar Lander particularly stable" are precisely the conditions that minimise the space for increasing the number of seeds to produce a visible improvement. I would have been much more interested to see if the "More Data' intervention helped on the environments where surrogate performance was bad. The paper then reports across Section6.1–Section6.3 that more data, more seeds, and full learning curves provide only modest improvements, and uses this as evidence that data-collection interventions are insufficient. This conclusion is difficult to assess  given the choice of test environment. The high-CI environments contributing the bad outliers in Figure 1 (with Kendall τ down to 0.3 for ARLBench) are exactly where more seeds should help, and they are not tested. A single-environment evidence base also cannot support the strong workflow recommendation in section 7 or the strong "do not rely on surrogates for insights" recommendation in section 7.

**Compounding clarity issues.** Several pieces of methodology are insufficiently specified for the figures to be evaluated on their empirical merits. The section 3.2 protocol paragraph describes 200 random-search trials, 10-fold cross-validation, and 5 refits per model without saying whether the hyper-hyperparameter search uses the same CV folds as the final reported metric (which would constitute selection-on-test) or whether nested cross-validation is used. The "appropriately chosen bounds" used to normalise performance to [0,1] in section 3.1 are never specified per task; this matters because Figure 3's "covered performance range" — itself an undefined quantity ("relative coverage of the theoretical performance bounds per task") — depends entirely on the choice of normalisation. The "interval" parameter in the Section5.2 reachability analysis is given as ranging from 0% to 10% "of the performance range of each point," without specifying whether is an asymmetric downward dip threshold, or a two-sided constraint on each step. "FLC" in Figure 6 is never expanded in the body text, and the surrogate's mechanism for consuming learning-curve data  is not described.

**Requested Changes:**

**1. Rescope the central claims to match the evidence.** Observations 1 and 2 should be rewritten so that the scope of each Observation matches the data that supports it.



**2. Test the section 6 interventions on at least one environment where surrogates currently fail.** The LunarLander-only evidence base cannot support the conclusions about whether data scaling, learning-curve data, SMAC-warmstart, or HyperShap-based reduction "help" in RL. Choose at least one environment from the high-CI / high-RMSE outliers in Figure 1 (for instance, the ARLBench environments contributing Kendall τ values near 0.3) and re-run the key Section6 interventions there, even at smaller scale than the 2048-configuration LunarLander dataset. The current conclusion should be supported by these results.

**3. Improve methodological specification for the central figures.** The following definitions and protocol details must appear in the body text (not only in figure captions):

- Section 3.1: The specific bounds used for [0,1] performance normalisation, per task or per environment, with justification. For RL environments where a theoretical maximum return is often ill-defined, state the choice (empirical max across all algorithms? random-policy baseline as lower bound? expert/human baseline?).
- Section 3.2: Explanation of the surrogate tuning protocol. State whether the 200-trial random search over surrogate hyperparameters uses the same 10-fold cross-validation folds as the reported final metric, or whether nested cross-validation is used. State explicitly when the 5 refits occur in the pipeline.
- Section4.1 / Figure 1: State what each box-plot point represents (presumably one score per (algorithm, environment) task within a benchmark). Add visual differentiation between RL and SL benchmarks in the legend. Justify the choice to subsample RL data to 3 seeds for comparability with LCBench, and show whether the RL/SL gap shrinks when all 10 RL seeds are used.
- Section 4.2 / Figures 2 and 3: For Figure 2, state what each violin distributes over (the three source configurations? the five refits? both? per-environment averages?) and how ground-truth LPI was computed (where do the grid evaluation points around each source configuration come from, and how many additional rollouts does this require). For Figure 3, write the explicit formula for "covered performance range" and clarify how  aggregation was done.
- Section 5.2 / Figure 4: Give the explicit mathematical definition of "interval". The current text and figure leave the definition ambiguous.
- Section6 / Figure 6: Expand "FLC" in the body text on first use and describe the mechanism by which the surrogate consumes learning-curve data.



### Details (typos and small clarifications)

- Section6.1: "We chose this environmen, because…" — should be "environment" (missing 't').
- Figure 4 caption: "supervised learning landscapes show significantly greater unimodality, which is unexpected, considering the strong unimodality observed in AutoML landscapes by Pushak & Hoos (2022)" — this sentence is logically inverted as printed. SL > RL unimodality is what P&H's findings would predict, hence *expected*. The intended meaning is likely either "expected" rather than "unexpected," or that the subject is RL landscapes showing *less* unimodality than expected. Please rephrase.
- Section 3.2: The sentence "every model was fitted 5 times to account for the variability in surrogate training" leaves ambiguous whether the 5 refits happen inside each CV fold during hyper-hyper search or only at final evaluation.
- Section 4.1: "The RL landscapes consist of three seeds per configuration to be comparable to LCBench" this is a big methodological choice that deserves justification in the body text and at least one ablation showing the impact (does the RL/SL surrogate-quality gap in Figure 1 shrink when all 10 RL seeds are used?).

---

##

---

> ### Author Response · Authors · 2026-05-28
>
> Dear Reviewer,
>
> Thank you very much for engaging with our work so carefully and raising many important points. We believe they will substantially improve the paper. Please find below our clarifications on the questions you raised and your requested changes. We will publish our revised paper by June 12th at the latest.
>
> Below, we address each of the reviewer’s requested changes:
>
> ### 1. Rescoping the central claims:
>
> **Observation 1:**
>
> >No statistical test accompanies the box-plots in Figure 1; the conclusion is drawn from overlapping distributions without inferential support.
>
> We agree that a statistical test would improve the soundness of our analysis. To quantify the differences in Figure 1, we applied a Mann-Whitney U test comparing the RL datasets (ARLBench, HPO-RL-Bench) against the supervised learning datasets (LCBench, PD1). The test confirms a statistically significant degradation in surrogate performance for RL tasks across all metrics ($p < 0.001$).
>
> Computing the common language effect size (probability of superiority) from the U-statistic demonstrates the qualitative size of this gap:
>
> 1. Kendall  tau: $p < 0.001$. A supervised learning task yields superior ranking correlation 60.7% of the time.
> 2. RMSE: $p < 0.001$. An RL task yields higher overall prediction error 64.0% of the time.
> 3. Top-10% RMSE: $p < 0.001$. An RL task yields worse errors 71.0% of the time.
>
> We will integrate these statistical findings and effect sizes into the revised paper.
>
> >Reading the table in Figure 1: HPO-RL-Bench (RL) achieves a Kendall τ of 0.77 and RMSE of 0.05 — better RMSE than both supervised-learning benchmarks, and Kendall τ essentially tied with PD1 (the best SL benchmark). PD1 (SL) has Kendall τ of 0.78 and RMSE of 0.13 — worse RMSE than HPO-RL-Bench (RL) and ARLBench (RL). The within-category variation between ARLBench and HPO-RL-Bench, and between LCBench and PD1, is at least as large as the between-category variation between RL and SL.
>
> We agree that Observation 1 needs to be refined to better align with the data. In particular, we need to distinguish more clearly between ARLBench and HPO-RL-Bench. We included HPO-RL-Bench in this analysis because it is an important piece of related work and further sheds light on RL surrogates as a reference point. However, the search spaces are substantially smaller than the ones in ARLBench, using only three hyperparameters and a small discrete search space (compared to e.g. 12 for PPO in ARLBench). Further, as we will discuss later in our rebuttal, we are going to include the ARLBench data for 10 seeds in Figure 1, which closes the gap between ARLBench and LCBench further. We will make the observation more precise in the revised paper, similar to the following:
>
> ---
>
> Performance prediction for RL algorithms with full configuration spaces is generally harder than for simple supervised learning tasks commonly found in surrogate benchmarks (e.g. LCBench). While on some tasks, similar performance can be achieved, RL shows more tasks with low prediction performance in both Kendall Tau and RMSE. Using smaller configuration spaces (as in HPO-RL-Bench) or collecting more random seeds per configuration allows to close this gap.
>
> ---
>
> We will further extend the analysis in the body text, discussing the differences between ARLBench and HPO-RL-Bench in further depth and how to interpret the boxplots as aggregated results over a representative number of fixed tasks.
>
> **Observation 2:**
>
> >The Observation as boxed is therefore a description of ARLBench's behaviour, not RL's. Additionally, PD1 (SL) shows comparable RMSE degradation on top-10% configurations (0.13 → 0.15) — undermining the supervised learning versus-reinforcement learning framing the Observation invokes.
>
> We agree that this observation is formulated too general, given our empirical data. As in Observation 1, we again have to distinguish between the substantially smaller search space of ARLBench and HPO-RL-Bench. Additionally, the similarity in Top-10% RMSE with PD1 needs to be considered accordingly. We would adjust observation 2 similarly to the following:
>
> ---
>
> Performance prediction quality for RL on the full algorithm configuration spaces is lower for better-performing configurations compared to simple supervised learning tasks (e.g. LCBench) and is instead comparable to more complex deep learning problems, e.g. PD1. Reduced configuration spaces as given in HPO-RL-Bench allow to close this gap, whereas increased number of seeds do not seem to narrow the gap.
>
> ---
>
> We believe that for both observations 1 and 2, the explicit discussion of the reduced configuration spaces and additional seeds will help to further motivate the interventions tested in Section 6 of the paper. We are planning to revise the paper accordingly.

---

> ### Author Response · Authors · 2026-05-28
>
> ### Test Section 6 interventions:
>
> >Choose at least one environment from the high-CI / high-RMSE outliers in Figure 1 (for instance, the ARLBench environments contributing Kendall τ values near 0.3) and re-run the key Section6 interventions there, even at smaller scale than the 2048-configuration LunarLander dataset.
>
> We agree that our current selection of the Lunar Lander environment for this use case is confounded. We are currently running experiments on Acrobot using PPO. Acrobot ranks among the worst environments by Kendall tau (0.4) and is also among the environments with the highest RMSE (0.2). This means it is among the worst environments in both metrics, which we believe qualifies it as an interesting candidate for studies or more meaningful interventions. The respective results will be added and discussed in Section 6 of the revised paper.
>
> ### 3. Improve Methodology Specification:
>
> >Section 3.1: The specific bounds used for [0,1] performance normalisation, per task or per environment, with justification. For RL environments where a theoretical maximum return is often ill-defined, state the choice (empirical max across all algorithms? random-policy baseline as lower bound? expert/human baseline?).
>
> The respective normalisation terms are already contained in the paper, in Appendix A in Table 5. However, we fully agree with the reviewer that the reference needs to be made clearer in the body text and an additional description of how normalisation was conducted should be added to the main text. PD1 and LCBench both use accuracy to measure performance and are thereby already naturally normalised between 0 and 1. For ARLBench and HPO-RL-Bench we used different strategies per benchmark suite as given below:
>
> 1. MiniGrid: The environments yield a sparse reward of +1 for a successfully finished episode and are thereby already naturally normalised between 0 and 1.
> 2. ALE: We used the normalised humanised scores to normalise performance. If the maximum performance in a landscape of an environment exceeded the human performance, we used the maximum performance instead as the upper bound.
> 3. For Box2D, ClassicControl (CC) and Brax we used random performance as the lower bounds and competitively tuned results from the literature in similar RL setups as the upper bounds. The respective references to these baselines are given in Appendix A.1.
>
> We will describe and reference this better in the main text of the revised paper.
>
> > Section 3.2: Explanation of the surrogate tuning protocol. State whether the 200-trial random search over surrogate hyperparameters uses the same 10-fold cross-validation folds as the reported final metric, or whether nested cross-validation is used. State explicitly when the 5 refits occur in the pipeline.
>
> We will extend the description of the surrogate tuning protocol. We hope the following description already helps to answer the open questions on the setup.
>
> All surrogate models were optimised via random search comprising 200 trials, with the exception of GPs, which were allocated 50 trials owing to their substantially smaller hyperparameter space. Each configuration was assessed using 10-fold cross-validation. For each fold, the data was partitioned into a 90% training set, a 5% validation set (utilised to identify the optimal configuration from the random search), and a 5% test set. The optimal configuration was determined by the maximum mean performance across all 10 validation folds. Subsequently, the generalisation capability of this selected model was computed as its mean performance across the 10 test folds. All surrogate performance metrics reported in this paper reflect these test set evaluations. To obtain an uncertainty value of the surrogate model optimisation performance, the random search was repeated with 5 different pseudo-random seeds, each yielding one surrogate model.

---

> ### Author Response · Authors · 2026-05-28
>
> >Section4.1 / Figure 1: State what each box-plot point represents (presumably one score per (algorithm, environment) task within a benchmark). Add visual differentiation between RL and SL benchmarks in the legend. Justify the choice to subsample RL data to 3 seeds for comparability with LCBench, and show whether the RL/SL gap shrinks when all 10 RL seeds are used.
>
> Each boxplot aggregates the performance for one benchmark suite; this is, each data point corresponds to one environment, algorithm and surrogate model fit. A clarification will be added to the revised paper in the caption of the figure. We will add a line in the legend that separates the RL and SL benchmarks in the revised paper.
>
> Regarding the number of seeds, we want to examine the difficulty of fitting surrogate models for RL in contrast to SL. Therefore, we want the number of pseudo-random seeds and the number of configurations to be as comparable as possible to ensure we have a fair comparison. We do agree, however, that the difference in performance with more seeds is an important consideration (as studied by us already later in the paper). We will extend Figure 1 by including both the boxplot for ARLBench with 3 seeds and with 10 seeds and will extend the analysis of the Figure in the body text of the revised paper accordingly.
>
> >Section 4.2 / Figures 2 and 3: For Figure 2, state what each violin distributes over (the three source configurations? the five refits? both? per-environment averages?) and how ground-truth LPI was computed (where do the grid evaluation points around each source configuration come from, and how many additional rollouts does this require). For Figure 3, write the explicit formula for "covered performance range" and clarify how aggregation was done.
>
> **Figure 2:** The violin plots are distributed over the three source configurations and for each source configuration it evaluates the five model fits for the surrogate mode. We noticed that due to a configuration error we only plotted the first three of the five model fitting seeds. The missing two seeds will be added in the revised paper and do not change the results. The grid-evaluation points are computed similarly to the DeepCAVE package (Segel et al.), where an appropriate neighbourhood relation is defined for each hyperparameter value (e.g. for a continuous hyperparameter a grid of all possible values in the hyperparameter’s configuration space). The ground-truth values are obtained by running the training for each respective configuration over 30 seeds. The surrogate values are obtained by evaluating the configurations under the surrogate model. Continuous hyperparameters were evaluated with 100 grid points and all neighbours for discrete or categorical values. For PPO this amounts to 705 configurations per LPI that need to be trained for in order to obtain ground-truth performance values. We will add this description in the revised paper.
>
> **Figure 3:** The covered performance range is the upper confidence interval - the lower confidence interval. A formal definition will be added in the revised paper.
>
> >Section 5.2 / Figure 4: Give the explicit mathematical definition of "interval". The current text and figure leave the definition ambiguous.
>
> For a specific configuration $c$ with mean task performance $p_c$ for an algorithm, the upper and lower bounds of performance interval are defined as $p_c \pm \epsilon$ with $\epsilon \in [0, 0.01, 0.02,0.05,0.1]$.
>
> We will clarify this in the revised paper.
>
> >Section6 / Figure 6: Expand "FLC" in the body text on first use and describe the mechanism by which the surrogate consumes learning-curve data.
>
> We agree that FLC (full learning curve) should be defined and described in the body text of the paper. Training on the full learning curve means the surrogate model gets the number of training steps normalised to 0 and 1 as an additional input to the hyperparameter configuration to predict performance at intermediate evaluations. We will give formal definitions for the RL setup and AutoML terms in our revised paper (e.g. hyperparameter landscapes, performance prediction, surrogate models). Consistent with this definition of the surrogate model, the FLC learning extends the surrogate model to $\hat{f}_{\mathcal{A},\mathcal{M}}: \Lambda \times [0, 1] \to \mathbb{R}$ with $\Lambda$ being the hyperparameter configuration space, $\mathcal{M}$ the respective MDP and $\mathcal{A}$ the RL algorithm. The additional interval $[0,1]$ represents the time-step to predict performance for. We will include such a definition in our revised paper.

---

> ### Author Response · Authors · 2026-05-28
>
> The reviewer raised several minor issues, which we are addressing below:
>
> >Figure 4 caption: "supervised learning landscapes show significantly greater unimodality, which is unexpected, considering the strong unimodality observed in AutoML landscapes by Pushak & Hoos (2022)" — this sentence is logically inverted
>
> This is indeed inverted and will be fixed in the revised paper.
>
> >Section 3.2: The sentence "every model was fitted 5 times to account for the variability in surrogate training" leaves ambiguous whether the 5 refits happen inside each CV fold during hyper-hyper search or only at final evaluation.
>
> The five fits are done at the optimisation level of the surrogate models. This is, each random search optimising the surrogate model hyperparameters was repeated for five different pseudo-random seeds. We hope our earlier clarification on the surrogate tuning protocol helps to answer this question further.
>
> >"The RL landscapes consist of three seeds per configuration to be comparable to LCBench" this is a big methodological choice that deserves justification in the body text and at least one ablation showing the impact (does the RL/SL surrogate-quality gap in Figure 1 shrink when all 10 RL seeds are used?).
>
> The reviewer raises a valid point here. We aim to address this in the revised paper by adding both the ARLBench results for 3 seeds and 10 seeds in Figure 1 and extending our discussion of the figure accordingly. Indeed, using 10 seeds substantially closes the gap between ARLBench and LCBench. This is consistent with our later results that RL has substantially higher variance in its performance values and more seeds help to obtain better surrogate models.
>
> Again, we would like to thank the reviewer for the detailed assessment of our work and his proposed improvements.

---

### Decision · Action_Editor_Wgeo · 2026-06-10

**Recommendation:** Reject

**Additional Comments:**

NA

**Audience:**

Yes

**Audience Explanation:**

The paper explores the effectiveness surrogate models for predicting the performance of an algorithm with a particular set of hyperparameters; popular and effective in supervised learning, but under explored in RL. This is of great interest to the community because such approaches could dramatically speed up algorithmic RL research and make real-world deployments more feasible. The paper is potentially very practically relevant as well: providing recommendations for practitioners on how to use surrogates and analysis of when surrogates fail.

**Claims And Evidence:**

No

**Claims Explanation:**

As the reviewers have noted in their extremely detailed reviews, there are many things to change her before this paper could be accepted. All three knowledgable reviewers voted to reject, but noted the authors should certainly resubmit once substantial changes where made. The author response clearly indicates the authors are well aware of the scope of the changes needed.

This is certainly a promising piece of work, but the reviewers must review the work as submitted, not a future version of the paper with major revisions. They all agreed the revisions needed were not minor.

**Resubmission Of Major Revision:**

The authors may consider submitting a major revision at a later time.